# Occult polyclonality of preclinical pancreatic cancer models drives in vitro evolution

Maria E. Monberg [1,2,3,8] ✉, Heather Geiger [4,8], Jaewon J. Lee [1,3], Roshan Sharma[4], Alexander Semaan [1,3], Vincent Bernard[1,3], Justin Wong [5], Fang Wang[6], Shaoheng Liang [6], Daniel B. Swartzlander[1,3], Bret M. Stephens[1,3], Matthew H. G. Katz[7], Ken Chen [6], Nicolas Robine [4,9], Paola A. Guerrero [1,3] & Anirban Maitra [1,3,9]

Heterogeneity is a hallmark of cancer. The advent of single-cell technologies has helped uncover heterogeneity in a high-throughput manner in different cancers across varied contexts. Here we apply single-cell sequencing technologies to reveal inherent heterogeneity in assumptively monoclonal pancreatic cancer (PDAC) cell lines and patient-derived organoids (PDOs). Our findings reveal a high degree of both genomic and transcriptomic polyclonality in monolayer PDAC cell lines, custodial variation induced by growing apparently identical cell lines in different laboratories, and transcriptomic shifts in transitioning from 2D to 3D spheroid growth models. Our findings also call into question the validity of widely available immortalized, non-transformed pancreatic lines as contemporaneous "control" lines in experiments. We confirm these findings using a variety of independent assays, including but not limited to whole exome sequencing, single-cell copy number variation sequencing (scCNVseq), single-nuclei assay for transposase-accessible chromatin with sequencing, fluorescence in-situ hybridization, and single-cell RNA sequencing (scRNAseq). We map scRNA expression data to unique genomic clones identified by orthogonally-gathered scCNVseq data of these same PDAC cell lines. Further, while PDOs are known to reflect the cognate in vivo biology of the parental tumor, we identify transcriptomic shifts during ex vivo passage that might hamper their predictive abilities over time. The impact of these findings on rigor and reproducibility of experimental data generated using established preclinical PDAC models between and across laboratories is uncertain, but a matter of concern.

[1] Department of Translational Molecular Pathology, The University of Texas MD Anderson Cancer Center, Houston, TX, USA. [2] University of Texas MD Anderson Cancer Center Graduate School of Biomedical Sciences, Houston, TX, USA. [3] Sheikh Ahmed Center for Pancreatic Cancer Research, The University of Texas MD Anderson Cancer Center, Houston, TX, USA. [4] New York Genome Center, New York, NY, USA. [5] Department of Epidemiology, The University of Texas MD Anderson Cancer Center, Houston, TX, USA. [6] Department of Bioinformatics and Computational Biology, The University of Texas MD Anderson Cancer Center, Houston, TX, USA. [7] Department of Surgical Oncology, The University of Texas MD Anderson Cancer Center, Houston, TX, USA. [8] These authors contributed equally: Maria E. Monberg, Heather Geiger. [9] These authors jointly supervised this work: Nicolas Robine, Anirban Maitra. ✉email: m.monberg219@gmail.com

Pancreatic Ductal Adenocarcinoma (PDAC) is a highly lethal malignancy, projected to be the second-leading cause of cancer-related death in the United States by 2030[1,2]. With a dismal 5-year survival rate of 10%[3] and few clinically meaningful therapeutic advances in recent years, the need for clinical and research advances is urgent. Preclinical models of PDAC represent essential prerequisites for advancing cancer research and experimental therapeutics in this lethal disease. The most commonly used and widely published preclinical models of PDAC that have been, and continue to be, leveraged in research laboratories globally are adherent cell lines distributed by the American Type Culture Collection (ATCC). For example, a perusal of current Google Scholar metrics demonstrates ~35,000 publications for PANC-1 and ~20,000 publications for MiaPaCa2, two prototypal ATCC cell lines nearly ubiquitous in every PDAC cancer research laboratory's incubator. Included in many of these publications, and typically used as contemporaneous "control" cells in experiments, are two immortalized, non-transformed human pancreas-derived cell lines – so-called Human Pancreatic Ductal Epithelial (HPDE) and Human Pancreatic Nestin-expressing (HPNE) cells. While the reach of these preclinical models in furthering our knowledge of PDAC biology is extensive[1–4], several inherent assumptions have been made regarding the stability of the genomic and transcriptomic clonal architectures of these cells over time, including the fidelity to "normal" parameters for HPDE and HPNE during ex vivo passage.

The justifiably increased focus on rigor and reproducibility in scientific research studies has meant more stringent requirements for authentication of cell line resources, but the microsatellite assays typically used for confirmation of cell identity provide minimal insights into genomic and transcriptomic variabilities that might arise when the apparently identical line is propagated at different laboratories[5] or in disparate culture conditions, such as two-dimensional (2D) versus three-dimensional (3D) growth. A recent seminal publication by Ben-David et al.[6] identified widespread genomic and transcriptomic variabilities in single cell clones isolated and expanded from the same parental cancer cells, which further translated into differences in drug responsiveness in vitro, suggesting a staggering level of pre-existing intra-culture heterogeneity within cancer cell lines. Although PDAC cells from surgical resections were not included in this analysis, it underscores the need for an in-depth benchmarking study using commonly used cancer and non-transformed "control" cells, in order to fully glean the extent of intra- and inter-culture heterogeneity that exists in these preclinical model systems.

In addition to the use of adherent cell lines, efforts at better recapitulating the biology of in vivo disease have led to the burgeoning adoption of patient-derived organoid (PDO) models as an ex vivo platform in the PDAC field. Encouragingly, early passages of PDAC PDOs maintain genomic and transcriptomic features of the tumor of origin, and harbor a gene signature of response to commonly used cytotoxic agents, enabling therapeutic prediction[7,8]. Nonetheless, whether the clonal architecture of PDOs remains stable over more prolonged ex vivo propagation, considering both the time in culture and the lack of extrinsic cues from the tumor microenvironment in vivo, remains less studied.

Here we present an in-depth single cell genomic and transcriptomic assessment of clonal heterogeneity in a panel of established and globally utilized PDAC cell lines (Panc-1, Mia-PaCa2, HPAF-II, and BxPC-3), immortalized "control" cells (HPNE and HPDE), and in an independent PDO which is compared to its earlier passage prior to prolonged ex vivo propagation. We demonstrate that pancreatic cell lines—neoplastic and non-neoplastic—are composed of complex sub-clonal heterogeneity at single cell resolution, which is unlikely to be detectable by conventional "bulk" profiling, despite extensive "traditional phenotypic" characterization of these lines[9,10]. Unexpectedly, HPDE cells harbor substantial genomic alterations and a transcriptome that essentially resembles cancer cells, questioning their use as a "control" line in research studies. We further demonstrate marked transcriptomic differences (including expression-based subtype classification) in microsatellite authenticated MiaPaca2 cells obtained from three independent sources. Additionally, we report a marked transcriptomic and epigenetic divergence (incorporating the appearance of distinct therapeutically actionable pathways) when adherent (2D) parental Panc1 cells are grown in 3D cultures. Finally, we describe the significant genomic and transcriptomic divergence of a later-passage PDO from an earlier passage, reiterating the importance of limiting the window for ex vivo therapeutic prediction and other biological experiments in this model type. Overall, our findings provide data-driven benchmarks for the limitations of the most commonly utilized preclinical models and platforms in PDAC research, with implications for the rigor and reproducibility of data generated in the in vitro setting.

## Results

**Single cell analysis identifies clonal heterogeneity of pancreatic cancer and non-transformed cells.** We cultured six pancreatic cell lines, including three MiaPaca2 lines from independent sources and a pair of Panc1 lines subjected to differing growth conditions, for a total of nine samples analyzed with single-cell transcriptomic profiling (scRNAseq, 10x Genomics) and seven samples also analyzed with genomic copy number variant detection (scCNVseq, 10x Genomics). All samples were submitted for fingerprinting via MD Anderson Core Facilities prior to sequencing to confirm cell line identity. Visualization of the combined scRNA-seq data (77,068 cells) with UMAP[11] (Fig. 1a) revealed substantial heterogeneity across the cell lines, including only partial overlap in apparently identical cells obtained from distinct sources or cultured under different conditions (see later). We first validated that cell lines of published epithelial or mesenchymal differentiation state maintained key marker identities[12] (Fig. 1b). For example, we observed high expression of the epithelial transcripts EPCAM and KRT8 in the previously described "epithelial" lines, HPAF-II and BxPC3, and conversely, the prototypal mesenchymal transcript vimentin (VIM) was essentially restricted to the previously described "mesenchymal" lines Panc1 and MiaPaca2. HPNE cells, which were originally derived from a nestin (NES) and NOTCH2 expressing pancreatic progenitor cellular population (Lee 2005)[13], were confirmed to retain both markers (Fig. 1b). Notably, HPNE cells, along with Panc1 and MiaPaca2, also demonstrated extensive CD44 expression, underscoring the enrichment of putative stem-like cells in culture. We observed that cell lines that had undergone differing culture conditions (Panc1 2D, Panc1 3D) or were from distinct laboratories of origin (MiaPaca2-A, B, and C) were still more transcriptionally similar to each other than to the other cell lines (Fig. 1a).

To quantify transcriptional similarity, we computed phenotypic distance, defined as the distance between the samples in the diffusion space (see Methods). Upon projecting the distances onto a cladogram, we observed that all three culture variations of MiaPaCa2 cell lines fall into a common clade (MP2-A, MP2-B, MP2-C), while Panc1 cells, irrespective of growth as monolayer or in 3D, branch into their own clade (Supplementary Fig. 1a, b). To ensure rigor and reproducibility in our work, and to assure the reliability of our analysis when yet simpler algorithms are applied, we confirmed our findings by running a Pearson correlation

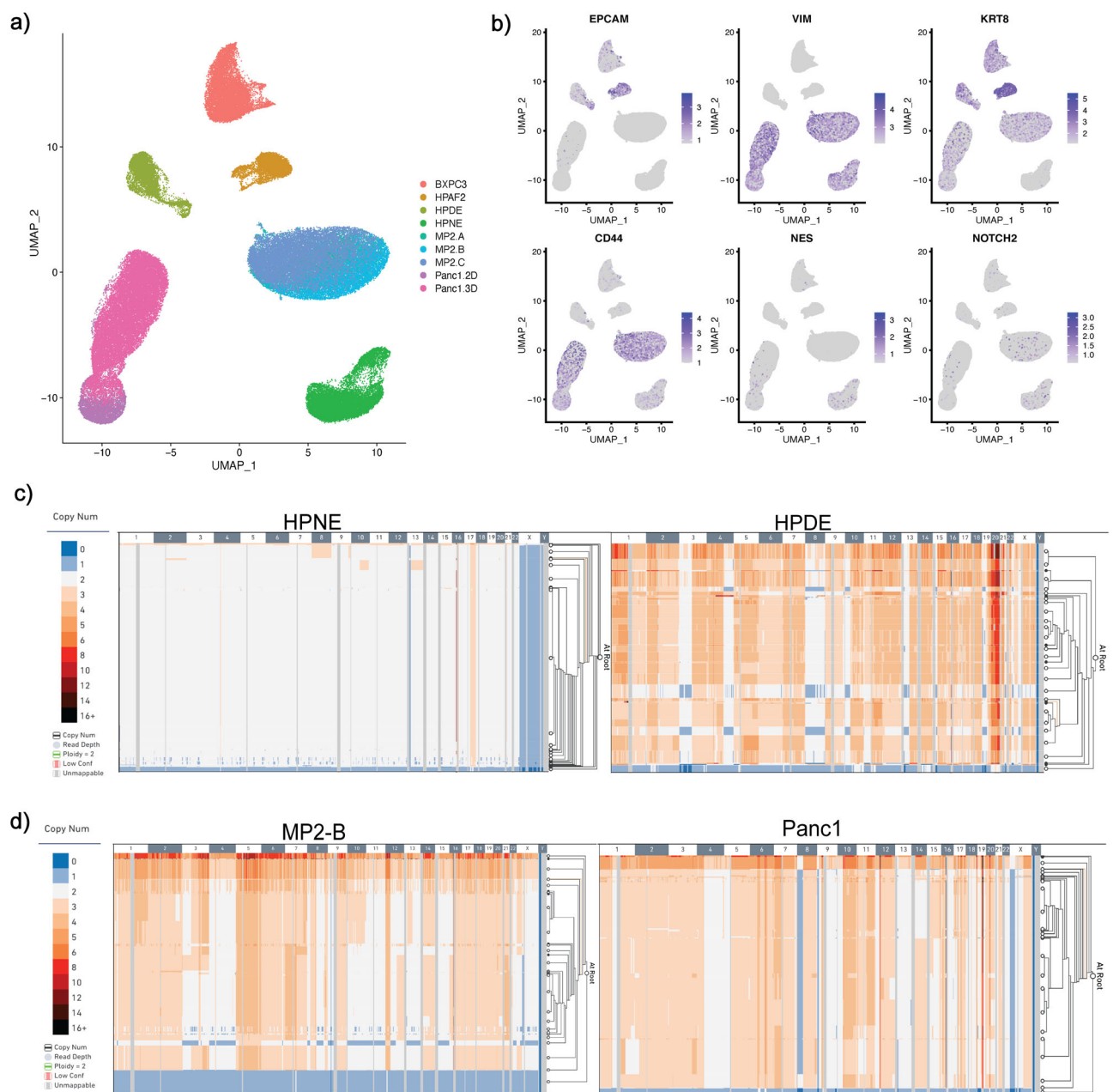

**Fig. 1 PDAC Cell Lines display heterogeneity at single-cell level. a** Uniform manifold approximation projection (UMAP) plot of single cells from PDAC cell lines used in this study. **b** UMAP Feature Plotting for known cell-type markers (EPCAM, MUC1 for epithelial; vimentin, and KRT8 for mesenchymal cellular origin). **c**, **d** Representative scCNV plots for cell lines. Columns indicate chromosomes (numbers labeled in gray and white), rows indicate individual cells organized into clonal clades. Phylogeny determined using 10× Genomics scCNV analysis software.

analysis (data available upon request)[14]. In doing so, we report that the overarching structure of how cell line clades are organized remains consistent with what was determined by diffusion mapping.

Next after the scRNA-seq data, we interrogated paired scCNVseq data. The scCNV analysis demonstrated subclonal heterogeneity within each of the pancreatic cell lines, with various divergent genome-wide copy number alterations identified within subsets of each parental entity (Fig. 1c, d, Supplementary Fig. 2). We found 10x Cell Ranger DNA software had underestimated scCNV event calls, likely due to low sequencing coverage susceptible to this data type. To adjust for underestimation, we used our in-house bulk WES data for HPDE and HPNE as bioinformatic controls to re-estimate baseline CNV events, and

then normalized cells across each sample to create a "pseudo-bulk" profile based on copy number ratios for each cell line (Supplementary Fig. 3a). In this "pseudo-bulk" summary, previously described hallmark genomic events characteristic of PDAC lesions[1–6] were readily observed, albeit in some cases only in a subset of genomic subclones for each line. Specifically, amplifications in regions of chromosomes 1,2,3,5,7,8,11,14, and 17, and losses at regions of chromosomes 3 (HPDE), 9 (BxPC3, MP2-B, MP2-C, HPDE), and 13 (HPAF-II) were distinguishable.

Finally, we mapped the clonal families of each cell line to genes known to be amplified, deleted, or mutated in PDAC (see *Methods*), and uncovered extensive heterogeneity in amplification and deletion profiles on a per-locus basis (Supplementary Fig. 4). We report that HPNE has a benign CNV profile at these loci, with

little heterogeneity in copy number across clones of this culture (Supplementary Fig. 4a). Conversely, the BxPC3 cell line, which is historically used as a *KRAS* wild type PDAC cell line, harbors copy number > = 4 in nearly all clones at the *TERT* gene locus, wherein over 2/3 of cells have a copy number for TERT more than 1 above the mean ploidy (Supplementary Fig. 4b).

BxPC3 TERT copy number was also confirmed via fluorescence in situ hybridization (FISH), as compared to copy number events at this locus in cell lines HPDE, HPNE and HPAF-II (Supplementary Fig. 5). Of interest, a number of BxPC3 clones harbored amplifications at the BRAF locus (Supplementary Fig. 4b). It is known that in contrast to melanoma, cases of *KRAS* wild type PDAC, including the BxPC3 cell line itself, acquire in-frame deletions of *BRAF* that result in constitutive activation of the kinase domain (Chen 2016)[15]. However, amplification of the mutated "driver" oncogenic locus (e.g. *KRAS*, *BRAF*, *EGFR*) is not unusual, as it can provide additional survival and growth cues.

**Single cell profiling reveals pitfalls of commonly utilized immortalized pancreatic controls.** The Human Pancreatic Duct Epithelial (HPDE) cell line is a widely used model system for human pancreatic carcinogenesis[13,16] typically as a non-transformed, immortalized "control". HPDE cells were derived from explanted human pancreatic ductal epithelium obtained from a pancreas resected for benign etiology, and immortalized using retrovirus-mediated expression of human papillomavirus 16 (HPV16) oncoproteins E6 and E7[16]. The immortalizing effects of E6 and E7 are mediated primarily by targeting of the p53 and retinoblastoma (Rb) tumor suppressor genes[17]. Parental HPDE cells are wild type at the endogenous *KRAS* locus, non-transformed in vitro, and non-tumorigenic in mice, unless transformed through ectopic expression of mutant *KRAS*, in combination with other tumor promoting genomic events[18].

We thus hypothesized that the HPDE cell line would be transcriptionally distanced from the PDAC-derived lines, given its broad experimental application as a control line. However, our scRNAseq interrogation instead shows HPDE is most transcriptionally similar to HPAF-II, and on a distinct clade from the second commonly used "control" line, HPNE (Supplementary Fig. 1b). On scCNV profiling, we find that HPDE cells retain a subclonal architecture akin to a neoplastic genotype in some clones (Fig. 1c). We also find evidence of distinct amplification and deletions at loci linked to key PDAC-progressor genes (Supplementary Fig. 4b). We conducted "conventional" bulk whole exome sequencing (WES) on HPDE cells curated in our laboratory from the same pool used to generate scCNV and scRNAseq datasets (see *Methods*). This confirmed that HPDE cells had undergone further bi-allelic losses of chromosomal arms 3p, 8p, and 9p, which are all "hot spots" housing known tumor suppressor genes in PDAC and other solid tumors (Fig. 2b, top), and this finding was validated in the "pseudo-bulk" CNV profile generated from scCNV data (Fig. 2b, bottom). Notably, all HPDE subclones harbor an amplification event (CN > 10 in a minor fraction of subclones) at chromosome 20q (Fig. 2d), bookending the *AURKA* gene locus within the most common region of overlap. Almost no heterogeneity was detected in the amplification event, reiterating that all HPDE single cells sequenced have undergone this amplification at some level (Fig. 2d, bottom purple bar measuring near-0 Heterogeneity value). FISH confirmed AURKA amplification in HPDE (Supplementary Fig. 5), indicating that high AURKA copy numbers in scCNV clones were not a function of technical artifacts in this dataset. Examination of *AURKA* transcript expression, when HPDE scRNAseq data was normalized to HPNE scRNAseq data, showed

a spectrum of expression levels in HPDE, at least 20% of cells (presumably representing the subclones with highest levels of genomic amplification) expressing 2-4 fold higher expression of *AURKA* transcripts compared to HPNE cells (Fig. 2e). The encoded protein Aurora kinase A is commonly overexpressed in pancreatic cancers, and contributes to both chromosomal instability through destabilization of the mitotic spindle assembly and towards tumor progression via phosphorylation of substrate proteins[19,20]. One can speculate that the widespread genomic perturbations observed in HPDE are at least partially a consequence of instability introduced by aberrant *AURKA* activity, in conjunction with the functional inactivation of p53 caused by HPV oncoprotein expression. The functional import of *AURKA* amplification in HPDE, a purportedly "control" cell used in PDAC research, may have far-reaching consequences in the PDAC preclinical space, including in experimental therapeutics studies.

These findings in HPDE demarcate an important distinction from the second immortalized cell line commonly used as an experimental "control", Human Pancreatic Nestin-expressing (HPNE) cells[21]. In contrast to HPDE, HPNE cells are non-ductal in derivation (as demonstrated by lack of epithelial markers, and widespread expression of progenitor cell transcripts *NES*, *NOTCH2* and *CD44* on scRNA-seq (Fig. 1b). HPNE cells were immortalized using retroviral transduction of the human telomerase reverse transcriptase (*hTERT*), which has been used in more recent times for derivatizing immortalized epithelial cells. Unlike HPDE cells, scCNV sequencing demonstrates that HPNE cells have not evolved a divergent subcellular taxonomy, but rather are homogenous in their copy number profiles when viewed both broadly and at key oncogenic loci (Fig. 2a, Supplementary Fig. 4a). This scCNV finding is consistent with WES analysis of HPNE (Fig. 2a, top), which shows a "flat" bulk segmentation profile, and is confirmed on the "pseudo-bulk" profile generated from scCNV data (Fig. 2a, bottom).

While we do not observe amplifications or deletions of most of the HPNE genome, there is a discernible amplification event at chromosome 17q. As with HPDE, we interrogated genes contained within the amplified 17q locus (Fig. 2c), and *BRCA1*, among others, stood out as a candidate gene. When interrogating scRNAseq data, we report that at least 24% of HPNE cells in culture highly express BRCA1 transcripts relative to HPDE cells (Fig. 2e). The encoded Brca1 protein is a component of the homologous recombination repair (HRR) machinery, and *BRCA1* is uncommonly mutated in PDAC (~1%), typically in the germline, and associated with HRR defective cancers[22,23]. While the direct mechanism by which 17q would have been amplified remains to be investigated, we speculate that in the case of HPNE, the relative "genomic quiescence" reported here may be due, in part to the overexpression of *BRCA1* enabling DNA repair mechanisms.

Ultimately, while HPNE does not exhibit the subclonal heterogeneity observed with HPDE, its lack of a ductal transcriptional profile and enrichment in markers associated with progenitor populations of uncertain histogenesis need to be factored in the use of these cells as an appropriate control for PDAC cells. To that end, the need for developing improved preclinical patient-derived "control" models for PDAC research is apparent from our findings.

**Custodial variability of MiaPaca2 cell lines drives transcriptomic heterogeneity.** An identical parental line, when maintained under comparable passaging conditions in different laboratories (MiaPaca2), reflects a degree of custodial variability that impacts the translational value of that cell line as a controlled

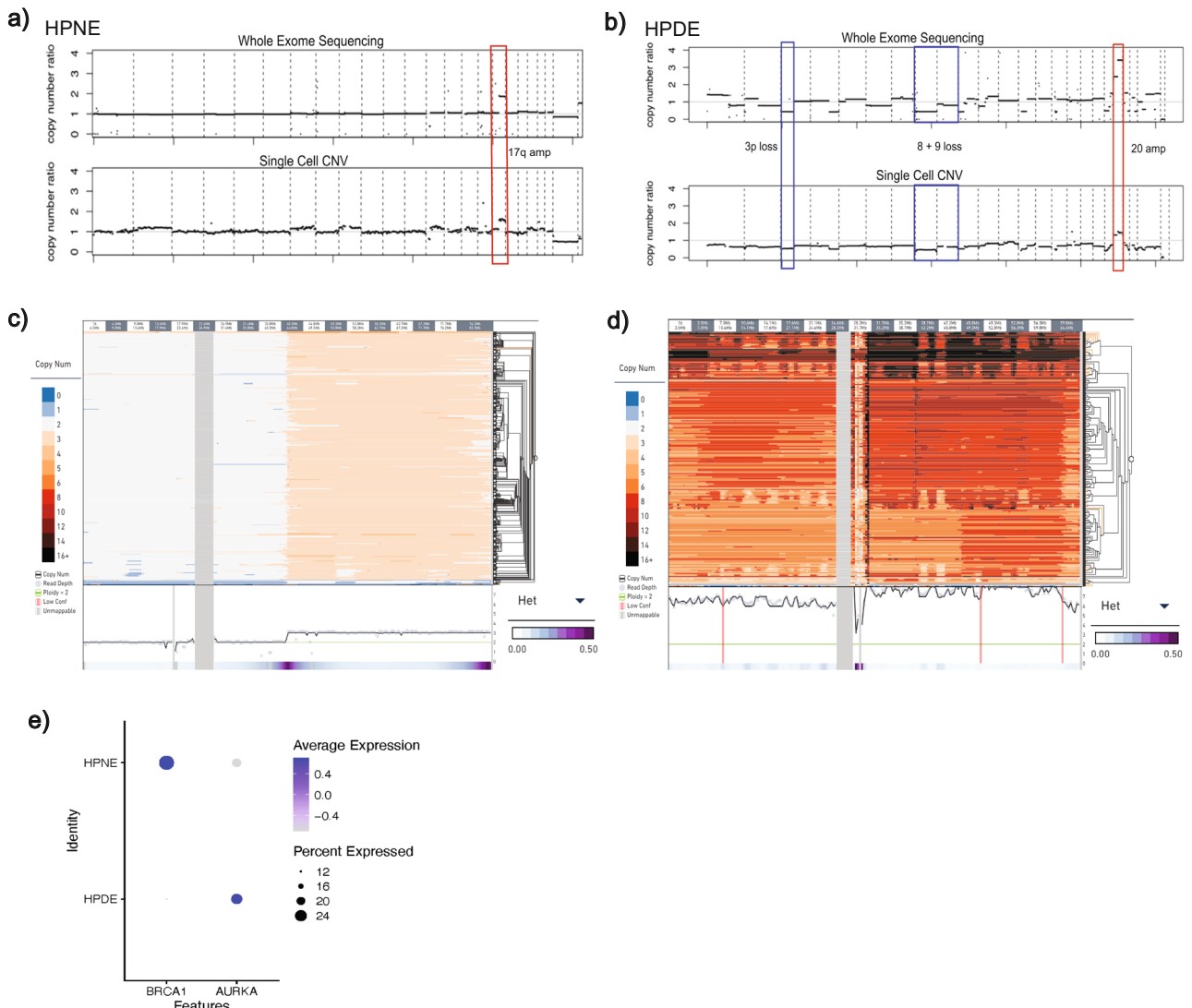

**Fig. 2 Genomic characterization of HPDE and HPNE normal control cell lines. a** scCNV comparison to WES of HPNE cell line depicting amplified 17q as the only notable CNV event. **b** scCNV comparison to WES of HPDE cell line. CNV events representing losses of chromosomal arms 3p, 8p, and 9p and amplifications at chromosome 20. **c** scCNV high-resolution cell phylogeny of HPNE for all chromosome 17 locations showing ploidy = 3 for all cells at 17q arm, region inclusive of BRCA1 loci. Columns indicate chromosomal intervals measured in Mb, rows indicate individual cells. Chromosomal regions depicted (labeled along top x-axis) are representative of CNV segments outlined in red in Fig. 1d. **d** scCNV high-resolution cell phylogeny of HPDE for all chromosome 20 locations showing ploidy >3 (as high as 13 in some cells at some locations) for all cells. Columns indicate chromosomal intervals measured in Mb, rows indicate individual cells. Chromosomal regions depicted (labeled along top x-axis) are representative of CNV segments outlined in red in Fig. 1e. Corresponding scRNA data shows elevated expression of AURKA, located within amplified HPDE region as a potential target of amplification. **e** scRNAseq data of HPNE and HPDE cell lines shows elevated expression of BRCA1 in >20% of HPNE cells, increased AUKRA expression in >20% of HPDE cells.

model. Previous work challenged the long-maintained assumption in cancer biology research that cell lines are clonal, stable entities, and explored the existence of phenotypic variability between widely used HeLa[5] and MCF7[6] cell lines via cross-laboratory multi-omic comparisons. In such studies, cultured cells were maintained in uniform conditions after being obtained from different laboratory settings. Subsequent analysis revealed extensive inter-laboratory heterogeneity with respect to phenotype, copy number variations, and even drug response experiments, elucidating the high degree of variability underlying presumed "same" cell lines. Now, in light of single-cell analytics, we sought to investigate the extent of deviation from assumptions of clonality, both within and between cultures, in PDAC cell lines.

The MiaPaca2 cell line was initially generated from a primary PDAC lesion of a 65-year-old male, where the tumor had involved the pancreatic body, tail, and demonstrated periaortic

invasion[24]. We assayed 3 "strains" of the MiaPaca2 cell line: one ordered directly from the American Type Culture Collection (ATCC), located in Gaithersburg, MD (MP2-A), a vial originating from an academic laboratory in Boston (MP2-B) and one that had been grown in our laboratory at MD Anderson Cancer Center (MP2-C). Following fingerprinting of the MiaPaca2 cells, we confirmed using ddPCR that all three versions of the cell line carried the same activating KRAS[G12C] mutation (Supplementary Fig 6a–c), as previously published in characterization studies of this cell line[24–26].

We report phenotypic divergence between these assumptively "identical" cultures that appears to be driven by changes at the RNA level. In our comparative scRNA-seq analysis of these samples, we find that the MiaPaca2 lines segregate mostly independently, with only a single small cluster where all three overlap (Fig. 3a, b). MP2-A and MP2-B also share an additional

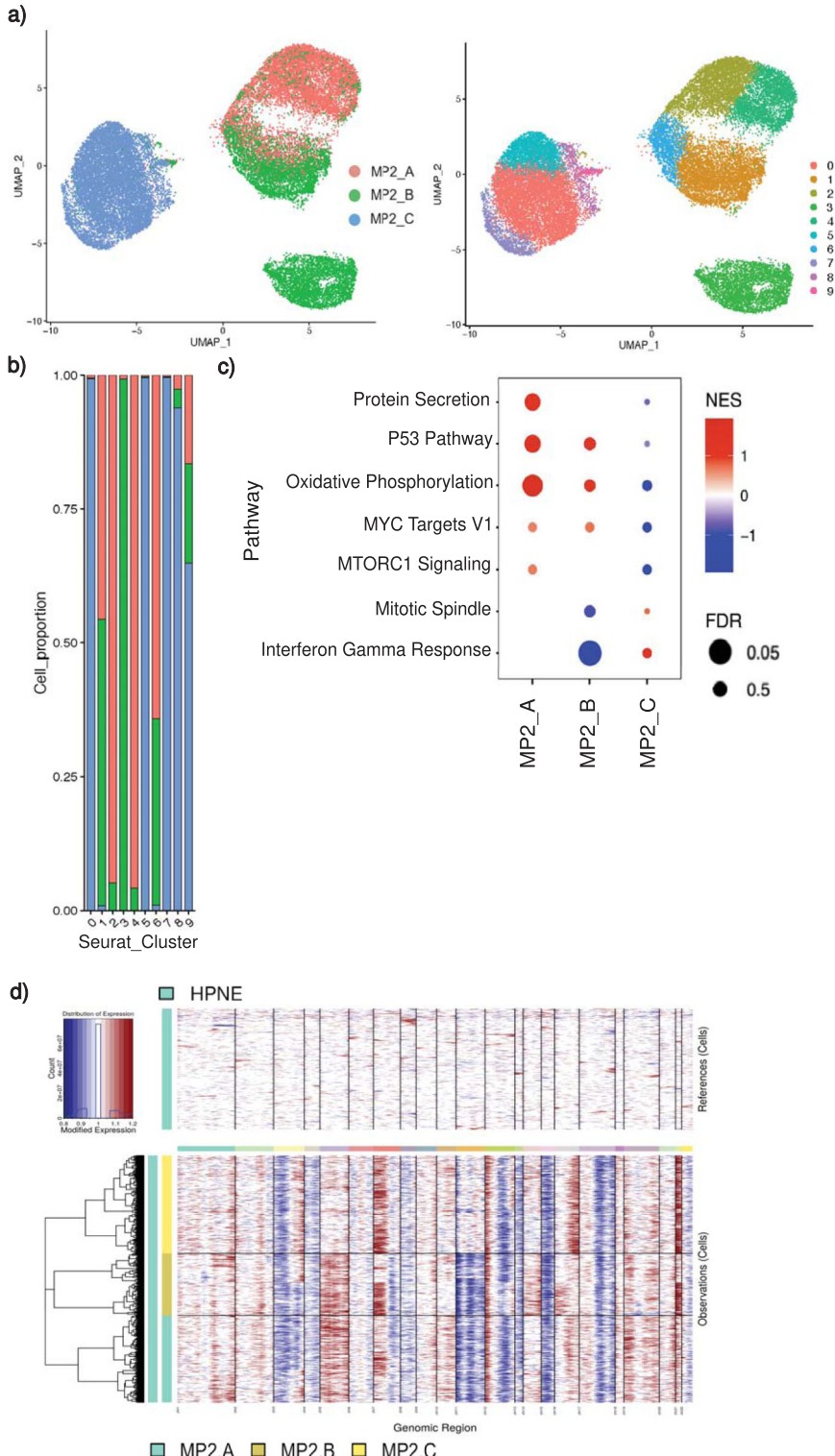

**Fig. 3 Custodial variability of MiaPaca2 cell lines drives transcriptomic heterogeneity. a** UMAP overlaying MP2 samples and their distinct clusters. **b** Bar graph displaying distribution of cells per MP2 culture across clusters; MP2-A in red, MP2-B in green and MP2-C in blue. **c** Bubble plot showing enrichment of pathways in MP2 cultures compared to each other with normalized enrichment score (NES) on the x-axis. Size of the bubble represents false discovery rate (FDR); red indicates upregulation, blue indicates downregulation of the pathway. **d** inferCNV heatmaps derived from scRNAseq data of all three MP2 samples, using HPNE as an analytical reference.

region of overlap, but distinct from MP2-C. Gene Set Enrichment Analysis (GSEA) was performed to identify general pathways that are differentially altered between the samples (Fig. 3c). MP2-A is enriched in GSEA hallmark pathways related to p53, oxidative phosphorylation, and protein secretion. In contrast, MP2-B downregulated GSEA hallmark interferon gamma response, and MP2-C has no statistically significant enrichment in hallmark pathways based on scRNA-seq GSEA analysis (Fig. 3c).

We find evidence for a genetically similar ancestral clone between the MP2-B and MP2-C strains, including several examples of chromosomes ubiquitously amplified in one strain versus subclonally amplified in the other (chromosomes 3q, 5, 7p) (Fig. 1d, Supplementary Fig 2c). We were also able to orthogonally validate this when we applied InferCNV (Tickle 2019)[27] to scRNAseq MP2 data (Fig. 3d). Further, these strains markedly differ in amplification and deletion events that correspond to chromosomal loci implicated in PDAC. For example, in MP2-C we find that nearly all cells have copy number $>= 4$ for oncogenes BLC6, PIK3CA, and BRAF. In contrast, MP2-B has lower copy number for these genes in a large fraction of cells, despite similar overall ploidy levels (Supplementary Fig. 4b). These initial findings demonstrate the heterogeneity, when viewed at the single cell level, across cell lines assumed to be derived from the identical parental origin.

**Evolution of divergent genomic subclones in MiaPaca2 strains has transcriptomic implications.** While profiling at single-cell resolution makes evident the transcriptional and genomic heterogeneity of PDAC cell lines, integrating subclonal lineages and relationships between these independent datasets is imperative for uncovering the impact of genomic alterations on transcriptomic diversity. Previous work in which DNA and RNA were assayed in the same cells has shown that many genes display copy-number gene-dosage effects on transcriptional expression[28–30]. Building on this finding, Campbell et al.

developed the clonealign algorithm[31], which established a framework for how genomic-transcriptome associations may be mapped in single-cell datasets derived from the same parental culture, but generated using individual assays. Using paired scRNA and scCNV from MiaPaca2 cells as a proof-of-concept for this method, we applied segmentation and copy number profiling from the scCNV data to clusters identified in scRNA transcriptomic profiles using clonealign (see *Methods*). An important caveat of this method is that clonealign's inherent design filtered out a large number of cells sequenced by scCNVseq. Using clonealign filtering criteria to scCNV-derived MP2-B cells (Fig. 4a), two distinct subclones were defined: one with 4 copies of chromosome arm 7p and 2 copies of chr10 (p and q), versus one with 3 copies for all of these chromosome arms. Of note, the clone with 3 copies of 7p (instead of 4 copies) also had lower copy number for a subset of 12q (2 vs. 3). Applying the same clonealign filtering criteria to scCNV-derived cells from MP2-C (Fig. 4d), three distinct subclones were defined: 1. With lower copy number (2 vs. 3) in chromosome 10 (p and q arms) and 14q. 2. With copy number $= 3$ in all of chr5, chr10, and chr14q. 3. With higher copy number (4 vs. 3) in chr5, and lower copy number (2 vs. 3 copies) in 10p, but higher in 10q and 14q (3 vs. 2 copies).

After the major subclonal populations were established from scCNV profiles, we used clonealign to identify subclones in the scRNA data that could be reliably mapped back to scCNV profiles (see *Methods*). Notably, we find that for MP2-B, the two unique genomic profiles we see in the scCNV data (Fig. 4a) correspond

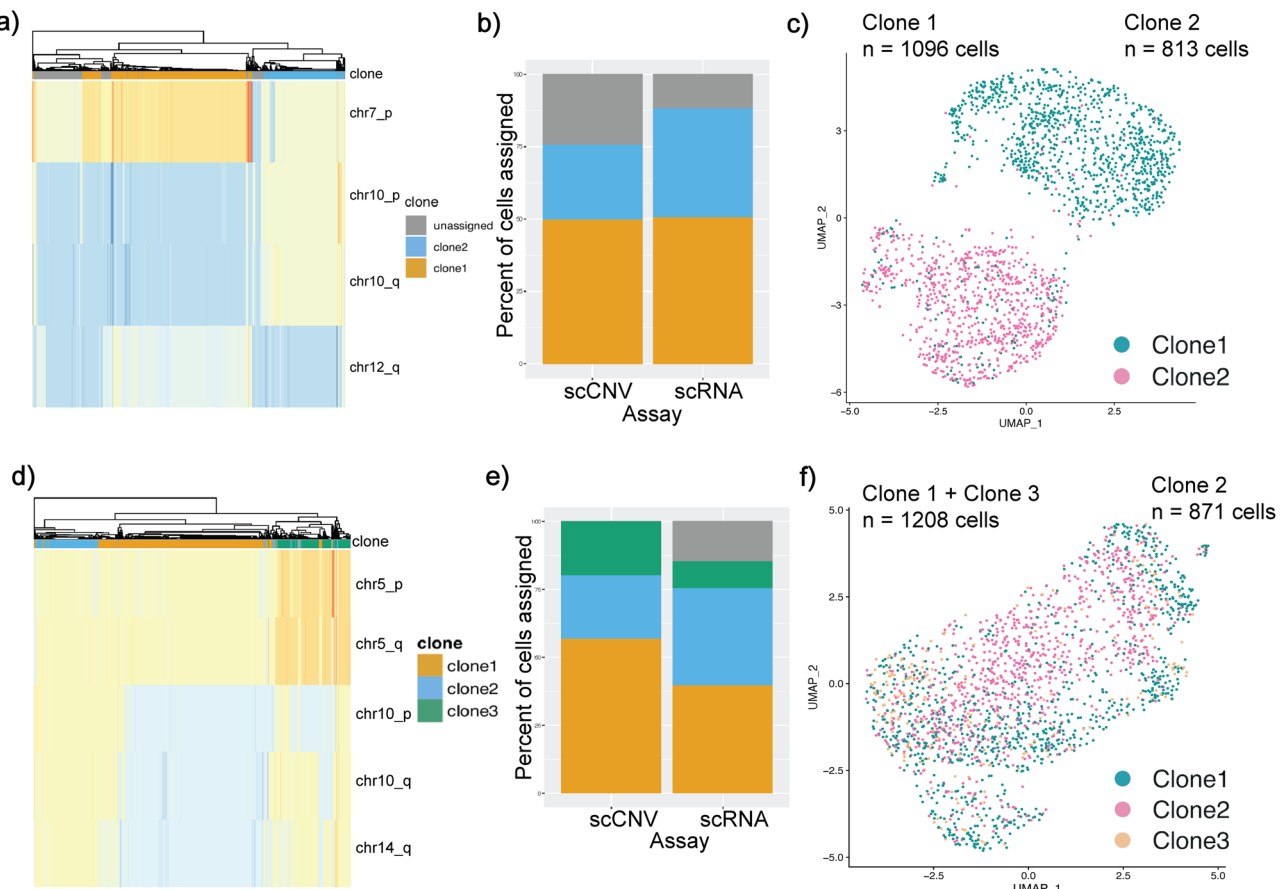

**Fig. 4 Evolution of divergent genomic subclones in MiaPaca2 strains has transcriptomic implications. a** Genomic clones 1 (orange) and 2 (blue) identified from scCNV data of MP2-B based on CNV events at chromosomes 7, 10, and 12. **b** Percentage of cells assigned to clones from scCNV and scRNA datasets. scCNV clones were mapped to corresponding scRNA data using CloneAlign. **c** UMAP data of MP2-B scRNA expression of single cells overlaid with CNV clones assignment for each cell; Clone 1 in green and Clone 2 in pink. **d–f** Same as **a–c** for MP2-C.

almost perfectly to transcriptional subpopulations as visualized in our UMAP analysis (Fig. 4b, c). Next, we used GSEA analysis to search for differences in expression signatures within each of the two delineated subclones. Subclone 1 of MP2-B (1096 cells) is characterized by five hallmark pathways, while MP2-B subclone 2 (813 cells) is defined by six, completely distinct gene sets (Supplementary Table 1). Given the difference in enrichment of cancer-related hallmark pathways that we found between the two MP2-B subclones, we then re-ran GSEA versus MSigDB's Oncogenic signatures. We found that subclone 1 is enriched for oncogenic pathways LEF1 upregulation, P53, and Cyclin D1, whereas subclone 2 is enriched for MTOR upregulation, EIF4E, RAF, and NEF2L2 (Supplementary Table 1).

Interesting significantly upregulated genes in subclone 1 from these oncogenic pathways include KRT13 and LGALS1 (in the LEF1 upregulation pathway; padj=4.39e-49 and 1.35e-21 respectively). Although KRT13 is not ubiquitous in clone 1, the difference in percent of cells that express the gene is significant (32% in clone 1 vs. 3% in clone 2). This gene has previously been shown to predict worse overall survival in pancreatic cancer[32]. LGALS1 overexpression has also been negatively associated with survival in PDAC patients[33], and knockdown of LGALS1 in vitro reduced cell migration and invasion capabilities in pancreatic cancer associated cells[34]. Clone 2 shows significant upregulation of NEAT1 (in the MTOR pathway, padj=1.20e-9). This gene has also been shown to promote PDAC growth and metastasis and predict poor prognosis (Feng 2020)[35]. As both subclones were found at relatively high frequency (Fig. 4e), it would suggest that neither has a clear fitness advantage over the other. Therefore it is not entirely unexpected that we see different genes that promote cancer growth in each subclone.

For MP2-C, we also see correspondence between the genomic profiles (Fig. 4d) and the transcriptional subpopulations (Fig. 4e, f), but the correlation is less obvious than the one-to-one correspondence observed in MP2-B. In MP2-C, a very different pattern of enrichment is observed in both hallmark and oncogenic pathway sets. MP2-C subclones 1 and 3 were merged (due to very similar distribution across RNA transcriptomic clusters) for a total of 1208 cells, the population of which is significantly enriched for two hallmark pathways and four oncogenic pathways (Supplementary Table 1). Surprisingly, MP2-C subclone 2 (871 cells) is not significantly enriched for any hallmark or oncogenic gene sets (Supplementary Table 1). Importantly, there were also zero overlapping gene sets between MP2-B and MP-C, across any subclonal comparison.

In summary, we confirm previous reports that progeny of the apparently identical parental cell line experience substantial genomic and transcriptomic divergence on prolonged ex vivo passaging, that such divergence is observed both within, and across, cultures of the parental cell line, and at least in some instances, the transcriptomic divergence can be ascribed to distinct subclonal genomic alterations.

**Modeling of tissue-based transcriptional PDAC subtypes using scRNA-seq data in established cell lines.** Recently developed transcriptome-based subtyping of PDAC have demonstrated both predictive and prognostic significance in early studies. The dichotomous Moffitt classification schema of "basal-like" and "classical" subtypes was initially established using bulk RNA sequencing strategies applied to human tumor samples (Moffitt 2015)[36]. PDAC transcriptional subtypes are known to associate with specific microenvironmental niches (Raghavan 2021)[8], and primary tumors have been shown to undergo subtype switching in response to microenvironmental cues (Torres 2018)[37]. However, it is not well established whether cell lines can reliably classified into Moffitt subtypes as freshly dissociated tumor tissue or PDO models, nor whether there would be an observable degree of subtype admixture when single-cell methods are applied to subclonal derivatives of parental cell lines. To investigate this, we sought to group the MiaPaca2 transcriptomic subclones into the dichotomous PDAC subtypes of the Moffit classification schema.

We observed that MP2-A and MP2-C have all (9,768 and 10,886 cells respectively) represented by the Moffitt "basal-like" subtype, while MP2-B has subtype admixture, with 3,456 cells sorting as the "basal-like" subtype and 3,556 cells sorting as "classical" (Supplementary Tables 2, 3). Upon further investigation, we found that these samples were being "forced" into one subtype or the other on the basis of a few genes being present or absent, as opposed to having a robust expression of the majority of genes specified by "basal-like" or "classical" subtypes. For example, MP2-A cells were sorted as "basal-like" based on the expression of AREG and KRT15 (Supplementary Fig. 7a), while MP2-C cells were sorted to "basal-like" based only on the expression of AREG, GPR87, KRT15, and LEMD1 (Supplementary Fig. 7c). Similarly, the MP2-B sample, for which we detected genes relevant for both "basal-like" and "classical" subtyping, we found that "classical-like" cells were classified as such based on only LYZ, while "basal-like" cells were classified as such due to expression of AREG, GPR87, KRT15, and KRT7 (Supplementary Fig. 7b).

Given the thousands of cells present and the depth of sequencing on these samples, it is unlikely that this scant gene representation is due to dropout rates or data sparsity, which could result in skewed scRNA data. Rather, our data indicates that PDAC cell lines are sub-optimal models for application of tissue-based classification systems like Moffitt, largely because they might lack expression of substantial numbers of transcripts required for meaningful classification. This finding may impact how prevalent PDAC classification schemes are applied to monolayer culture models and any subtype-based therapeutic outcome predictions derived from these that are then extrapolated to patients.

**Epigenetic alterations define transcriptional divergence observed in monolayer and 3D models.** Ex vivo 3D models of tumor growth are being increasingly adopted, as a number of comparative studies between 3D spheroids and 2D monolayer cultures have shown that 3D culture more faithfully recapitulates in vivo biology[38–40]. Additional work has also shown that 2D monolayers might lack critical features associated with in vivo tumor progression, such as hypoxia[41,42]. Within the context of single-cell analytics, there is scant data to address the extent of transcriptional adaptations that occur in the process of modifying "conventional" monolayer culture to a 3D spheroid model. We find here that upon growth in 3D, Panc1 cells exhibit demonstrable transcriptomic divergence, as well as increased transcriptional diversity, compared to matched 2D culture grown from the same parental vial, upon scRNA-seq analysis. While Panc1-2D and Panc1-3D cultures are overall more similar to each other compared to other cell lines at the transcriptome level (Fig. 1 and Supplementary Fig. 1b), they do not overlap when visualized using UMAP (Fig. 5a, left). To look further into pathway level differences between the 2D and 3D models, we computed differentially expressed genes and processed them with GSEA (Fig. 5b; also see *Methods*). This differential expression was done by subcluster within each of the monolayer and spheroid, so that one can see heterogeneity both within and between each of these. Panc1 spheroids show upregulation of epithelial mesenchymal transition in the largest subcluster relative to the monolayer. Meanwhile, we also see strong differences in pathways within spheroid subclusters, with significant upregulation of the p53

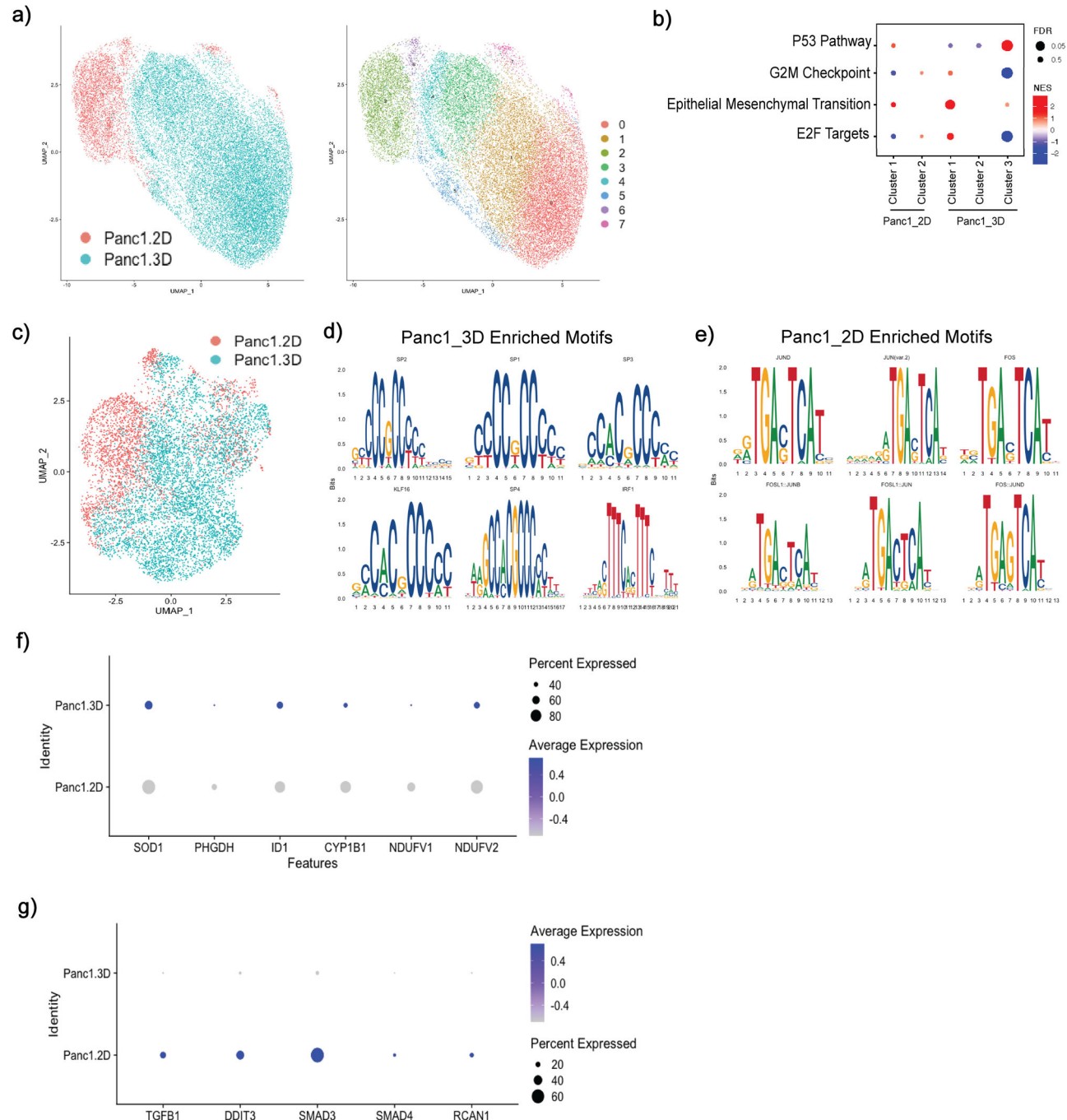

**Fig. 5 Spheroid growth model promotes transcriptional heterogeneity and epigenetic remodeling. a** UMAP overlaying scRNA of Panc1 samples and their unique clusters. **b** Pathway enrichment profiles defined by GSEA analysis for Panc1-2D and Panc1-3D spheroid cell clusters. Size of dot represents FDR, red indicates upregulation, blue indicates downregulation of the pathway. **c** UMAP representing snATAC-seq data of merged Panc1 samples. **d**, **e** Analysis of merged Panc1 snATAC data shows enrichment in Sp/Klf motifs in spheroid data, and enrichment of Fos/Jun motifs in 2D monolayer culture. **f** Comparison of expression of genes associated with Sp/Klf transcription factors in scRNA of Panc1 2D and Panc1 3D. **g** Comparison of expression of genes associated with Fos/Jun transcription factors in scRNA of Panc1 2D and Panc1 3D.

pathway and downregulation of G2M checkpoint and E2F target pathways in one subcluster relative to the other two. This kind of heterogeneity was not seen within the 2D model, with no significant difference in pathways found between the two subclusters.

As with MiaPaca2 samples, we sought to understand the basis of transcriptomic differences observed on scRNA-seq data, but using a slightly different analytic approach. As described earlier,

spheroids were grown from the same parental population of Panc1 cells as the monolayer culture. scCNVseq was conducted on this parental population, and subsequent scRNAseq + snATACseq was conducted after culturing the cells in their respective conditions. We then mapped this parental scCNV profile to paired scRNAseq data after growth under monolayer or spheroid conditions, to understand subsequent transcriptomic reprogramming given identical parental genomic material.

Applying the same clonealign filtering criteria we used in our approach for MP2 cells, we defined two mutually exclusive subclones based on lower copy number (2 vs. 3) in either 15q (subclone 1) or 14q (subclone 2) for Panc1 (Supplementary Fig. 8a, b). When we mapped these clonealign-defined unique genomic subclones to specific transcriptional programs across the two datasets, subclone 1 was comprised of 3,251 cells and subclone 2 of 404 cells in Panc1 2D scRNA-seq data (Supplementary Fig. 8c), while in the Panc1-3D scRNA dataset, subclone 1 was represented by 904 cells, and subclone 2 by 223 cells (Supplementary Fig. 8d). To confirm that these clonealign calls were not an artifact and that the monolayer and spheroid truly share similar scCNV profiles, we ran InferCNV (Tickle 2019) on the scRNA data for each growth condition (Supplementary Fig. 8e). This confirmed the clonealign results, with the majority of (but not all) cells in each condition showing lower copy number in chr15 compared to chr14 and a small subset of cells showing the reverse.

There was little to no overlap in the enriched hallmark pathways (sharing only late estrogen response and cholesterol homeostasis pathways) or oncogenic gene sets (no common pathways) for either of the two subclones between the two growth conditions (Supplementary Table 1). In other words, despite the common parental scCNV profiles, the transcriptomic divergence between the two growth conditions appears to be driven almost entirely by non-genomic events, such as epigenetic alterations.

Given the observed decoupling between Panc1 scCNV and scRNA datasets when comparing spheroids to 2D monolayer, we thus hypothesized that epigenetic modifications may be responsible for remodeling in the spheroid setting. To address this, we performed snATAC-seq and applied the Seurat analysis extension package Signac (Stuart 2020)[43] across a population of 9,785 nuclei analyzed from Panc1 monolayer and 7,453 nuclei analyzed from Panc1 spheroids isolated from the same population, processed in parallel, as the cells used for scRNAseq. As a first measure of epigenetic heterogeneity after conducting a quality control analysis on the sequenced nuclei (Supplementary Fig. 9a, b), we find the nuclei in at least two different chromatin states within each growth condition (Supplementary Fig. 9c, f). Further motif enrichment analysis within each population shows unique subcluster motif profiles (Supplementary Fig. 9d, e, g, h).

We bioinformatically merged Panc1-2D with Panc1-3D nuclei to obtain a differential motif enrichment profile between the samples (Fig. 5c). Notably, we observed enrichment of Sp/Klf-associated motifs in the spheroid model (Fig. 5d), and Fos/Jun motifs enriched in the 2D snATAC-seq data (Fig. 5e). We then confirmed the overexpression of putative target candidate genes affiliated with the enriched motifs that we had identified in corresponding snATAC-seq datasets in the paired scRNAseq data (Fig. 5f, g). In doing so, we find that the regulatory consequences of Sp/Klf motif enrichment are in genes associated with modulation of reactive oxygen species and metabolism such as SOD1[44], ID1[45], and NDUFV2[46]. The low expression, or, in some cases, downregulation, of these genes in the monolayer culture suggest that in the process of epigenetic reprogramming, Panc1 spheroids also undergo a metabolic reprogramming by upregulating oxidative phosphorylation or glycolysis. Conversely, the enrichment of Fos/Jun motifs results in the transcriptional upregulation of classic effectors in this signaling pathway, such as SMAD3, SMAD4, TGFB1, and DDIT3 (Zhang 1998)[47]. The contrasting expression and chromatin profiles between these models could have implications in the results of studies concerning metabolism in PDAC.

In summary, our results suggest that the differences observed in scRNA-seq profiles between spheroid and monolayer growth systems root from a divergence in their underlying epigenetic landscapes, and the resulting differences in transcription factor activity and downstream pathways may have significant implications for the way these models are used interchangeably for in vitro experimentation.

**Long-term maintenance of organoids leads to transcriptomic change.** PDOs have emerged as a promising preclinical model for therapeutic prediction and functional studies across multiple cancer types, including in PDAC[18,19,48–50]. Much like established cell lines, PDOs are maintained and expanded via ex vivo passaging in specially formulated "organoid" media. Early passages of PDOs retain the clonal genomic alterations observed in the parental tumor, as well as the transcriptomic signature(s) predictive of response to cytotoxic agents. However, whether the transcriptome of PDOs evolves over time in culture is less well studied. Therefore, we generated scRNA-seq profiles on a PDAC PDO at two timepoints, "early" and "late", separated by 10 passages.

We found that in the later passage of the PDO, two transcriptionally distinct clusters (Fig. 6a) had emerged, which respectively formed two separate cell fates on trajectory inference (Fig. 6b). Transcripts upregulated in cell fate 1 showed enrichment of epithelial-mesenchymal transition, cell growth, reactive oxygen species (ROS) biosynthesis, and axon development, while transcripts in cell fate 2 were enriched for telomere maintenance, DNA replication, cell cycle, and hypoxia pathways (Fig. 6c). To test whether there were genomic alterations accompanying the transcriptomic divergence, we inferred copy number aberrations from the scRNA-seq data using the inferCNV package[44], which revealed a chromosome 16p gain in the later passage PDO, and an additional chromosome 19q loss specifically in cell fate 2 (Fig. 6d).

Bearing in mind the emerging translational relevance of transcriptional subtyping of PDOs, we paneled for genes relevant to the Moffitt subtyping schema to understand subtype-specific shifts that may occur during ex vivo passaging. The earlier passage PDO has demonstrable expression of "basal-like" transcripts (Fig. 6e) but these did not overlap with the "basal-like" transcripts expressed by the later passage PDO (Fig. 6e). This differs from the expression of transcripts that define the "classical" subtype, which are expressed almost exclusively by the later passage PDO (Fig. 6f).

We confirmed that this mix of classical and basal genes in PD01late is not due solely to admixture within any given cell, and show that some cells classify as basal and others as classical, resulting in a true subtype admixture that suggests gradual transcriptional reprogramming in the PDO over time (Supplementary Fig. 10). We thus observe a subtype admixture over time, which is consistent with findings by Raghavan et al.[8], demonstrating that later passages of PDOs in culture undergo transcriptional reprogramming towards a "classical" phenotype, and reiterating the inherent plasticity of cellular subtypes. Of note, a recent assessment for "basal-like" and "classical" cells in PDAC tissues by subtype-specific protein markers has uncovered bi-phenotypic "hybrid" cells[51], while Ting and colleagues have shown evidence of subtype "drift" in PDAC tissues with various therapies[52], both lines of evidence supporting the plasticity of subtype states. As PDOs become increasingly adopted in translational research, the impact of transcriptomic reprogramming in later passages should be accounted for in the context of any predictive modeling studies.

## Discussion
As a globally utilized backbone of cancer research, established PDAC cell lines have played an indispensable role in enhancing our understanding of disease biology, and therefore, ensuring the

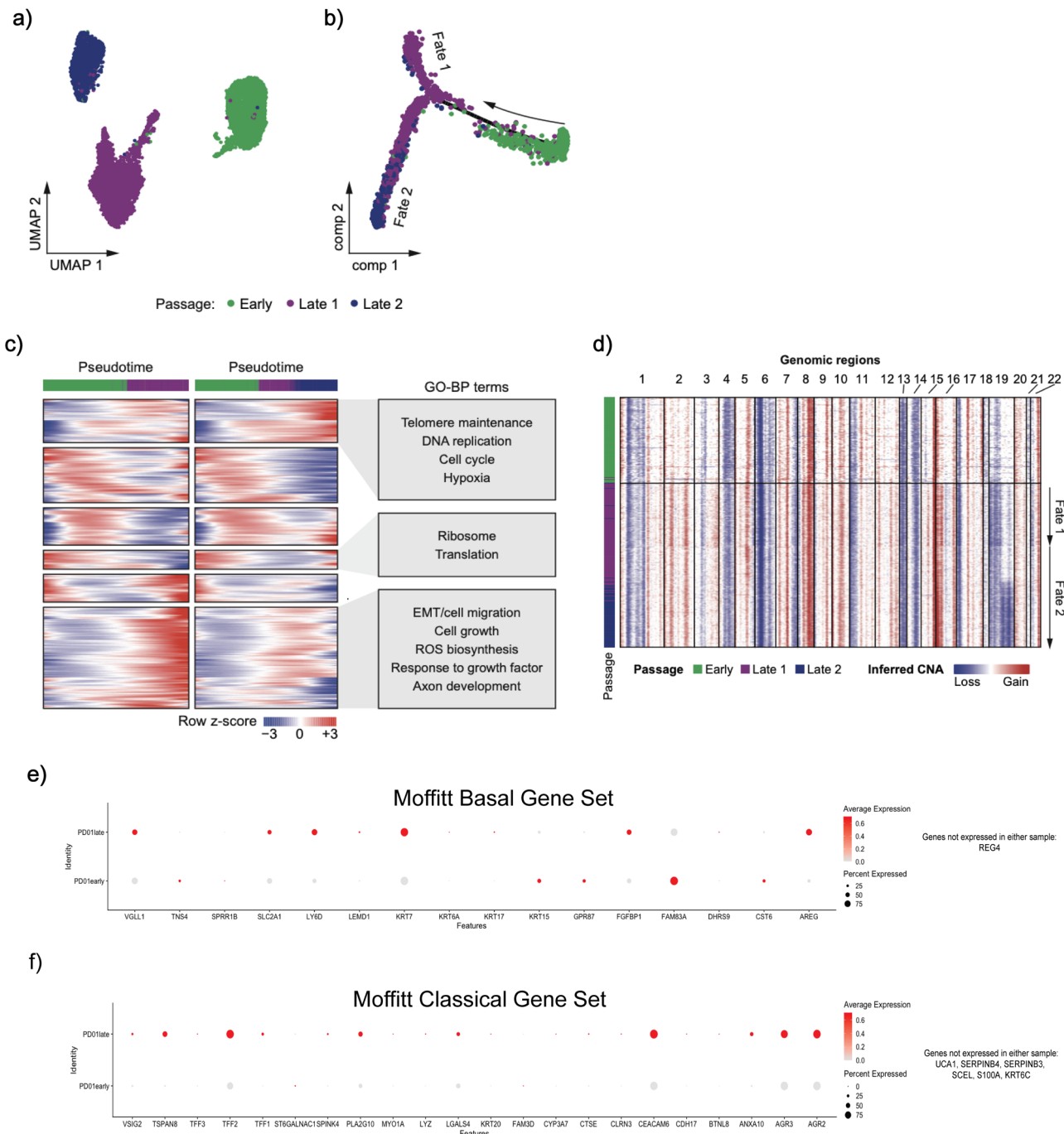

**Fig. 6 Patient-derived organoids evolve with time. a** UMAP plot of cells from a different PDO at early (green) and late (purple, navy) passages. **b** Pseudotime analysis performed using Monocle of single cells from early (green) and late (purple, navy) passage organoids. **c** Branched heatmap (left) showing dynamic gene expression along the pseudotime trajectory for each cell fate in **b**. Pseudotime progresses from left to right. Enriched Gene Ontology biological process terms for each gene cluster are listed on the right. **d** Heatmap showing inferred copy number alterations of early and late passage organoids. Cells (rows) are ordered by pseudotime. Red represents copy number gains and blue represents copy number losses. **e** Comparison of expression of Moffitt Basal subtype genes in scRNA of early (bottom) vs. late (top) PDO1. **f** Comparison of expression of Moffitt Classical subtype genes in scRNA of early (bottom) vs. late (top) PDO1.

accurate characterization of these tools is a prerequisite for ensuring downstream success in translational studies. Using in depth single cell analytics, we evaluate and present a characterization of subclonal heterogeneity in the most commonly used PDAC cell lines. While established PDAC models are assumptively oligoclonal and reflect human tumors on a global level based on published data, the level of subclonal heterogeneity observed on single cell analysis here is unexpected, leading us to

propose the phrase "occult polyclonality" as a molecular paradigm upon which research involving cell lines could be predicated. Our data are comparable to the recent findings of Navin, Michor and colleagues, who identified between seven to 22 subclones on scCNV analysis of triple negative breast cancer tumors and cell lines (Minussi 2021)[53].

Notably, these investigators also reported that cancer cells retain a "reservoir" of subclonal diversity, such that isolated single

cell clones re-diversify their genomes within a relatively short time frame, at a calculated pace of one CNV event every four cell divisions. In line with this observation, we report that three independent strains of the same PDAC parental cell line (Mia-PaCa2) demonstrably deviate from one another on single cell analysis, both in terms of underlying transcriptional profiles and the outgrowth of unique genomic subclones. As another concordant finding between the two studies, we confirm that "major" subclonal genomic events have a significant impact on transcriptomic divergence and clustering, not only within the altered chromosomal loci, but also more broadly in terms of enriched gene sets and pathways. The report by Navin, Michor and colleagues in breast cancer, and our own data in PDAC, harbors the potential to have a profound impact on the rigor and reproducibility of research, both within and across laboratories. For example, it calls into question the assumptions behind isogenic perturbation experiments in cultured cell lines, or the use of archived, cell line-specific profiling data for drawing inferences that are temporally disparate from the data generation point.

We further show that culturing a PDAC cell line typically grown in adherent monolayers as a spheroid model alters the underlying chromatin architecture and induces a more diverse transcriptional repertoire. In particular, we demonstrate acquisition of transcription factor programs in 3D culture that could confound the interrogation of therapeutic dependencies when compared to the identical cell line grown in 2D condition. While we do not posit that one culture condition is necessarily more reflective of in vivo biology than the other, it is noteworthy that certain dependencies (such as metabolic dependencies) initially identified in monolayer conditions have not subsequently been validated within an in vivo context.

As model sophistication increases to more widespread adoption of the PDO format, we also report that, perhaps not unexpectedly, the molecular identities of PDOs tend to evolve over time, recapitulating what is observed with cell lines. In particular, our demonstration of selection against a basal phenotype in a PDO during ex vivo culture contributes to the growing body of evidence that therapeutic prediction studies germane to the parental tumor are best limited to early passages of derivative PDOs. While later passages of PDOs can continue to serve as a bona fide preclinical model for PDAC research, they might need periodic re-interrogation with single cell analytics (or a reliable surrogate assay) to measure divergence over time.

Finally, we report here that the commonly used contemporaneous "controls" for PDAC cell lines, especially HPDE cells, do not meet either genotypic or transcriptional criteria expected for healthy, non-malignant samples. We identify considerable subclonal heterogeneity using single cell analysis (with an "occult polyclonality" that mirrors PDAC cell lines), and "major" chromosomal copy number events in HPDE and HPNE that translate to transcriptional upregulation of *AURKA* and *BRCA1*, respectively. Future studies may warrant a thorough investigation of whether omission of these chromosomal events in the context of comparative functional assays led to skewed results. Better yet, the systematic development, characterization, and dissemination of more reliable in vitro cell line "normal" controls for PDAC studies may be a requisite for maintaining the rigor of preclinical findings in the field.

## Materials and methods

**Cell line selection and 2D culture**. For non-neoplastic cell lines, HPDE and hTERT-HPNE were used, given their prevalence as "normal controls" in literature (over 800 citations for each line in PubMed). hTERT-HPNE (ATCC Cat. CRL-4023, passage no. 1) was plated in Complete Growth Medium, consisting of 75% DMEM without glucose (Sigma D-5030) with an additional 2mM L-glutamine and 1.5 g/L sodium bicarbonate, 25% Medium M3 Base (Incell Corp. M300F-500), 5% fetal bovine serum (ThermoFisher Scientific Cat. 10082147), 10 ng/mL human

recombinant EGF, 5.5 mM D-glucose (1 g/L) and 750 ng/mL puromycin. HPDE (Kerafast Cat. ECA001-FP, passage no. 4) cells were cultured using Keratinocyte Basal Medium with supplied supplements (Lonza, Clonetics KBM, Cat. CC-3111). For PDAC-derived cell lines, we wanted to ensure that lines selected were both widely-used in the field, and be representative of a range of characteristics in terms of differentiation and epithelial versus mesenchymal origin. MP2-A cell line was purchased from the American Type Culture Collection (ATCC Cat. CRL-1420). MP2-B was obtained from a collaborating lab and plated upon arrival (passage no. 2). MP2-C was thawed and plated from an in-house stock vial (passage no. 3). All MiaPaCa-2 cells were cultured using DMEM (ATCC Cat. 30-2002) supplemented with 10% fetal bovine serum for up to 2 expansions. BXPC3 (ATCC Cat. CRL-1687, passage no. 1) and HPAF-II (ATCC Cat. CRL-1997, passage no. 1) cell lines are commercially available, and cultured using RPMI 1640 (Gibco) supplemented with 2 mM glutamine, 10% fetal bovine serum (ThermoFisher Scientific Cat. 10082147). Panc-1 cell line (ATCC Cat. CRL-1469, passage no. 3) for 2D studies was cultured in DMEM with 2 mM glutamine and 10% fetal bovine serum (ThermoFisher Scientific Cat. 10082147). Cell lines were harvested for single-cell dissociation and sequencing at 80-90% confluency. A volume of 3 mL of media from each cell line was harvested and sent to the MD Anderson core facilities for mycoplasma testing. All cell lines were washed with PBS and incubated with 0.25% Trypsin for 3–5 min. For all lines except HPDE, trypsin was neutralized using each respective cell line's media; in HPDE, trypsin was neutralized using a Trypsin Inhibitor (Invitrogen Cat. 17075-029). Cell lines were centrifuged for 5 minutes at $1250 \times g$, resuspended in cold PBS + 0.04% BSA (Roche Ref. 03116956001), filtered using Flowmi 40 µM pipette filter tips (Sigma BAH136800040), and counted to ensure debris-free single cell suspensions. A portion of cells were sent to the MD Anderson core facility for molecular fingerprinting, and all cell lines were counted using a Countess II automated cell counter. 2,000 cells were loaded per lane into the 10× Chromium Platform for scCNV sequencing, and 10,000 cells per lane were loaded for scRNA sequencing.

**Panc-1 3D spheroid culture**. For 3D spheroid growth, Panc1 cells from the same passage as the cultured 2D cells were thawed, washed with PBS, suspended in growth-factor reduced Matrigel (Corning #356231), and plated in 80 µL spheres on 24-well Nunclon Delta surface plates (TMO140620). Spheroids were incubated for 10 min at 37 ℃ to allow the matrigel to harden, and 650 µL of DMEM + 2 mM glutamine and 10% fetal bovine serum were added. For spheroid dissociation in single-cells, media was removed, and spheroids were manually dissociated in 500 µL of TryplE Express (Gibco Cat. 12604013). DMEM media was then added, and spheroids were centrifuged at $1500 \times g$ for 5 min. To dilute the matrigel and obtain single-cells, cells were resuspended in 5 mL PBS, filtered twice with 70uM cell strainers (Corning CLS 431751), resuspended once more in cold PBS + 0.04% BSA, filtered again using Flowmi 40 µM pipette filter tips, counted, and loaded at 10,000 cells per lane into the 10× Chromium Platform for scRNA sequencing.

**Sample acquisition for organoid generation**. A patient was recruited at the University of Texas MD Anderson Cancer Center through informed written consent following institutional review board (IRB) approval (Human Research Protection Program Lab00-396 and PA15-0014). The study was conducted in accordance with Good Clinical Practices concerning medical research in humans per the Declaration of Helsinki. An EUS-FNA biopsy procedure was conducted and confirmation of pancreatic ductal adenocarcinoma present in biopsy-obtained tissue specimens was provided by a pathologist. The tissue specimens were subsequently delivered to the research laboratory setting, where they were processed for organoid generation.

**Organoid generation**. Dissociated cells were resuspended in PaTOM media[54] and centrifuged at $1500 \times g$ for 5 min. Cell pellet was gently resuspended in Matrigel Growth Factor Reduced (Corning) and plated as a dome in a Nunclon Delta Surface 24-well plate (Thermo Fisher). Cells embedded in matrigel were incubated at 37 C for 15 min and covered with PaTOM media supplemented with Y-27632 dihydrochloride (Tocris). Organoids were expanded and grown up to passage 4 (PDO early), at which point there were a sufficient number of cells in culture to be processed for single cell sequencing while also maintaining a stock in culture for further growth. After an additional 10 passages, PDO cells were harvested for single cell sequencing to be compared as "PDO late" in this study.

**Library construction for whole exome sequencing (WES) of HPDE and HPNE cell lines**. DNA was prepared as described previously[55], including fragmentation with a Covaris LE 220 ultrasonicators system (Covaris) and library construction with the SureSelect XT HT–targeted enrichment protocol (Version A1, July 2017). Libraries were subsequently hybridized to a whole-exome capture library (SureSelect AllExonV7; Agilent, C/N: 5991-9039) and sequenced on a NextSeq500 (Illumina) using 300-cycle kits, paired-end.

**Whole exome sequencing and segmentation analysis**. Bcl2Fastq (2.20.0, Illumina) was used to convert raw Illumina data (bcls) into fastqs. Fastqs were then trimmed with SureCallTrimmer (v4.0.1, AGeNT v2.0, Agilent), aligned to hg19 with BWA (Burrows-Wheeler Aligner, 0.7.15-r1140, Li 2010)[56], and barcodes were

collapsed with LocatIt (v4.0.1, AGeNT v2.0, Agilent) to family size of 1. Resulting bam files were then processed by GATK (v4.1.2.0) according to best practices (Van der Aurwera 2013)[57], including base quality recalibration (BQSR). Segmentation analysis was performed as described by GATK best practices[56]. Briefly, we use the PBMC data, sequenced with the same platform and library to create a CNV panel of normals (PoN) to establish a baseline. We then collected read counts for HPDE/HPNE on padded (100 bp) targeted regions, and then standardized and denoised using the PoN. Read counts were then standardized and denoised before making segmentation calls.

**Single-cell RNA and CNV library construction and sequencing**. All cell lines were processed using 10× Genomics platforms: Single Cell 3′ Reagent Kits v2 for scRNA (Demonstrated Protocol CG00052, Rev F) and the Chromium Single Cell DNA Reagent Kit for scCNV (Demonstrated Protocol CG000153, Rev C). All libraries were assessed for quality on Agilent TapeStation 2200 and quantified using both Agilent TapeStation 2200 reagents and Qubit dsDNA HS Reagent kits (Invitrogen Cat. Q33231) before loading onto NextSeq500 (Illumina). Multiplexed scRNA libraries were sequenced using 150-cycle kits, paired-end, Read 1 26 cycles, Read 2 98 cycles, i7 index 8 cycles, i5 index 0 cycles. Multiplexed scCNV libraries were sequenced using 300-cycle kits, paired-end, Read 1 150 cycles, Read 2 150 cycles, i7 index 8 cycles, i5 index 0 cycles. Cell Ranger version 2.0 was used to convert Illumina base calls (bcls) to FASTQ files. FASTQ files for scCNV data and scRNA were aligned to the reference genome GrCh38 using files provided by 10x Genomics. Cell Ranger DNA Software v1.1 was used for subsequent downstream analysis of each cell line's aligned data.

**Single-nuclei ATAC library construction and sequencing**. Of the Panc1 monolayer and spheroid cultures, a subset of whole cells was isolated from the pools processed for scRNA + CNV sequencing. Using the nuclei isolation protocol and methods proscribed by 10× Genomics (Demonstrated Protocol CG000169, Rev D), we were able to load 5000 nuclei per well for Panc1-3D and 7,000 nuclei per well for Panc1-2D for processing using the 10× Genomics Single Cell Atac Kit, v1. As with the previously described scRNA and scCNV libraries, all quantification and quality control was conducted on an Agilent 2200 TapeStation using D1000 High Sensitivity reagents, with additional quantification using Qubit dsDNA HS Reagent kits. A pooled library containing both Panc1-2D and Panc1-3D snATAC libraries was loaded on the NextSeq500 using 150-cycle kits, paired-end, dual-index mode, Read 1 50 cycles, Read 2 50 cycles, i5 index 16 cycles, i7 index 8 cycles. Cell Ranger version 3.1.0 and cellranger-atac version 1.2.0 were used to convert Illumina bcls to fastq files and align samples to reference genome GrCh38.

**scCNV clonal analysis**. A literature search was conducted to identify commonly amplified/deleted genes in PDAC, and the output of that search was combined with a Maitra Lab in-house panel of commonly mutated genes in pancreas cancers. Using Ensembl v84 for gene locations and bedtools to intersect coordinates, chromosomal locations and copy number values as provided by 10× Genomics software in node_cnv_calls.bed were mapped to gene-specific loci. Cells listed as noisy or with a haploid genome (mean ploidy < 1.5) were excluded, and then a random 1000 cells per cell line were chosen using the "sample" function in R for visualization.

**scRNA Seurat analysis of cell lines**. Seurat (v3.1)[58] merged analysis was used to profile the scRNA cell line data. For removal of mitochondrial genes in cell lines, we subsetted all cells with less than 20% mitochondrial gene expression. Log normalization, variable feature identification (FindVariableFeatures), z-scoring (ScaleData) were applied to the merged object of all cell lines, and principal component analysis (RunPCA, npcs = 30) and subsequent dimension reduction (UMAP) were applied. Seurat's FeaturePlot command was used to visualize and scale expression of known mesenchymal (Vimentin), epithelial (EpCAM/KRT8), or neither phenotypes based on surface expression marker gene sets (Castillo 2018)[59].

For confirmation of AURKA and BRCA expression in HPDE and HPNE cell lines, scRNAseq data for these samples were combined into a single seurat object, were log-normalized (NormalizeData) and scaled (ScaleData), and seurat's FindMarkers command was used to measure expression of each gene.

**Gene set enrichment analysis (GSEA) for MiaPaca2 clusters**. To obtain significantly enriched pathways on a per-cluster basis from MP2 cell lines, marker genes were extracted using FindMarkers function in Seurat, on a per-sample basis (ex: FindMarkers, ident.1 = "MP2.A", ident.2 = c("MP2.C", "MP2.B")). Resulting genes were ranked by -log10(FDR) multiplied by fold-change directions, and the output file was formatted according to requirements for downstream input to GSEA. This yielded an output wherein upregulated genes had positive values and downregulated genes had negative values. Pre-ranked GSEA was thus performed using fgsea R package v1.12.0[60] specifying 10,000 permutations, and fgsea output was filtered based on FDR and adjusted p-values for each pathway.

**Copy number inference of MiaPaca2 cell lines from scRNA-seq**. Copy number alterations from scRNA-seq were inferred using inferCNV R package v1.0.4 (https://github.com/broadinstitute/inferCNV). Raw gene counts from filtered cells were used, and cutoff was set to 0.1. HPNE cells were used as reference.

**Gene set enrichment analysis (GSEA) for Panc1 sample-specific clusters**. To obtain significantly enriched pathways for clusters specific to either Panc1-2D or Panc1-3D, marker genes were extracted on a per-cluster basis using FindMarkers function in Seurat from a merged Panc1-2D + Panc1-3D object (ex: panc3_c2.-markers < - FindMarkers(object = panc1, ident.1 = 2, ident.2 = c(0,1,3,4,5,6))). Resulting genes were ranked by -log10(FDR) multiplied by fold-change directions. This yielded an output wherein upregulated genes had positive values and downregulated genes had negative values. Pre-ranked GSEA was thus performed using fgsea R package v1.12.0 specifying 10,000 permutations, and fgsea output was filtered based on FDR and adjusted p-values for each pathway, as previously described for MiaPaca2 GSEA analysis.

**snATAC analysis of Panc1 for motif activity, coverage plotting, and integration with matched scRNA**. Using Seurat v3 and its extension package Signac version 1.0.0 (https://github.com/timoast/signac), along with additional R packages for gene annotation (EnsDb.Hsapiens.v86) and the JASPAR 2020 Motif Database (JASPAR2020), 10× genomic output files (metadata, fragments file, fragments index, filtered peak matrices) were used to generate Seurat objects containing motif information, gene annotations, and genomic ranges. Quality control on nuclei was done by quantifying nucleosome signal, TSS enrichment, number of peaks in fragmentation regions, and nucleosome signal according to Signac software recommendations (TSS Enrichment > 2, nucleosome signal < f10, blacklist ratio < 0.05, percent reads in peaks > 15, peak region fragments between 1,000 and 20,000). Samples were normalized using term frequency-inverse document frequency (TF-IDF) normalization, top features were identified, and dimensional reduction using singular value decomposition (SVD) on the TF-IDF matrix using peaks as features was done on both individual samples and a merged analysis object that had been created using fragmentation and peak data to identify common features to use as merging anchors. Motif activities were subsequently calculated using chromVar, as adapted for Signac. We then identified motifs that were enriched in each subcluster per sample, as well as differentially enriched between the two samples (merged analysis). To correlate regions of chromatin activity with scRNA data, per-sample snATAC-Seurat objects containing gene activity data were scaled, normalized, processed with latent semantic indexing (LSI), and UMAP dimensional reduction. We use all peaks that have at least 100 reads across all cells, and reduced dimensionality to 50, as recommended by Seurat guidelines. The processed snATAC dataset was then merged with the previously-described processing of scRNA data for Panc1 samples, with merging based on common anchors identified between the datasets, using scRNA data as a reference (FindTransferAnchors), to ensure commonalities and cluster structure between the datasets. Next, gene lists were curated using the GeneCards Human Gene Database- genes specifically activated by each enriched motif were queried and plotted using scRNA data (DotPlot).

**Cladogram analysis of cell lines**. We sought to quantitatively assess the transcriptional similarity among the cell lines and establish if the families of origin for the cell lines are preserved. We use graph based methods called diffusion maps to compute the distances between the cell lines. In particular, we first map the cells from high dimensional transcriptomic space to a low dimensional rescaled diffusion space (Setty 2019)[61], where phenotype distance between cells are more faithfully represented and compute distances between the cells in this new space.

Briefly, diffusion maps[62] is a family of techniques to reduce the dimensionality of any high-dimensional data and have gained a lot of application in the analysis of single-cell data[63–65]. Essentially, they transform data from high dimensional gene expression space to a space of much lower dimensions, where usual Euclidean distance is reflective of the cellular phenotypic distance. Computing distances in diffusion space requires choosing the optimal number of diffusion coordinates. However, it has been proposed that rescaling the diffusion coordinates by their eigenvalues can circumvent this issue[61].

For this, we combined all the samples and the resulting data was then reduced in dimensionality first using PCA. Diffusion Maps was then computed on the resulting PCA for which we first constructed a k-nearest neighbor (k = 30) graph using Euclidean distance and converted the resulting distance matrix into an affinity matrix using an adaptive Gaussian kernel (Setty 2019, van Dijk 2018). The resulting affinity matrix was row normalized (each row sum to 1) to obtain a Markov matrix, whose eigenvectors represent the diffusion coordinates. These coordinates were rescaled using the associated eigenvalues as described in ref.[61]. Finally, we computed the Euclidean distance between every pair of cells between any two samples in the rescaled diffusion space. Thus, we defined the phenotypic distance between the two samples as the average distance between all pairs of cells as computed above. The resulting distance between samples was represented as a cladogram using the dendrogram function on linkage function in the scipy.cluster.hierarchy package in Python with parameter method = "average". This was also represented as a heatmap using the clustermap function in seaborn package in Python.

**Phenotype map of existing gene sets**. We sought to understand the intra-sample heterogeneity of MiaPaCa2 samples by doing subtype analysis. We utilized gene sets from Moffitt et al.[36] and used the expression of the genes to assign every cell to one of the subtypes. We constructed a simple assignment scheme to achieve this. We began by computing the average subtype expression for each single cell, i.e. average over the genes that define the subtype. We then reasoned that a single cell can be labeled as belonging to the subtype with the highest average expression.

**scCNV processing for scRNA clonal expression correlates**. CNV calls per segment, per cell (from "node_cnv_calls.bed") were intersected with mappable regions (as defined in "mappable_regions.bed") using bedtools (v2.29.2). Next, the copy number for each mappable region for each cell was defined as the integer copy number with the most total bases covering the region. Finally, only regions of at least 2 Mb on autosomes (chr1-22) were retained for downstream analysis. Cells defined as noisy in the QC summary file ("per_cell_summary_metrics.csv") were removed prior to downstream analysis as well.

This data was converted to a cell × region matrix, with all integer values. Next, for each chromosome and chromosome arm, mean copy number was calculated by weighting based on the total length at each integer copy number, across all mappable regions. Ploidy was defined based on the median of these values across all whole chromosomes. Cells with ploidy more than one standard deviation from the mean across cells were removed.

Next, visual assessment of the chromosome arm mean copy number values per cell was used to determine which regions warranted further examination. Per sample, the chromosome arms that showed variation in a large percentage of cells were: MP2-B – chr7p, chr10p, chr10q, chr12q; MP2-C - chr5p, chr5q, chr10p, chr10q, chr14q; Panc1-2D – chr14q, chr15q. Next, all mappable regions in these chromosome arms were plotted for further visual assessment. In Panc1-2D, chr15q mappable regions starting at coordinates <40 Mb or >80 Mb, which showed a different profile from the other mappable regions, were also excluded. In MP2-B, only mappable regions from 93.5 to 126.3 Mb on chr12q were considered, as the remaining regions had consistent copy number across cells. Next, clones were defined as described in the main text. The end result of the scCNV workflow for each sample was a segment x clone matrix, where the segments were the mappable region coordinates and the values were integer copy number values.

**scCNV-scRNA integration**. The mappable region coordinates were intersected with the gene coordinates (from Ensembl v84, same as in the CellRanger reference) to produce a gene x clone matrix. This matrix was used as input to clonealign (v2.0), which was used to assign cells in the RNA to each clone. Default parameters (including min total nUMI per gene of 20 and min total nUMI per cell for all copy-variant genes of 100) were used to filter genes and cells before running clonealign. Next, clonealign was also run with default parameters, except for the minimum probability required to call a cell as assigned to each clone. This parameter was set to 90% for MP2-B and Panc1-2D. For MP2-C this was lowered to 50%, due to very low numbers of cells with high confidence clone calls found in this sample.

Seurat (v3.1) was used for dimension reduction (UMAP). For this analysis, all annotated genes expressed in at least 10 cells were included. Next, only cells assigned to a clone with confidence above the threshold were included for downstream analysis. Finally, standard analysis was performed including SCTransform for normalization and variable gene selection, followed by principal component calculation on the variable genes (RunPCA). The first 30 PCs were used for input into UMAP (RunUMAP) for the MiaPaca2 samples, while the first 50 were used for Panc1.

Seurat's FindMarkers command was used for differential expression between clones. For MP2-C, clones 1 and 3 were combined into one group to create a pairwise comparison versus clone2. This was because cells called as either of these clones showed very similar expression. All default parameters were used except for reducing the log-fold-change threshold to 0.10 from 0.25.

**GSEA of clonealign-defined scRNA clones**. Pre-ranked GSEA was thus performed via the GSEA software version GSEA_4.0.3. Molecular Signature Databases h.all.v7.1.symbols_1.gmt (hallmark pathways) and c6.all.v7.2.symbols.gmt (oncogenic signature gene sets) were both used to align gene lists extracted from clonealign-defined scRNA clones in Seurat (see Fig. 2g, j, Supp. Fig. 6c, d), as described above using FindMarkers. A false discovery rate (FDR) cutoff of 25% was used to analyze the GSEA results for significant pathway enrichment per sample. Manual curation of genes lists from significantly-enriched per-clone pathways was conducted to identify individual genes relevant to pancreatic cancer biology (i.e. KRT13, LGALS1).

**Dimensionality reduction and cell clustering of PDO pair**. Initial single-cell analysis was performed using Seurat v3.1.0. Cells containing less than 200 genes were removed, and log-normalized expression of 2,000 variable genes were used to reduce the data into two-dimensional space. Cells from PDO1 early and PDO1 late samples were combined using FindIntegrationAnchors and IntegrateData functions. To retain the highest number of cells for subsequent analyses, no additional filters were initially applied. Cells with higher expression of mitochondrially encoded genes were filtered out *post hoc* based on the results of the clustering.

**Cell state trajectory analysis of PDO pair**. Trajectory inference was performed using Monocle v2.13.0[66]. Analysis was done using raw counts obtained from Seurat objects with preserved cellular metadata. Top 1,000 variable genes were used to reduce the data and order the cells. Pseudotime heatmap was created using genes with dynamic expression changes obtained using differentialGeneTest function specifying fullModelFormulaStr = "~sm.ns(Pseudotime)" and q-value cutoff of $1 \times 10^{-40}$. Branched pseudotime heatmap was created using BEAM function and q-value cutoff of $1 \times 10^{-5}$. Genes were clustered based on pattern of expression changes and extracted from Monocle object for subsequent analysis.

**Reporting summary**. Further information on research design is available in the Nature Research Reporting Summary linked to this article.

## Data availability

The scRNAseq and scATACseq datasets generated in this study have been deposited in the Gene Expression Omnibus repository under accession code GSE173339. These datasets are publicly available. All data pertaining to DNA sequencing assays (both single cell CNV and Whole Exome Sequencing) generated and analyzed in this study are available in the European Nucleotide Archive, Project: PRJEB44785.

## Code availability

For reproducibility, we have posted code for scCNV, scRNA, and snATAC analyses on Github at https://github.com/heathergeiger/PDAC-cell-line-heterogeneity.

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

## Acknowledgements

We thank L. Nakhleh and M. Edrisi for lending their bioinformatic expertise, and S. Muthuswamy for providing organoid growth media reagents. We thank our funding sources wherein A.M. is supported by the MD Anderson Pancreatic Cancer Moon Shot Program, the Khalifa Bin Zayed Al-Nahyan Foundation, and the National Institutes of Health (NIH U01CA196403, U01CA200468, P50CA221707). J.J.L. is supported by the NIH (T32CA009599). A.S. is supported by the German Research Foundation (SE- 2616/2–1). V.B. is supported by the NIH (U54CA096300, U54CA096297, and T32CA217789).

## Author contributions

M.E.M contributed to study design, experimental setup and execution, data collection, sequencing, data analysis, methods development, manuscript writing, and manuscript editing. H.G. contributed to study design, methods development, data curation, data analysis, figure design, manuscript writing, editing, and review. J.J.L. contributed to data analysis, figure design, and manuscript writing. R.S. contributed to data analysis, methods development, and manuscript writing and editing. A.S. and V.B. contributed to study design, single-cell library construction, whole exome sequencing library construction and sequencing, and manuscript editing. J.W., F.W, and S.L. contributed to copy number variant analysis, whole exome sequencing analysis, data interpretation, and manuscript editing. D.B.S. contributed to whole exome sequencing library generation and sequencing. B.M.S. contributed to ddPCR library construction and analysis, methods development, and laboratory tech support. M.H.G.K. contributed to surgical acquisition of clinical tissue for PDO generation. K.C. contributed to data analysis, methods development, data interpretation, and manuscript editing. P.A.G. contributed to study design and execution, laboratory oversight, FISH data acquisition and analysis, and manuscript editing. N.R. and A.M. jointly supervised this work as principal investigators, providing laboratory funds, personnel, infrastructure support, study inception, and oversight to publication.

## Competing interests

A.M. receives royalties for a pancreatic cancer biomarker test from Cosmos Wisdom Biotechnology, and this financial relationship is managed and monitored by the

UTMDACC Conflict of Interest Committee. A.M. is also listed as an inventor on a patent that has been licensed by Johns Hopkins University to Thrive Earlier Detection (Patent No. US20190256920A1, "Differential Identification of Pancreatic Cysts"), and is a consultant for Freenome and Tezcat Biotech. The remaining authors do not have competing interests to disclose.
