## [Peer review file · Nature Communications]

REVIEWER COMMENTS

Reviewer #1 (Remarks to the Author): Expert in pancreatic cancer genomics and transcriptomics

This manuscript by Monberg et al. uses cutting-edge single-cell technologies to investigate genetic and transcriptomic heterogeneity within cell lines commonly used for PDAC research – including both cancer-derived lines, and immortalized lines used as controls. This work is potentially of high value and impact to the field because it revisits basic assumptions about the biology of commonly used cell lines and culturing techniques in light of new profiling methods that can reveal previously hidden heterogeneity. The authors report previously underappreciated genetic and transcriptomic heterogeneity both within individual cultures of these cell lines, and between independent cultures of the same lines. Based on these findings, the authors express concern about the widespread use of these cell lines in the field, and urge careful consideration about whether/how to include these lines in future preclinical studies.

Despite the high potential value to field, this study has several fundamental flaws as currently constructed that, in my opinion, should preclude publication in current form. Chief among these are the lack of technical replicates to account for variability in the profiling technologies themselves, and major deficiencies in the bioinformatic analyses of the single-cell data. I describe these concerns in more detail below, and urge that major steps are taken by the authors to address these issues. The analysis of single-cell data is a new and rapidly-evolving in the field, and I think we as a field need to be very careful that observations made with single-cell technologies are very rigorously analyzed and validated given the high complexity and inherent noisiness of this data type.

Major concerns

1. The authors perform single-cell profiling (mostly scCNV and scRNA) on separate cultures of the same cell lines (e.g. MiaPaca2 lines from three sources) and, observing differences in the single-cell data from these different sources, they conclude that the underlying cultures are different. However, the authors do not seem to account for the alternative explanation that some of the differences they observe are due to technical variability between runs of the single-cell profiling methods themselves. The single-cell data is not a perfect representation of the underlying samples. It has inherent technical biases and artifacts that can vary between runs, and can also suffer from sampling bias as <10K cells are generally profiled in a single run. In the context of this study, where single-cell data is being used to draw conclusions about differences in the underlying biosamples, I think it is essential to perform technical replicates where cells from the same culture are split into two aliquots and assayed independently. I am unable to draw conclusions about the differences between three MiaPaca2 cell sources, for example, based on a single replicate from each. Given the emphasis of these authors on reproducibility and rigor, this seems to be a particularly important flaw.

2. Several of the transcriptome comparisons use GO term enrichment to point to potential biological differences between subclusters. However, it seems to me that in several instances the authors are using an FDR cutoff of 0.5 instead of the standard 0.05, and are thus basing conclusion on results that are not statistically significant. In all cases where this GO data is presented or discussed I think the authors should make explicitly clear which results exceed the threshold for statistical significance and refrain from overinterpreting results that do not meet this threshold (e.g. Figure 2C, 3B, and the results section “Epigenetic alternations define ...” where results in the supplemental excel file are discussed).

3. Aside from the GO analyses discussed above, I am concerned with the lack of other analyses into the biological differences between clusters, as well as the lack of independent experimental validation of these differences. Clustering analysis of single-cell data rely on arbitrary “resolution” parameters that can dramatically alter the results. Thus, in my opinion it is critical not to rely on clustering analyses to conclude that two groups of cells are biologically different, but rather, to use these analyses to help point to biological features (genes, pathways, etc) that may differ between groups of cells, and then validate these. I suggest that the authors perform additional analyses into the potential biological differences between cell clusters to include at a minimum differential gene expression analyses (not just GO enrichment) and cell cycle analyses (now readily inferred from single-cell RNA-seq data – see PMID: 32312741 for example; differences in cell cycle stage could easily confound these culture-to-culture comparisons). Moreover, findings should be validated with independent experiments. For example, differentially expressed genes between two cell culture sources can be validated by FISH at the single-cell level or at least rt-qPCR at the bulk level.

4. With regard to the scATAC-seq analysis in figure 3, the number of cells profiled seems very low and probably too small to draw conclusions from clustering analysis like those shown in supplemental Figure 6 where all of the cells essentially are essentially in one large “blob.” I was also unable to find quality control assessment for any of these libraries (or scRNA-seq libraries) including read counts per cell and signal/noise metrics like Transcription Start Site (TSS) enrichment. In my opinion, these QC analysis should be included in supplement if not already there somewhere.

5. A number of the figure panels are very hard to interpret due to illegibly small or blurry text, clutter, and/or lack of labeling (e.g. 1C,D,G,H; 2B,D,G). This detracts from the manuscript because I am unable to independently assess these data as presented, and am left to rely only on the authors description/interpretation in text. I suggest a major overhaul of these panels in any resubmission.

Reviewer #2 (Remarks to the Author): Expert in pancreatic cancer genomics and organoids

In this study, Monberg et al report that there exist an occult polyclonality in preclinical pancreatic cancer models, which drives in vitro evolution. In a nutshell this polyclonality reflects as heterogeneity in transcriptomic profile – even within the same cancer cell line cultured in 2D vs 3D or over an early or late passage period. The authors utilized state-of-the-art technologies in genomic/transcriptomic profiling (e.g., scRNA, snATAC seq) and covered a spectrum of in vitro models from the traditional/routine methods (i.e., 2D) up to organoids, which they rightly note has become a well appreciated preclinical model for studying human cancer.

The study obviously involved a considerable amount of effort that has the potential to impact the choice and use of cell lines for future pancreatic research. The authors also did a nice job in describing their Methods, especially on cell culture conditions (very important in the context of my comments below). However, I am bit concerned about whether the overall message conveys any substantial novelty given that cancer cell line heterogeneity is already a well-known issue. In addition, the rationale for choosing the cell lines (e.g., Panc-1, MiaPaCa2, HPAF-II and BxPC-3) are unconvincing, and the potential aspects of clinical translation are not particularly clear. These issues should be addressed to improve the impact of this work.

Major comments

1. There are numerous studies detailing the heterogeneity of pancreatic cancer cell lines, including differences in marker expression, nutrient utilization (e.g., glycolysis vs OXPHOS), and growth properties. Deer et al (cited as ref. #11 in the manuscript) reviewed many aspects of divergence among pancreatic cancer cells. This reviewer therefore believes that the manuscript in the current state has not demonstrated novelty despite the technologies applied. For example, could the authors relate any of the observed transcriptomic differences to phenotypic differences – e.g., do the three ‘strains’ of MiaPaca2 cells differ in growth rate, nutrient consumption, etc, and does that correlate with their respective gene changes? This will help readers to better contextualize the potential translational relevance of this study.

2. It seems the rationale for selecting the cell lines was just based on the fact that they are commonly used in research. This does not necessarily add value. These cell lines even differ in their growth media conditions (as rightly noted by the authors in ‘Methods’); that alone can drive differences in transcriptomics. The cell lines should thus be presented based on their distinct phenotypes. One good parameter could be basal differences in their growth rate or migration (fast vs slow, etc), which can be related to cancer progression. Then, determine whether cells that more closely mimic pancreatic cancer features undergo similar degree of heterogeneity upon culture or are more robust. This level of detail should be provided. With such information, the potential value of using these cell lines will become more or less clear especially when complemented with the novel technologies used in this study.

3. Based on the above point on differences in culture condition, the authors may consider focusing more on cells that are grown under the same culture conditions, e.g., the MiaPaca2 'strains' – transcriptomic and phenotypic differences, overlap with pancreatic cancer subtypes, etc. This 'strains' comparison is most likely to stimulate interest and is potentially a novel aspect of this work, with far-reaching implication considering that exchanging cell lines between labs is a routine practice in cancer research. The authors wrote... "We report phenotypic divergence between these assumptively "identical" cultures that appears to be driven by changes at the RNA level". What 'phenotype' is this? Given the authors' finding that passage duration can impact gene signatures, did they confirm that the 3 strains of MiaPaca2 were of the same/similar passages?

4. Another important aspect that should be developed more is the "normal" or "control" cell lines (HPNE, HPDE). This reviewer very much agrees that these cell lines are not entirely 'normal' and in fact manifest several key features of established cancer cell lines. The authors have a unique opportunity to leverage on their various sequencing data to drive home that point. Although an attempt was made in Figures 1E-H, the pancreatic cancer cell lines should be included in such comparisons. Perhaps also determine the degree of gene overlap between the individual pancreatic cancer cell lines and HPNE/HPDE cells.

5. That long-term culture of organoids or even any cell line, leads to transcriptomic changes is very much expected, albeit not well studied. Therefore, the authors may rather be specific and provocative here by emphasizing the unique features of the early vs late passage organoids. While the authors report that late passages acquire 'classical' subtype features, this observation is not novel as it has been reported in a yet to be published work (Raghavan et al, 2020, bioRxiv). Therefore, the authors should focus on other aspects such as proliferation, metabolism, expression of cancer pathway signatures, etc.

6. Besides, organoids normally do not do well with prolonged passaging as they best mimic human tumors at the early passages. This by itself raises concerns on the utility of comparing the early to late passages. The authors should discuss this concern, including e.g., the rationale for such comparison, in the Results and Discussions parts of the manuscript.

7. Where did these organoids come from? In describing the organoid generation, the authors started with 'Dissociated cells...' Are these patient-derived organoids or organoids made from cell lines? This is important because if the former, then details of acquisition and ethical approval should be provided; if the latter, then specify which cell line was used, etc. The lack of this information makes it hard for this reviewer to objectively critic this organoid part further.

Minor comments

On the brief title: “Intratumoral heterogeneity...”. Here ‘intratumoral’ is misleading since the study is purely in vitro.

While the manuscript is well written and easy to follow, there might be a few grammatical errors, e.g., “prototypal” was used rather than “prototypical” in the Introduction and “proscribed” (actually means ‘forbidden’) used in the Methods (instead of, I guess, ‘prescribed’). Such errors in choice of word or spellings should be checked throughout the manuscript to ensure clarity.

Reviewer #3 (Remarks to the Author): Expert in single-cell sequencing, analysis of cancer subclones and computational genomics

This study by Monberg et al. sets out to investigate the intratumoral heterogeneity of PDAC preclinical models. The study focused on the genomic and transcriptomic heterogeneity of cell line models of PDAC (BxPc-3, HPAF-11), between labs (3 x MiaPaCa-2 cell line sources), cell growth (Panc-1 2D vs 3D), a PDAC organoid and 2 commonly used “normal” pancreas immortalized cell lines. This paper has some extremely interesting findings, and others which are more confirmatory through newer techniques such as scRNA-seq/scCNV analysis.

However, as the paper is opening up critical questions of clonal variability among common pancreatic cancer cell lines and “normal transformed” lines, extensive detail on the growth conditions (serum, lot # etc.), passage number, length of time in culture before sequencing needs to be made clear in the materials and methods section.

The variability of clonal populations within the parental MiaPaCa-2 has long been established but this is the first time through scRNA-seq; additionally the custodial variability shows interesting clonal distinctions; however, the passage numbers of the three different stocks was not disclosed which would need to be factored into. How long the cells were cultured in the host laboratory before analysis was not disclosed either. Also the impact of different sources of serum in the media can also play a role in how cells behave. This may be true for the “normal transformed” HPDE line which may have been cultured over years of sub clonal growth selection, therefore lab passage number is important information to disclose.

How long was the Panc-1 cell line grown in 3D before analysis?

Additionally, the passage number of the PDO was not shared – what refers to early and late passage? 10 passages separate the PDO cultures but from what starting point? Can the authors clarify what they consider to be “early passages of derivative PDOs”? Can name of PDO be shared, as this would be good information for the research community.

The functional impact of sub-clonal populations with differing transcriptomic enriched pathways was not looked at. Although much literature will attest to the diverging biological functionality of 2D vs 3D; the biological impact of sub clonal populations within differing custodial vials of MiaPaCa-2 would have

added to the impact of the study; particularly as these cell lines are used to study drug toxicity and response to new therapeutic agents. Do these clonal diverse populations affect response to in vitro drug testing?

This study was carried out very comprehensively and study showed in the most part a high degree of concordance between the CNV profiles, scRNA-seq data and the bulk WES-seq data which adds to the strength of the findings. The data analysis mainly used publicly available software sources and data made available.

Challenging the status of PDAC cancer cell “models” in this paper highlights the vulnerabilities of reproducibility of research, and questions the repeatability and productivity of real findings. Furthermore, the implication of “normal transformed” with major chromosomal copy number events and genomic alterations highlights the urgent needs for better development, characterization and sharing of effective normal controls for in vitro research, all of which are critical for progress against this devastating disease.

Minor comments:

Results description of figure 1 (and supplementary data) a bit disjointed and difficult to follow as references to figures/supplementary files is disorderly.

Submitted version of supplementary excel file is formatted incorrectly.

Reviewer #4 (Remarks to the Author): Expert in single-cell ATAC-seq

In this work, Monberg et al. generate single-cell RNA and copy number profiling for multiple cell lines used as models of pancreatic adenocarcinoma (PDAC), along with commonly used "control" lines. This includes detailed investigations of cell lines obtained from multiple sources, as well as those grown in differing physical conditions (2D vs 3D). The main overall conclusions are: (i) significant transcriptomic variability between commonly used cell lines, (ii) single cell CNA profiling exhibits substantial genetic drift in multiple PDAC lines and in a "normal" control line, (iii) there is transcriptomic variability between supposedly identical lines from different sources, and (iv) there is transcriptomic variability depending on if the line is grown in 2D vs 3D, which scCNV-seq and follow up sn-ATAC-seq confirms is majority epigenomic plasticity rather than genomic.

Overall, the findings and data generated by this study will be an important resource for the community. The manuscript could be greatly improved in several ways, most notably in relation to detailing of computational methods and validation of some of the scCNA findings. Major and minor comments below.

Major -----

A major impact of this study comes from the resulting data to which researchers can compare their results without having to generate de novo. However, there is no data availability statement included in the manuscript.

The figures need a significant amount of improvement to be publication quality and many are too pixellated to read. In particular:

- Fig 1C is pixellated and it's impossible to read chromosome names
- Fig 1D has no X or Y axes and is too pixellated to make out what's going on
- Figures are referenced out of order which makes it highly confusing to read
- Figure 4E has zero average expression (grey) for several points but the size indicates a decent proportion of non-zero expression?

The methods for the analysis of the scCNA data are inadequately described. Most of the methods description come down to the fact that "the Cell Ranger DNA software" was run, but this does not appear to be a peer reviewed approach and is in fact a discontinued product. The authors should detail:

- how are raw sequencing reads processed to copy number calls? What is the bin size? Is any GC correction performed?
- how are copy number (breakpoints) determined? how is ploidy determined?
- how are cells clustered into clades / clones?
- what does "Heatmap data for trees containing all leaves within the maximum node on a per-sample basis were exported" mean?

Can authors report % genome non diploid or # breakpoints as a measure of genomic heterogeneity rather than number of clones which will be sensitive to the definition of a clone?

The authors find BxPC3 has copy number > 4 in TERT as a striking finding - this should be validated using e.g. FISH to confirm it is not an artefact of the analysis.

Similarly, the authors find "AURKA amplification in HPDE, a purportedly "control" cell used in PDAC research, including in experimental therapeutics studies" as a major finding of the method, but this should be validated as above.

In figure 1E and 1F the authors compare their pseudobulk single cell CNV data to WES. However, it is not actually detailed where this WES comes from. Is this existing bulk data? Was it generated for this study? How was it processed and computationally analyzed?

The authors find that HPDE is more transcriptionally similar to HPAF-II when looking at "average distance in rescaled diffusion space" - given the complexity of the underlying algorithm to arrive at these distances, can the authors show this result holds when using a simpler distance or similarity metric? For example, Pearson correlation has been show to work well (<https://pubmed.ncbi.nlm.nih.gov/30137247/>).

The authors assert that "HPNE cells are non-ductal in derivation (as demonstrated by lack of epithelial markers)", but only show expression profiles for 2 epithelial markers (EPCAM and KRT8). Can the authors show a more comprehensive list of epithelial markers (e.g. E-Cadherin, KRT18, KRT19)? Also given this is shallow scRNA-seq there is a chance these genes are expressed but simply not detected (i.e. a zero count from scRNA-seq does not confirm "non-expression"). Can the authors validate this observation via FISH/IHC?

In multiple places in the manuscript (e.g. MP2 analysis) the authors find cells cluster distinctly. However, can this be ruled out as some technical artefact e.g. the effect of cell cycle, sequencing depth, etc? Could the authors take marker genes and IHC to confirm it's real?

Figure 4E - these do not seem to be all the classical/basal genes - how was this subset selected? Again authors use "not expressed" when really it should be not detected.

The authors identify distinct expression of basal genes between early/late and the switching on of several classical genes from early/late, and conclude "We thus observe a subtype admixture over time", but it seems rather the transcriptomic profiles don't actually match the classical/basal subtypes rather than representing an admixture. Since Fig 4E/F represent average expression over cells, can the authors check this distinction by examining whether classical/basal identity is truly maintained at single cell resolution and the changes represent subtype admixture switching, or whether the actually expression profiles don't agree with the classical / basal subtyping (which on its own would be an important observation).

Minor -----

"chromosomal locations as provided by 10x Genomics software were mapped to gene-specific loci" how often do genes span breakpoints?

The authors introduce that they perform clonal assignment using clonealign but then state "An important caveat of this method is that inferCNV's inherent design filtered out a large number of cells sequenced by scCNVseq" - is "inferCNV" a typo or is inferCNV used as part of the analysis too?

Discussion of basal/classical assignment: "sorted as" -> "assigned as"

"we paneled for genes relevant to the Moffitt subtyping" -> what does panel mean as a verb?

We thank the reviewers for their time and investment in our manuscript. We greatly appreciate their effort and comments to improve the quality of this work. In our efforts to proceed with the publication of our manuscript, titled "Occult polyclonality of preclinical pancreatic cancer models drives in vitro evolution", we have chosen to address the reviewers' comments to the best of our abilities in the responses detailed below. Reviewer comments are in *Italics*, our responses are non-italicized. References cited for each reviewer have been listed at the end of the rebuttal document, organized on a per-reviewer basis. Supporting figures, tables, and discussions have been incorporated into the body of each response. Additionally, we have attached a copy of all revisions figures and tables, with annotated legends, at the end of this document.

REVIEWER COMMENTS

Reviewer #1 (Remarks to the Author):

Major comments

1. There are numerous studies detailing the heterogeneity of pancreatic cancer cell lines, including differences in marker expression, nutrient utilization (e.g., glycolysis vs OXPHOS), and growth properties. Deer et al (cited as ref. #11 in the manuscript) reviewed many aspects of divergence among pancreatic cancer cells. This reviewer therefore believes that the manuscript in the current state has not demonstrated novelty despite the technologies applied. For example, could the authors relate any of the observed transcriptomic differences to phenotypic differences – e.g., do the three 'strains' of MiaPaca2 cells differ in growth rate, nutrient consumption, etc, and does that correlate with their respective gene changes? This will help readers to better contextualize the potential translational relevance of this study.

- We agree with and thank the reviewer for highlighting the importance of biological activity and functional readouts of transcriptionally-derived differences across these cell lines. To address the question of growth across the different cell lines, we have conducted a cell cycle scoring analysis on MP2 cell lines and show that, cell cycle differences between cell lines are not extreme enough to explain the very large variability we see *between* cell lines. We find that a large proportion of cells are in S or G2M phase in all of these cell lines (MP2 51.5-62.6%). We do not wish to include these data in the Revised Manuscript, but hope that the reviewer is satisfied with our explanation as provided in Reviewer-only Revisions Figure 2.
- While we did not conduct "traditional" growth or migration assays for this study, we point to our GSEA analysis findings, as supportive evidence that the phenotypes across the MiaPaca2 strains do indeed differ. For example, we have updated Main Figure 2 to now reflect only-significant GSEA pathways that are differentially enriched across the 3 strains, emphasizing differences in Hallmark Oxidative Phosphorylation and Hallmark Protein Secretion pathways, which indicate phenotypically distinct metabolic changes across MiaPaca2s. Next, we have strengthened this analysis by conducting inferCNV on the 3 MiaPaca2 strains, compared to HPNE as a reference control. The inferCNV results have also been added to Main Figure 2, and show differences in predicted scCNV events most strikingly in chromosomes 5, 11, 15, 17, and 19. This also suggests a true phenotypic difference between the cell strains. The main text has been amended to reflect this improved analysis and also point out the distinction between events that inferCNV predicted to be differential across the strains, versus the events that were actually sequenced at the scCNV level in these cells.

2. It seems the rationale for selecting the cell lines was just based on the fact that they are commonly used in research. This does not necessarily add value. These cell lines even differ in their growth media conditions (as rightly noted by the authors in 'Methods'); that alone can drive differences in transcriptomics. The cell

lines should thus be presented based on their distinct phenotypes. One good parameter could be basal differences in their growth rate or migration (fast vs slow, etc), which can be related to cancer progression. Then, determine whether cells that more closely mimic pancreatic cancer features undergo similar degree of heterogeneity upon culture or are more robust. This level of detail should be provided. With such information, the potential value of using these cell lines will become more or less clear especially when complemented with the novel technologies used in this study.

- We thank the reviewer for their perspective and wish to emphasize that the aim of this study was not to present a comparative analysis across the phenotypes of well-known PDAC cell lines. Given the widely-accepted use and culture conditions of the cell lines studied here, we do not believe that we need to continue to overstate the value of these lines to the PDAC field, or their relevance for preclinical study. Many previous reports have already done so [Refs. 1-5]. Instead, we provide here novel findings of the existence of multiple different genomic and transcriptomic clones within each line, point out the reprogramming induced by transitioning to a 3D culture medium, To ensure that readers understand the extensive phenotype study that has already been done on these lines at the bulk level, we have clarified our language in the introduction to include the references described here, now stating on page 3:
 - “Here we present an in-depth single cell genomic and transcriptomic assessment of clonal heterogeneity in a panel of established and globally utilized PDAC cell lines (Panc-1, MiaPaCa2, HPAF-II and BxPC-3), immortalized “control” cells (HPNE and HPDE), and in an independent PDO which is compared to its earlier passage prior to prolonged ex vivo propagation. We demonstrate that pancreatic cell lines - neoplastic and non-neoplastic - are composed of remarkable sub-clonal heterogeneity at single cell resolution, which is unlikely to be detectable by conventional “bulk” profiling, and has not been previously reported on, despite extensive “traditional phenotypic” characterization of these lines [21-28].”
- These cell lines have already been exhaustively characterized by “traditional” phenotypic assays (described in main text Ref. 11, Revisions Refs. 6-8, and innumerable other publications). The results of these studies have provided the foundation for the wide use and applicability of these cell lines, as well as established their exemplary phenotypes as “hallmark pancreatic cancer” cell lines. Because of this, our study does not aim to reinvent the wheel. Instead, we sought to apply high-resolution tools to better characterize preclinical models whose features and phenotypes are so well-understood in the field of pancreas research that they may not be considered by researchers as potential sources of experimental convolution.

3. Based on the above point on differences in culture condition, the authors may consider focusing more on cells that are grown under the same culture conditions, e.g., the MiaPaca2 ‘strains’ – transcriptomic and phenotypic differences, overlap with pancreatic cancer subtypes, etc. This ‘strains’ comparison is most likely to stimulate interest and is potentially a novel aspect of this work, with far-reaching implication considering that exchanging cell lines between labs is a routine practice in cancer research. The authors wrote... “We report phenotypic divergence between these assumptively “identical” cultures that appears to be driven by changes at the RNA level”. What ‘phenotype’ is this? Given the authors’ finding that passage duration can impact gene signatures, did they confirm that the 3 strains of MiaPaca2 were of the same/similar passages?

- The reviewer is correct in requesting further explanation of the interpretation of our findings, especially with respect to the portion of our study concerning MiaPaca2 strains. The MP2A strain was obtained directly from the ATCC and expanded twice in our hands. MP2B and MP2C strains were also thawed and expanded twice in our hands, with all three strains being permitted to grow in parallel, in the same

incubator, for 4 weeks total. As described in the Methods section of the manuscript, aliquots of each of the cultures were sent for fingerprinting and confirmed to be MiaPaca2 cells prior to harvesting the cells for scRNAseq + scCNVseq. Our goal, in designing the experiment in this way, was to mimic, as closely as possible, what any lab would do in preparation for an experiment or drug screen using cell lines. The general phenotype of MiaPaca2 cells is very well described in literature, as these represent a form of PDAC driven by a KRAS G12C mutation (present in ~1.6% of PDAC cases) instead of the KRAS G12D mutation (present in >40% of PDAC cases). We apologize for the use of the word “phenotype” in describing our findings, and will instead be replacing it with terminology referring to transcriptomic features, avoiding the functional connotation that “phenotype” carries.

- In the MiaPaca2 strains, the point we wished to highlight is not necessarily that passage duration drives transcriptional reprogramming, as we’ve shown with our PDO data. Instead, we wished to demonstrate that while passage number and supposed culture conditions of a cell line may be nearly identical across different laboratories, the practicality of such findings and their translational implications might greatly vary.
- To further address the extent of transcriptomic divergence across MiaPaca2 strains, we mapped scRNAseq data from each strain to gene sets representative of existent clinical subtypes of pancreatic cancer, depicted in Supplementary Figure 5. To that end, we find that MP2A and MP2C are most representative of the Moffitt Classical subtype, which has been shown to yield more favorable outcomes in the clinic, but this “representation” is based solely on the expression of a very few genes specific to the subtype. The MP2B strain possesses a seemingly more heterogeneous phenotype, harboring cells of both Moffitt Classical and Basal clinical subtypes. However, as we note in the main text, “Rather, our data indicates that PDAC cell lines are sub-optimal models for application of tissue-based classification systems like Moffitt, largely because they might lack expression of substantial numbers of transcripts required for meaningful classification”.

4. Another important aspect that should be developed more is the “normal” or “control” cell lines (HPNE, HPDE). This reviewer very much agrees that these cell lines are not entirely ‘normal’ and in fact manifest several key features of established cancer cell lines. The authors have a unique opportunity to leverage on their various sequencing data to drive home that point. Although an attempt was made in Figures 1E-H, the pancreatic cancer cell lines should be included in such comparisons. Perhaps also determine the degree of gene overlap between the individual pancreatic cancer cell lines and HPNE/HPDE cells.

- To visualize the degree of overlap in structural events affecting key PDAC progressor genes, we have amended supplementary figure 2 to include HPDE in comparison to the other cell lines within the same figure. In this figure, we can see overlap in key structural events between HPDE and other pancreatic cancer cell lines. This overlap includes deletion of FGFR1 common to both HPDE and PANC1, as well as amplification of GNAS common between HPDE, HPAF2, and BXPC3.

Although we focus on the comparison between HPDE and HPNE in the main text, the FISH also confirms the finding from the scCNV data that HPDE, HPAF2, and BXPC3 all have at least some degree of amplification of AURKA relative to HPNE.

5. That long-term culture of organoids or even any cell line, leads to transcriptomic changes is very much expected, albeit not well studied. Therefore, the authors may rather be specific and provocative here by emphasizing the unique features of the early vs late passage organoids. While the authors report that late passages acquire ‘classical’ subtype features, this observation is not novel as it has been reported in a yet to

be published work (Raghavan et al, 2020, bioRxiv). Therefore, the authors should focus on other aspects such as proliferation, metabolism, expression of cancer pathway signatures, etc.

- We thank the reviewer for their astute observation that reprogramming of PDOs toward one clinical phenotype versus another is an insufficient comparison. Because of this, a comparison of Moffitt phenotypes was only allocated 2 panels of Main Figure 4. By conducting pseudotime analysis and subsequent GSEA (using Gene Ontology terms) on scRNAseq pseudotime-derived clusters, depicted in Main Figures 4A-4D, we show that as this particular PDO progresses in culture, genes associated with DNA Replication, Cell Cycle, Cell Growth (and many more pathways associated with proliferation/metabolism) are differentially enriched between the 2 resulting fates of the late-passage PDO, and that these 2 resulting fates are also phenotypically unique from the GSEA enrichment profile of the early PDO. We also conducted inferCNV in early versus late passage PDO clusters (Main Figure 4D) and show differential copy number aberrations in the late passage, implying a rather substantial transcriptomic reprogramming over time in culture.

6. Besides, organoids normally do not do well with prolonged passaging as they best mimic human tumors at the early passages. This by itself raises concerns on the utility of comparing the early to late passages. The authors should discuss this concern, including e.g., the rationale for such comparison, in the Results and Discussions parts of the manuscript.

- We agree that prolonged culture of organoids, especially of the patient-derived nature, can produce misleading experimental results. However, in the field of pancreatic cancer research, it is not well-documented, or reported on, for that matter, how long organoids can be in culture before profound transcriptional reprogramming takes place. For these reasons, it is critical to the pancreatic cancer research field that parameters/best practices be established to encourage laboratories and research groups working with organoid models to regularly characterize their specimens to assess the extent of “drift” from the original tumor. We have amended our discussion and presentation of PDO results to reflect these concerns.

7. Where did these organoids come from? In describing the organoid generation, the authors started with ‘Dissociated cells...’ Are these patient-derived organoids or organoids made from cell lines? This is important because if the former, then details of acquisition and ethical approval should be provided; if the latter, then specify which cell line was used, etc. The lack of this information makes it hard for this reviewer to objectively critic this organoid part further.

- We thank the reviewer for their concern for research ethics and proper management of clinical tissues. This PDO was generated from biopsy cores taken during an EUS-FNA diagnostic procedure at a metastatic site from a diagnosed pancreatic cancer primary lesion. The biopsy cores were dissociated according to the Methods described, and the PDO was thus grown from those cells which survived dissociation. The patient was seeking treatment at MD Anderson Cancer Center. The details of our IRB specifications and clinical tissue acquisition protocols have been added to the materials and methods section.

Minor comments

On the brief title: “Intratatumoral heterogeneity...”. Here ‘intratumoral’ is misleading since the study is purely in vitro.

- We agree that the “intratumoral” label is not entirely specific or appropriate for the context presented. Brief title now reads: “Characterizing heterogeneity of PDAC preclinical models.”

While the manuscript is well written and easy to follow, there might be a few grammatical errors, e.g., “prototypal” was used rather than “prototypical” in the Introduction and “proscribed” (actually means ‘forbidden’) used in the Methods (instead of, I guess, ‘prescribed’). Such errors in choice of word or spellings should be checked throughout the manuscript to ensure clarity.

- We thank the reviewer for their mindfulness and concern of word choice. “Prototypal”, meaning “representing or constituting an original type after which other similar things are patterned”, used in our Introduction and Results sections (paragraph 1 of both) was indeed what we had intended. “Proscribed” appears to have been a typo, it has been corrected and is no longer used in the manuscript.

Reviewer #2 (Remarks to the Author):

This study by Monberg at al. sets out to investigate the intratumoral heterogeneity of PDAC preclinical models. The study focused on the genomic and transcriptomic heterogeneity of cell line models of PDAC (BxPc-3, HPAF-11), between labs (3 x MiaPaCa-2 cell line sources), cell growth (Panc-1 2D vs 3D), a PDAC organoid and 2 commonly used “normal” pancreas immortalized cell lines. This paper has some extremely interesting findings, and others which are more confirmatory through newer techniques such as scRNA-seq/scCNV analysis.

However, as the paper is opening up critical questions of clonal variability among common pancreatic cancer cell lines and “normal transformed” lines, extensive detail on the growth conditions (serum, lot # etc.), passage number, length of time in culture before sequencing needs to be made clear in the materials and methods section.

- We thank the reviewer for their rigorous attention to detail! The growth conditions and percentage of fetal bovine serum used for each culture are now specified in the methods. Catalog numbers for all cell lines used in this study have also been added. The cell lines were all in culture for 4 weeks following thawing, prior to being harvested for sequencing. The Methods section has now been edited to reflect the passage number of each line used, where applicable.

The variability of clonal populations within the parental MiaPaCa-2 has long been established but this is the first time through scRNA-seq; additionally the custodial variability shows interesting clonal distinctions; however, the passage numbers of the three different stocks was not disclosed which would need to be factored into.

- We thank the reviewer for this comment. Indeed, passage numbers are important! We wish to emphasize that we are referring to passage numbers in our hands, and we (nor can any laboratory) cannot account for true passage numbers in lines that are long standing and have been used for decades. We also wish to note that the general lack of “passage accountability” in these long-standing lines does present myriad issues for research in general. In this study, the MP2-A was obtained from the ATCC, thawed & plated, and underwent two expansions prior to harvesting for sequencing. MP2-B was labeled at passage 2 when it was “donated” to us by a collaborating lab, and plated for an additional two expansions in our hands, and then harvested for sequencing. MP2-C was thawed and plated from passage 3, underwent 2 expansions, and was then harvested for sequencing experiments. The Methods have been edited to reflect this information.

How long the cells were cultured in the host laboratory before analysis was not disclosed either.

- MP2-B was obtained at “passage 2” from a collaborating lab, and was in culture in our hands for less than 10 passages prior to sequencing.

Also the impact of different sources of serum in the media can also play a role in how cells behave.

- We agree with the reviewer. Thus, in our experimental design, we were careful to use the same “type” of FBS for our cell lines, catalog information is now listed in our Methods section.

This may be true for the “normal transformed” HPDE line which may have been cultured over years of sub clonal growth selection, therefore lab passage number is important information to disclose.

- We agree with the reviewer. However, HPDE cell line is commercially available from Kerfast, and while it is true that any cell line may have been cultured over years of subclonal growth selection, we tried our best to avoid this by ordering a new stock from Kerfast for this experiment. The cells were plated, underwent 4 passages/expansions, were fingerprinted by the MDACC Core to ensure cellular molecular identity, and were then harvested for sequencing in this experiment set.

How long was the Panc-1 cell line grown in 3D before analysis?

- Panc1 cells from frozen stock were thawed- half into a 2D culture dish, half into matrigel for spheroid generation. The cells in both culture conditions then underwent 2 passages over the course of 4 weeks of total growing time prior to being harvested for parallel processing of scRNAseq + scCNVseq.

Additionally, the passage number of the PDO was not shared – what refers to early and late passage? 10 passages separate the PDO cultures but from what starting point? Can the authors clarify what they consider to be “early passages of derivative PDOs”? Can name of PDO be shared, as this would be good information for the research community.

- We apologize for the lack of clarification regarding early and late passaging of the PDO. This organoid was generated and allowed to grow in PDO media/matrigel for no more than 4 passages prior to being initially sequenced. Thus, “early” = Passage 4, “late” = Passage 14 of the PDOs. The Methods have been edited to reflect this information.
- To protect patient privacy, the PDO described here is titled PDO1. Given that we did not order the PDO from a large biobank or repository (such as the ATCC) and generated the PDO “in house”, it is not available to the research community at this time. However, we fully acknowledge the importance of equal access and sharing of critical/rare specimens within the research community, and support doing so.

The functional impact of sub-clonal populations with differing transcriptomic enriched pathways was not looked at. Although much literature will attest to the diverging biological functionality of 2D vs 3D; the biological impact of sub clonal populations within differing custodial vials of MiaPaCa-2 would have added to the impact of the study; particularly as these cell lines are used to study drug toxicity and response to new therapeutic agents. Do these clonal diverse populations affect response to in vitro drug testing?

- We agree with the reviewer that our study would be strengthened by a “functional” element of investigation, although that was not within the scope of our research questions when we designed the work. While we did not conduct in vitro drug testing in these cell lines, to demonstrate functional relevance detectable at the scRNA level, we applied an in-house bioinformatic pipeline to predict compounds using the Broad Institute’s Clue.io Database derived from the Connectivity Map Project [Ref. 9] in Panc1 2D

versus 3D samples. In doing so, we find striking differences in compounds that even individual clusters within cell lines may be susceptible to, indicated by a red color, or incur minimal transcriptional consequences from, indicated by a blue color (Reviewer-only Revisions Figure 6). In particular, we show here that Panc1 3D Cluster 0 is most likely sensitive to PI3K/mTOR inhibitors including dactolisib (Reviewer-only Revisions Figure 4A), and Panc1 2D Cluster 0 would likely respond to HSP inhibitor BIIB021 (Reviewer-only Revisions Figure 4B). It is important to note that in the data presented in Main Figure 3, Panc1 2D clusters were grouped together after determining high transcriptomic similarity. But the scRNAseq data shown here is essentially the same as what we've presented in the Main text and figures. While we do not wish to include these analyses in our final manuscript, nor do we have the resources at this time to conduct a drug screen in silico to validate these predictions, we are sharing this data with the reviewers here to demonstrate a Proof-of-Concept that diverse clonal populations may very well impact response to therapeutics, thereby highlighting the importance of our study to the research community's selection of appropriate preclinical models for cancer research.

Reviewer-only Revisions Figure 4. Comparative analysis of scRNA subclusters derived from Panc1 3D (A) and 2D (B) models. Using Clue.io database, genes differentially up or downregulated per cluster were correlated to chemical agents that would induce transcriptional reprogramming (red) or not (blue). Clear differences between subclusters across samples demonstrate intrasample heterogeneity, as well as spheroid reprogramming resulting in predicted sensitivity to MTOR inhibition.

This study was carried out very comprehensively and study showed in the most part a high degree of concordance between the CNV profiles, scRNA-seq data and the bulk WES-seq data which adds to the strength of the findings. The data analysis mainly used publicly available software sources and data made available.

Challenging the status of PDAC cancer cell “models” in this paper highlights the vulnerabilities of reproducibility of research, and questions the repeatability and productivity of real findings. Furthermore, the implication of “normal transformed” with major chromosomal copy number events and genomic alterations highlights the urgent needs for better development, characterization and sharing of effective normal controls for in vitro research, all of which are critical for progress against this devastating disease.

Minor comments:

Results description of figure 1 (and supplementary data) a bit disjointed and difficult to follow as references to figures/supplementary files is disorderly.

- The reviewer is completely correct, and we appreciate their contribution to the improvement of our study. The figure legends have been adjusted to be more clear, and the order of the figures and panels within them now reflects their discussed order in the text, as much as we could to still allow for the figures to be aesthetically pleasing.

Submitted version of supplementary excel file is formatted incorrectly.

- We thank the reviewer for this comment. We have distilled the GSEA results initially submitted in Supplementary Excel File to a single table now submitted as Supplementary Table 2. We hope this has solved the formatting issue and has made our results more clear and accessible.

Reviewer #3 (Remarks to the Author):

Major -----

A major impact of this study comes from the resulting data to which researchers can compare their results without having to generate de novo. However, there is no data availability statement included in the manuscript.

- The reviewer is correct in that all data should be accessible and made public to the community. We have already submitted all scRNAseq + snATACseq data to GEO, and scCNV data to the ENA. In our initial submission to Nature Communications, we included a data availability statement specifying these details, as well as Reviewer Tokens for GEO so that the Reviewer could interrogate the data we present here.

The figures need a significant amount of improvement to be publication quality and many are too pixellated to read. In particular:

- *Fig 1C is pixellated and it's impossible to read chromosome names*
- *Fig 1D has no X or Y axes and is too pixellated to make out what's going on*
- *Figures are referenced out of order which makes it highly confusing to read*
- *Figure 4E has zero average expression (grey) for several points but the size indicates a decent proportion of non-zero expression?*

- We are happy to improve the quality of our submitted figures, and have thus increased the resolution and improved the readability of our figures, as per the Reviewer's recommendation. We've also rearranged the text to reflect the figure numbers so that they are in a more chronological order. We hope that these changes have addressed concerns of readability and quality.
- In order to address size and readability, we have added axes labels to Main Figure 1D from the original format, and have now moved it to Supplemental Figure 1C. This allows for more space for Figure 1C so that readers can fully appreciate the copy number variations across thousands of cells in each scCNVseq dataset. In the "new" Supplemental Figure 1C, we have also improved the legend to better depict variations in event heterogeneity across cells between cell line samples.
- Regarding the reviewer's comment on Figure 4E: we are unsure how to address this concern, and are not exactly sure why the reviewer has raised this as an issue. We agree that a large proportion of cells in the

PDO have non-zero expression of certain genes depicted, and there are also large proportions of cells that have very low relative expression of certain genes, as well (represented as near-zero values in the dotplot). To provide qualitative evidence for this portion of the analysis, we have included here tables derived from our scRNAseq data showing the average expression values for each of the genes described (Reviewer-only Revisions Tables 2 + 3, below). We do not wish to include this data in the supplementary data or describe it in the main text, but are affixing it here to sufficiently address the reviewer's comment.

Reviewer-only Revisions Tables 2 + 3 (Table 2 = Moffitt Basal Dataset, Table 3 = Moffitt Classical Dataset).

Moffitt Basal Gene Set Expression Values

	avg.exp	pct.exp	features.plot id	avg.exp.scaled
AREG	0.41885346	34.2384888	AREG PD01early	-0.707106781
CST6	0.32663247	32.1920504	CST6 PD01early	0.707106781
DHRS9	0.0306013	4.64384101	DHRS9 PD01early	-0.707106781
FAM83A	7.77524101	97.9929162	FAM83A PD01early	0.707106781
FGFBP1	0.5395463	52.2628886	FGFBP1 PD01early	-0.707106781
GPR87	0.23653783	34.9468713	GPR87 PD01early	0.707106781
KRT15	0.35558438	44.077135	KRT15 PD01early	0.707106781
KRT17	0.00095855	0.19677292	KRT17 PD01early	-0.707106781
KRT6A	0.00148028	0.27548209	KRT6A PD01early	-0.707106781
KRT7	4.99248259	98.2683983	KRT7 PD01early	-0.707106781
LEMD1	0.04245986	6.02125148	LEMD1 PD01early	-0.707106781
LY6D	0.61249558	45.4939	LY6D PD01early	-0.707106781
SLC2A1	0.56566447	62.5737898	SLC2A1 PD01early	-0.707106781
SPRR1B	0.0181479	1.21999213	SPRR1B PD01early	0.707106781
TNS4	0.18768806	23.5733963	TNS4 PD01early	0.707106781
VGLL1	0.9381014	75.2853207	VGLL1 PD01early	-0.707106781
AREG1	2.40283871	73.313897	AREG PD01late	0.707106781
CST61	0.22494331	15.1603499	CST6 PD01late	-0.707106781
DHRS91	0.04226508	3.45966958	DHRS9 PD01late	0.707106781
FAM83A1	3.73043495	85.9280855	FAM83A PD01late	-0.707106781
FGFBP11	1.33601402	57.7259475	FGFBP1 PD01late	0.707106781
GPR871	0.10932271	11.1175899	GPR87 PD01late	-0.707106781
KRT151	0.32626943	27.9105928	KRT15 PD01late	-0.707106781
KRT171	0.16077655	5.48104956	KRT17 PD01late	0.707106781
KRT6A1	0.00314922	0.15549077	KRT6A PD01late	0.707106781
KRT71	9.11948105	97.0845481	KRT7 PD01late	0.707106781
LEMD11	0.13020025	11.6423712	LEMD1 PD01late	0.707106781
LY6D1	2.61900164	67.696793	LY6D PD01late	0.707106781
SLC2A11	0.6335786	47.5218659	SLC2A1 PD01late	0.707106781
SPRR1B1	0.00936573	0.56365403	SPRR1B PD01late	-0.707106781
TNS41	0.04630835	4.72303207	TNS4 PD01late	-0.707106781
VGLL11	1.11406856	61.0495627	VGLL1 PD01late	0.707106781

Moffitt Classical Gene Set Expression Values

	avg.exp	pct.exp	features.plo id	avg.exp.scaled
AGR2	2.8211069	92.483274	AGR2 PD01early	-0.707107
AGR3	1.8740684	90.633609	AGR3 PD01early	-0.707107
ANXA10	0.0762617	12.081858	ANXA10 PD01early	-0.707107
BTNL8	0	0	BTNL8 PD01early	-0.707107
CDH17	0.00157	0.3148367	CDH17 PD01early	-0.707107
CEACAM6	4.5521738	89.177489	CEACAM6 PD01early	-0.707107
CLRN3	0.0009363	0.1574183	CLRN3 PD01early	-0.707107
CTSE	0.0203259	2.8728847	CTSE PD01early	-0.707107
CYP3A7	0	0	CYP3A7 PD01early	-0.707107
FAM3D	0.0016949	0.3148367	FAM3D PD01early	0.7071068
KRT20	0.0016988	0.1180638	KRT20 PD01early	-0.707107
LGALS4	0.5073716	47.540338	LGALS4 PD01early	-0.707107
LYZ	0.001626	0.2361275	LYZ PD01early	-0.707107
MYO1A	0	0	MYO1A PD01early	-0.707107
PLA2G10	0.3390262	42.030697	PLA2G10 PD01early	-0.707107
SPINK4	0.021462	3.227076	SPINK4 PD01early	-0.707107
ST6GALNAC	0.0311629	5.588351	ST6GALNAC PD01early	0.7071068
TFF1	0.0054617	1.0232192	TFF1 PD01early	-0.707107
TFF2	1.2610681	73.396301	TFF2 PD01early	-0.707107
TFF3	0.0123643	2.3612751	TFF3 PD01early	-0.707107
TSPAN8	0.0606379	9.4844549	TSPAN8 PD01early	-0.707107
VSIG2	0.0825113	14.049587	VSIG2 PD01early	-0.707107
AGR21	5.3796299	92.225462	AGR2 PD01late	0.7071068
AGR31	2.5813036	83.945578	AGR3 PD01late	0.7071068
ANXA101	0.5133468	37.356657	ANXA10 PD01late	0.7071068
BTNL81	0.0004682	0.058309	BTNL8 PD01late	0.7071068
CDH171	0.0038318	0.505345	CDH17 PD01late	0.7071068
CEACAM61	10.601322	94.499514	CEACAM6 PD01late	0.7071068
CLRN31	0.0206115	2.1963071	CLRN3 PD01late	0.7071068
CTSE1	0.0816948	5.3838678	CTSE PD01late	0.7071068
CYP3A71	0.0007674	0.0971817	CYP3A7 PD01late	0.7071068
FAM3D1	0.0012943	0.1749271	FAM3D PD01late	-0.707107
KRT201	0.0134115	1.1273081	KRT20 PD01late	0.7071068
LGALS41	0.7265637	35.393586	LGALS4 PD01late	0.7071068
LYZ1	0.0391218	3.1098154	LYZ PD01late	0.7071068
MYO1A1	0.0021196	0.1943635	MYO1A PD01late	0.7071068
PLA2G101	0.755368	49.757046	PLA2G10 PD01late	0.7071068
SPINK41	0.1994586	13.100097	SPINK4 PD01late	0.7071068
ST6GALNAC1	0.0256968	2.9931973	ST6GALNAC PD01late	-0.707107
TFF11	0.2841931	24.586978	TFF1 PD01late	0.7071068
TFF21	5.5394443	90.631681	TFF2 PD01late	0.7071068
TFF31	0.0207815	1.7881438	TFF3 PD01late	0.7071068
TSPAN81	1.2002164	55.918367	TSPAN8 PD01late	0.7071068
VSIG21	0.1753376	17.55102	VSIG2 PD01late	0.7071068

The methods for the analysis of the scCNA data are inadequately described. Most of the methods description come down to the fact that "the Cell Ranger DNA software" was run, but this does not appear to be a peer reviewed approach and is in fact a discontinued product. The authors should detail:

- how are raw sequencing reads processed to copy number calls? What is the bin size? Is any GC correction performed?

- *how are copy number (breakpoints) determined? how is ploidy determined?*
- *how are cells clustered into clades / clones?*
- *what does "Heatmap data for trees containing all leaves within the maximum node on a per-sample basis were exported" mean?*

- The Cell Ranger DNA software is a publicly-available software pipeline, which is also contained in the MD Anderson Cancer Center High Performance Cluster for computing purposes. The link to this widely-used and accepted software package is here: <https://support.10xgenomics.com/single-cell-dna/software/pipelines/latest/what-is-cell-ranger-dna> .
 - All of the specific algorithmic features of the software are also contained in this page, which is maintained by 10x Genomics. The library preparations reagents for constructing scCNVseq libraries are no longer sold by 10x Genomics as a function of a lawsuit with Biorad and also as a result of the extreme price of these products having decreased accessibility to the general research community. However, the software package itself is not at all defunct and is still highly applicable to scCNVseq datasets, wherever they happen to exist, and can be applied to scCNV datasets generated by other library preparation kits, i.e. those sold by MissionBio and Takara.
- We now provide more details for the analysis,
 - The Cell Ranger DNA software first preprocesses the data by aligning the reads to the reference genome and marking potential PCR duplicates. Viable cells are identified by thresholding barcodes with sufficient reads. Coverage of reads is computed over 20 kb bins over the genome.
 - The cells are clustered using the complete linkage hierarchical clustering method on a coarse-grained copy number matrix, whose window size allows for 200 reads per cell.
 - Copy numbers are then called for each of the cell clusters using a Poisson model with multiplicative effect of the CNV, sequencing depth, GC content, and mappability. The effect of GC content is modeled by a quadratic function of the percentage of GC fitted on the data. Bins are grouped into segments using a likelihood-ratio model to determine breakpoints, and the sequencing depth is estimated by finding a value that results in most integer CNVs for the segments. The mappability is defined as simulated reads that can be mapped back to account for repetitive genome regions.
- However, over the process of our revisions, and encouraged by this reviewer's comments regarding the potential issues with the 10x genomics software, we have revisited our analysis of scCNVseq data, updated our methods section, and also reconfigured Main Figure 1C and Supplementary Figure 2 to avoid relying on the 10x software's definition of a clone. Instead, we only use the 10x software output of segment coordinates and integer copy number values, and display the values for relevant regions or genes on a per-cell rather than per-clone basis. Our main findings of clonality and chromosomal amplifications/deletions remain very similar to what was initially reported in our first submission, but we hope the reviewer finds this analysis more technically acceptable. Our methods section for supplemental figure 2 now reads:

Gene coordinates (as defined by Ensembl v84 annotation) were intersected with the node_cnv_calls.bed file for each sample to get an integer copy number value per cell, per gene in a list of key PDAC progressor genes. The segments output by the CellRanger software are always at least 20kb, and often much larger, so this intersection almost always resulted in a one-to-one correspondence of segments to genes. In the handful of cells where multiple segments overlapped the same gene, the copy number for the segment with the most overlap by coordinates was used. For the heatmap of non-HPNE cell lines, a random 1000 cells per sample were chosen for visualization purposes using the "sample" function in R.

Can authors report % genome non diploid or # breakpoints as a measure of genomic heterogeneity rather than number of clones which will be sensitive to the definition of a clone?

- Certainly. The results of such quantification are given below in Reviewer-only Revisions Table 4.

Reviewer-only Revisions Table 4. Percent of genome non-diploid on a per-sample basis as derived from scCNVseq. This serves as a measure of genomic heterogeneity

	HPAF2	MP2-C	HPNE	HPDE	BXPC3	MP2-B
CN not 2	0.628328	0.815911	0.302481	0.794937	0.746022	0.808391

The authors find BxPC3 has copy number > 4 in TERT as a striking finding - this should be validated using e.g. FISH to confirm it is not an artefact of the analysis.

- The reviewer is correct! We have conducted FISH analysis to confirm our initially reported findings, and have included those results in Supplementary Figure 3. We have also amended the Main Text on page 6 to describe FISH results as they pertain to TERT amplifications across samples.

Similarly, the authors find "AURKA amplification in HPDE, a purportedly "control" cell used in PDAC research, including in experimental therapeutics studies" as a major finding of the method, but this should be validated as above.

- Again, the reviewer is completely correct! FISH has been done, and confirmed the AURKA amplification in HPDE, the results of which are now included in Supplementary Figure 3. The findings of this analysis have been added to the Main Text, page 7.

In figure 1E and 1F the authors compare their pseudobulk single cell CNV data to WES. However, it is not actually detailed where this WES comes from. Is this existing bulk data? Was it generated for this study? How was it processed and computationally analyzed?

- The reviewer points out an excellent oversight in our initial submission. WES was generated by our laboratory for this experiment and dataset. As such, the Methods section has been amended to include how WES was done and analyzed in our laboratory, as per our own previously-published protocols and methods in Semaan et al., 2021 [Ref. 10] The Main Text has also been edited on page 6 to reflect the curation of this dataset, now reading:
 - "(WES) on HPDE cells curated in our laboratory from the same pool used to generate scCNV and scRNAseq datasets confirmed that HPDE cells had undergone further bi-allelic losses of ..."

The authors find that HPDE is more transcriptionally similar to HPAF-II when looking at "average distance in rescaled diffusion space" - given the complexity of the underlying algorithm to arrive at these distances, can the authors show this result holds when using a simpler distance or similarity metric? For example, Pearson correlation has been show to work well (<https://pubmed.ncbi.nlm.nih.gov/30137247/>).

- Absolutely! We thank the reviewer for their thoughtfulness in considering Methods Complexity, and therefore the ability of readers to understand our work. While we do not wish to include the Pearson correlation method in our publication, we have conducted the analysis here for the reviewer’s consideration, showing that our reported results with Diffusion Space are consistent with “more traditional” analytical methods. Given the similarity of this reviewer’s comment with those made by Reviewer 4, we have attached the results of a Pearson correlation analysis in Reviewer-only Revisions Figure 1, panels C and D, as well as here, below. The main text of the manuscript has been altered to reflect this emphasis on algorithm reliability, now stating on page 4:
 - “To ensure rigor and reproducibility in our work, and to assure the reliability of our analysis when yet simpler algorithms are applied, we confirmed our findings by running a pearson correlation analysis (data not shown). In doing so, we report that the overarching structure of how cell line clades are organized remains highly similar to what was determined by diffusion mapping.”

Reviewer-only Revisions Figure 1, continued. Pearson correlation analysis of cell lines to corroborate reported findings of Diffusion Space scRNAseq analysis.

C.

D.

The authors assert that "HPNE cells are non-ductal in derivation (as demonstrated by lack of epithelial markers)", but only show expression profiles for 2 epithelial markers (EPCAM and KRT8). Can the authors show a more comprehensive list of epithelial markers (e.g. E-Cadherin, KRT18, KRT19)? Also given this is shallow scRNA-seq there is a chance these genes are expressed but simply not detected (i.e. a zero count from scRNA-seq does not confirm "non-expression"). Can the authors validate this observation via FISH/IHC?

- Given that the HPNE cell line was validated by way of Fingerprinting by the MD Anderson Cytogenetics Core and does not express epithelial-associated genes at the scRNAseq level, we believe that conducting IHC or FISH for these specific genes would be excessive. Further, EPCAM and KRT- genes are canonical markers for pancreatic ductal adenocarcinoma. HPNE cells, as published and as described in our manuscript, are not PDAC cells. HPNE cells instead represent a normal ductal-like phenotype, and do not express additional KRT markers when the scRNAseq assay is applied. Their “normal” genetic backgrounds, identified in scCNVseq and confirmed by WES, also provide support for this observation. We’ve included a violin plot in Reviewer-only Revisions Figure 5 showing expression across cell lines of a more exhaustive list of epithelial markers.

Reviewer-only Revisions Figure 5. Violin plot of expression of common epithelial marker genes in PDAC cell lines, showing very low relative expression in HPNE cells. Ductal marker SPP1 included as reference.

In multiple places in the manuscript (e.g. MP2 analysis) the authors find cells cluster distinctly. However, can this be ruled out as some technical artefact e.g. the effect of cell cycle, sequencing depth, etc? Could the authors take marker genes and IHC to confirm it's real?

- We thank the reviewer for their rigor in reviewing the quality of our submitted sequencing dataset. To ensure that clusters determined by scRNAseq were “real”, we insisted on including clonealign in our reported findings as a way to orthogonally validate scRNAseq-determined clones by the specific genomic events that might be driving those clusters. In Main Figures 2E-J and Supplemental Figures 6A-D, we describe the result of such analyses, pointing out that in MP2 and Panc1 samples (respectively), the “real” scRNA-derived clusters “boil down” to 4 categories: Clone 1, Clone 2, Clone 3, or unassigned. We then report specific GSEA differences across those clones as a function of their genetic uniqueness and the resulting transcriptional profiles. In doing so, we emphasize that scRNAseq-based “clusters” can be arbitrary if they are not orthogonally validated in some way. Here, our clonealign analysis linking

scCNVseq data argues exactly that point, and GSEA confirms transcriptional “realness” of such scRNAseq clones. Data stringency and analytical methods applied in scRNAseq, scCNVseq, and clonealign filtering ensured the highest data quality and ameliorated technical artifacts that may have otherwise been due to sequencing depth or cell cycle (all analysis packages applied have rigorous, peer-reviewed methods for overcoming the technical biases of these elements). Because we would be incapable of conducting such an analysis with IHC or FISH data alone, we believe that the findings presented here are not only “real”, but apply a relatively novel approach to assessing the validity of scRNAseq clustering methods.

- It is also important for us to note here that for the Panc1 2D/3D samples, we computationally merged scRNAseq datasets with snATACseq datasets, adding an additional validation method to understand the functional/“real” significance of clusters designated by scRNAseq analysis pipelines. Depicted in Main Figures 3F-I, we demonstrate that in addition to clusters being driven by genomic amplifications/deletions (Supplemental Figure 6), epigenetic differences drive cluster distinctions at the scRNAseq level in between these samples.
- We have also conducted a cell cycle analysis on the scRNAseq data of the cell lines, and have explained this analysis in our rebuttal to Reviewer #1, with the results depicted in Reviewer-only Revisions Figure 2. We do not wish to include these data in our final manuscript, but by our analysis, it is clear that the differences between clusters are not being driven by cell cycling differences across the cells of each sample.

Figure 4E - these do not seem to be all the classical/basal genes - how was this subset selected? Again authors use "not expressed" when really it should be not detected.

- The Moffitt Classification schema for pancreatic ductal adenocarcinoma transcriptional subtypes referred to in our manuscript is cited in the Main Text, and also listed at the end of this rebuttal document [Ref. 11]. We did not independently select these genes, they have been published in the literature for a number of years and are now applied in clinical trials for patient stratification, examples of which are also cited in the Main Text. As for the issue of not detected at all versus expression being low or uncommon (but not completely absent), we have included Reviewer-only Revisions Tables 3 + 4 above, quantifying the expression of each gene in each dataset, hoping that the reviewer will be satisfied with this explanation. We are happy to amend the language of the text to reflect the extremely low expression of certain Moffitt-associated genes in our dataset, and not “lack of expression”, and have done so on page 14 of the Main Text.

The authors identify distinct expression of basal genes between early/late and the switching on of several classical genes from early/late, and conclude "We thus observe a subtype admixture over time", but it seems rather the transcriptomic profiles don't actually match the classical/basal subtypes rather than representing an admixture. Since Fig 4E/F represent average expression over cells, can the authors check this distinction by examining whether classical/basal identity is truly maintained at single cell resolution and the changes represent subtype admixture switching, or whether the actually expression profiles don't agree with the classical / basal subtyping (which on its own would be an important observation).

- We agree with the reviewer that the distinction between subtype admixture switching versus disagreement with molecular profiles is important. To address this, we have generated a heatmap showing all measurable

genes within the scRNA analysis assay across early versus late PDO cultures. As you can see clearly in Reviewer-only Revisions Figure 6 below, there is very clear expression of many genes from both subtypes across many cells in the PDO late sample, whereas the majority of cells of the PDO early sample almost exclusively express genes associated with one subtype. As such, we will not be amending Main Figure 4 nor will we alter the original manuscript text regarding this issue, as the findings in Reviewer-only Revisions Figure 6 are agreeable with what we initially reported. However, given the importance this issue raises, we have decided to add this heatmap analysis to our newly-created Supplemental Figure 8 to support findings reported in Main Figure 4.

Reviewer-only Revisions Figure 6. Heatmap annotated per cell in Early versus Late PDO culture (columns) and genes (rows) associated with Moffitt Classical and Basal molecular subtypes. PDO1 early expresses genes almost exclusively associated with the Basal Subtype, while many cells within PDO1-late express genes that are either classical or basal in nature.

Minor -----

"chromosomal locations as provided by 10x Genomics software were mapped to gene-specific loci" how often do genes span breakpoints?

- Since the bin size is 20kb, genes spanning breakpoints are naturally rare.

The authors introduce that they perform clonal assignment using clonealign but then state "An important caveat of this method is that inferCNV's inherent design filtered out a large number of cells sequenced by scCNVseq" - is "inferCNV" a typo or is inferCNV used as part of the analysis too?

- We thank the reviewer for catching this important error on our part! We absolutely meant to refer to clonealign, NOT inferCNV here, and the main text has been amended to reflect this! Thank you very much!

Discussion of basal/classical assignment: "sorted as" -> "assigned as"

- Happy to adjust. Main text amended appropriately.

"we paneled for genes relevant to the Moffitt subtyping" -> what does panel mean as a verb?

- We had intended for “panel” to convey the process of obtaining a list or panel of genes and quantifying their expression in the scRNAseq assay. In this case, we understand the confusion our language here may create, and have opted to alter the Main Text. The text now reads, “Within the PDO scRNAseq analysis object, we searched for and quantified the expression of genes included in the Moffitt transcriptional subtyping schema, to understand subtype-specific ...”. We hope that the newly-worded text is more understandable to the readership, and is preferred by the reviewer.

Reviewer #4 (Remarks to the Author):

Major concerns

1. The authors perform single-cell profiling (mostly scCNV and scRNA) on separate cultures of the same cell lines (e.g. MiaPaca2 lines from three sources) and, observing differences in the single-cell data from these different sources, they conclude that the underlying cultures are different. However, the authors do not seem to account for the alternative explanation that some of the differences they observe are due to technical variability between runs of the single-cell profiling methods themselves. The single-cell data is not a perfect representation of the underlying samples. It has inherent technical biases and artifacts that can vary between runs, and can also suffer from sampling bias as <10K cells are generally profiled in a single run. In the context of this study, where single-cell data is being used to draw conclusions about differences in the underlying biosamples, I think it is essential to perform technical replicates where cells from the same culture are split into two aliquots and assayed independently. I am unable to draw conclusions about the differences between three MiaPaca2 cell sources, for example, based on a single replicate from each. Given the emphasis of these authors on reproducibility and rigor, this seems to be a particularly important flaw.

We appreciate the reviewer's dedication to ensuring the integrity of single-cell studies and raising very important commentary on the state of single-cell datasets. In our study conception and design, we were also acutely aware of the limitations and “cautionary tales” of generating single-cell experimental datasets, and thus generated the data presented in our manuscript according to

the following principles.

- To minimize the technical variability between sequencing runs, samples were sequenced together, in a single sequencing batch, according to their data type, to eliminate the potential of “sequencing-induced” batch effects, which are notorious to scRNAseq data.
- Given the exceptionally high cost of sequencing reagents, which is not at all feasible in the typical laboratory setting, we opted to forego constructing duplicate libraries and sequencing pools for this dataset. However, as further confirmation that “technical replicates” are not warranted in this experimental design, we have included here a comparative analysis of scRNAseq data between the Panc1 2D/3D (“Panc1.2D”) samples described in our manuscript with a previously-sequenced scRNAseq dataset derived from a Panc1 2D sample (“Panc1.2D.old”, done in our laboratory by the first author, M. Monberg). In doing so, we find a very close transcriptional similarity between the “old” and “new” Panc1 2D samples, and both are more transcriptionally similar to each other by both Pearson Correlation and Diffusion Map (Reviewer-only Revisions Figure 1, Supplemental Figure 1, respectively) than they are to the Panc1 3D model or any other cell lines described in our manuscript. The figures of this comparison are included for the reviewer’s consideration to this point that constructing duplicate libraries of each sample would have added very little to the dataset and primary findings as they are currently described in the main text.
- The reviewer very thoughtfully considers the number of cells per sample required to adequately represent a sample’s heterogeneity, and also rightly questions the use of technical replicates to accurately capture biological variation. The reviewer comments that any sample size less than 10,000 cells is inadequate for characterization purposes. However, existing literature shows as few as 500 cells per sample may be used to draw significant conclusions from scRNAseq data, so long as adequate sequencing depth per cell within a sample is achieved, and appropriate filtering and quality control analysis of the dataset is conducted [Refs. 12-14]. Data for the number of cells and sequencing depth per sample is contained in Reviewer-only Revisions Table 1 below. In our own recently published work, we also demonstrate that few cells from extremely limited tissue biopsies, not sequenced in technical replicates, were adequate to predict therapeutically-relevant inter-cellular interactions at the single-cell RNA level [Ref. 15] In Main Figure 1A, cells from each sample cluster together, a UMAP organization based on biological identity that is supported by another PDAC preclinical models paper recently published in Nature Communications [Ref. 16], that strongly suggests that batch effect does not drive the organization of cells in the scRNAseq analysis assay. To further confirm the non-existence of batch effect in a quantitative way in our scRNAseq dataset, we show in Reviewer-only Revisions Figure 1 below that within the scRNAseq datasets, the cells within each sample form their own “clades”, demonstrating that the cells are clustering as a function of biological similarity and not as a function of technical artifacts. If there was a batch effect present, we would see samples sequenced together in individual runs clustering in the same families. Instead, we see samples clustering in a way that supports our hypotheses of biological identities across the cell lines. Further, in Main Figures 2 and 3, where we dissect heterogeneity between MiaPaca2 and Panc1 samples, we show that clustering or UMAP separation at the scRNAseq level is driven by transcriptomic consequences of differential genomic events/chromatin accessibility, rather than “technical variation” within the samples themselves. Given that cell lines are inherently less heterogeneous than tissue biopsies, that each sample in this study was sequenced to a depth well above the “minimum required” read count, and that the very nature of single-cell profiling

guarantees hundreds of technical replicates within each sample, we are confident that the data presented here does truly capture the biological heterogeneity across these cell lines.

Reviewer-only Revisions Table 1. Sequencing metrics for scRNAseq data, provided in response to concerns raised by Reviewer #1.

	3' RNA			
Cell line	# cells targeted	# cells sequenced	Mean reads/cell	Median genes/cell
HPNE	10,000	12,810	12,458	1,809
HPDE	10,000	5,233	13,831	1,941
BxPc3	10,000	10,405	12,426	1,819
HPAF-II	10,000	5,104	27,166	1,989
Panc1- 3D	10,000	23,118	11,828	780
Panc1- 2D	10,000	5,559	3,407	3,407
MP2-A	10,000	9,913	6,372	1,191
MP2-B	10,000	7,171	9,811	1,663
MP2-C	10,000	11,177	6,209	1,083

Reviewer-only Revisions Figure 1. Pearson correlation analysis of cell lines to corroborate reported findings of Diffusion Space scRNAseq analysis. To address reviewer comments regarding technical replicates of samples, we show that a technical replicate of Panc-1 2D is ranked as highly similar to the other Panc-1 samples by both cladogram organization (A) and heatmap (B). This provides evidence that cell number and sequencing depth in the original presented dataset are sufficient to both ameliorate batch effects and to proscribe against generation of technical replicates for all samples.

2. Several of the transcriptome comparisons use GO term enrichment to point to potential biological differences between subclusters. However, it seems to me that in several instances the authors are using an FDR cutoff of 0.5 instead of the standard 0.05, and are thus basing conclusion on results that are not statistically significant. In all cases where this GO data is presented or discussed I think the authors should make explicitly clear which results exceed the threshold for statistical significance and refrain from overinterpreting results that do not meet this threshold (e.g. Figure 2C, 3B, and the results section “Epigenetic alterations define ...” where results in the supplemental excel file are discussed).

- The reviewer makes an excellent point concerning the statistical significance of our GSEA findings at the single-cell level. It is important to note that for the figures referred to in this comment, Hallmark Pathways, not GO, were used. As such, we have amended Main Figure 2C to include a legend with FDR cutoff 0.05, and listing only such pathways that were enriched at that level in the final figure. In Main Figure 3B, where GSEA was also included, the original legend did include cutoff values of both 0.5 and 0.05, and has been left unchanged. We recognize the confusion that might be caused by including pathways that do not have an FDR < 0.05. In the case of MP2 comparisons, there are pathways that have FDR < 0.05 for one sample but an insignificant FDR for another, an example being Oxidative Phosphorylation, where the FDR value for MP2-A is 0.05, but is insignificantly enriched in MP2-B, and insignificantly downregulated in MP2-C (Figure 2C). This comparison thus demonstrates differential enrichment of a pathway in one sample over the others, and supports our argument that “validated”/fingerprinted cell lines that are “supposed to be” the same, are, in fact, not the same at the single-cell level. We have amended the main text as such, see page 9, which now reads “MP2-A is enriched in GSEA Hallmark pathways related to P53, oxidative phosphorylation, and protein secretion. In contrast, MP2-B downregulated GSEA Hallmark interferon gamma response, and MP2-C has no statistically significant enrichment in Hallmark Pathways based on scRNAseq GSEA analysis (Figure 2C).

- The main text also describes which pathways were of significance across samples, as well as which pathways identified at the scRNAseq level also significantly correlated with genomic events detected at the scCNV level. The main text referring to those findings on a per-sample basis has also been left unaltered, and we have created Supplemental Table 2 to summarize GSEA enrichment profiles which are specific to scCNV-derived scRNAseq clones.

3. Aside from the GO analyses discussed above, I am concerned with the lack of other analyses into the biological differences between clusters, as well as the lack of independent experimental validation of these differences. Clustering analysis of single-cell data rely on arbitrary “resolution” parameters that can dramatically alter the results. Thus, in my opinion it is critical not to rely on clustering analyses to conclude that two groups of cells are biologically different, but rather, to use these analyses to help point to biological features (genes, pathways, etc) that may differ between groups of cells, and then validate these. I suggest that the authors perform additional analyses into the potential biological differences between cell clusters to include at a minimum differential gene expression analyses (not just GO enrichment) and cell cycle analyses (now readily inferred from single-cell RNA-seq data – see PMID: 32312741 for example; differences in cell cycle stage could easily confound these culture-to-culture comparisons). Moreover, findings should be validated with independent experiments. For example, differentially expressed genes between two cell culture sources can be validated by FISH at the single-cell level or at least rt-qPCR at the bulk level.

- We agree wholeheartedly with the reviewer that scRNAseq analysis-deemed clusters are insufficient to denote true biological difference within samples. The scRNAseq data described in our manuscript were analyzed according to parameters that were assessed by experts in the field of scRNAseq, whom we cite as authors (Drs. Ken Chen, Fang Wang, Roshan Sharma). We chose clustering parameters based on the principal components deemed appropriate by the Seurat analysis package. Further, the GSEA findings reported in Main Figures 2 + 3 are done so based on genes that are differentially expressed between clusters within samples, NOT based on “whole gene lists” of genes expressed on a per-cluster basis. Thus, in full agreement with the reviewer, the findings we have submitted for review are much in line with their prescribed analysis. The legends of Main Figures 2 + 3 have been amended to more clearly articulate the findings depicted, and our Methods have also been amended to more thoroughly describe how differential expression was conducted across the subclusters within samples to obtain the depicted results.
- Also in agreement with the reviewer, we believe that orthogonal validation of all reported findings is critical to the integrity of our work. For this reason, both scCNVseq and scRNAseq were conducted on cell lines as “complimentary” yet still independent datasets. Further, included in our analysis scCNVseq of cell lines are findings derived from our clonealign analysis to “link” events at the CNV level with biologically-relevant, independently-determined scRNAseq clusters. We subsequently conducted GSEA analysis on ONLY those scRNA-based clusters that correlated to genomic events in the scCNV seq data (Main Figures 2D-2I, Supplemental Figure 6E), an analysis unique to this study as very few published works have paired scCNVseq with independently-collected scRNAseq data of the same samples.
- With regards to the reviewer’s concern that Cell Cycle stages may have confounded our analyses: we have included here the results of cell cycle analysis of MP2 (Reviewer-only Revisions Figure 2B) and Panc1 (Reviewer-only Revisions Figure 2C) samples in Reviewer-only Revisions Figure 2, below, and quantified the percentage of cells per sample occupying each cell cycle phase. We do see that the clustering within a given cell line may be driven in part by cell cycle. For example, cluster 1 vs. cluster 4 in the MP2 analysis (where cluster 4 contains primarily MP2-A cells in G2M phase, while cluster 1 contains primarily of MP2-

A cells in G1 or S phase). However, cell cycle differences between cell lines are not extreme enough to explain the very large variability we see *between* cell lines. We find that a large proportion of cells are in S or G2M phase in all of these cell lines (MP2 51.5-62.6%, Panc1 42.2-51.9%).

- To further validate that events called at the scCNV level were not a function of sequencing artifact, we have conducted FISH analysis to probe for the major scCNV events reported in the Main Text, and included those data in Supplemental Figure 3.
- As additional confirmation that scCNVseq events could be validated with our independently-acquired scRNAseq data, we ran inferCNV on scRNAseq data per sample, and find a concordance between events predicted at the transcriptomic level with the actual scCNVseq groupings (Main Figure 2D, Supplementary Fig. 6E).

Reviewer-only Revisions Figure 2 (Cell Cycle Scoring). Cell Cycle Scoring Analysis of scRNAseq data across Panc-1 samples and MP2 samples. A) Table shows sample (row) by cell cycle phase (columns), numbers are

percentages of cells in each sample occupying each cycle phase. B) UMAP embedding of MP2 samples across

A.

	G1	G2M	S
MP2-A	38.6	41.3	20.2
MP2-B	37.4	41.5	21.0
MP2-C	48.5	32.8	18.7
Panc1_3D	48.1	30.6	11.6
Panc1_2D	57.8	32.7	19.2

B.

C.

clusters and cycle phases. C) Same as (B) for Panc-1 2D and 3D samples.

4. With regard to the scATAC-seq analysis in figure 3, the number of cells profiled seems very low and probably too small to draw conclusions from clustering analysis like those shown in supplemental Figure 6 where all of the cells essentially are essentially in one large “blob.” I was also unable to find quality control assessment for any of these libraries (or scRNA-seq libraries) including read counts per cell and signal/noise metrics like Transcription Start Site (TSS) enrichment. In my opinion, these QC analysis should be included in supplement if not already there somewhere.

- We agree with the reviewer that the number of cells used in snATACseq may appear quite small. However, rigorous quality control metrics were indeed run on the data, the results of which have been added to the supplemental data (Supplementary Fig. 7A, 7B), and are also attached here (Reviewer-only Revisions Figure 3) for the reviewer’s consideration. We subset nuclei according to the following criteria: TSS Enrichment > 2, nucleosome signal < 10, blacklist ratio < 0.05, percent reads in peaks > 15, peak region fragments between 1,000 and 20,000. The methods have been amended to reflect these parameters. After ensuring the quality of nuclei were sufficient, as proscribed by the Signac Package developed by the Satija Lab (cited in Main Text and Methods) as well as recent publications including this data type [Refs. 17-18], a final count of 9,785 nuclei are represented by the Panc1 2D sample, and 7,453 nuclei of Panc1 3D are depicted in the figures. As discussed above, the resulting number of nuclei is more than sufficient, by current research standards, to draw accurate conclusions and conduct correlative investigations across samples in this dataset.
- As per the scRNAseq data quality control metrics, those are included in our submission to the GEO portal and are now publicly available. It is not common practice in the literature to list quality control metrics of scRNAseq data in the supplementary or main figures, so we did not include them in this manuscript. However, cell count and sequencing depth per scRNAseq sample are listed in Reviewer-only Revisions Table 1.

Reviewer-only Revisions Figure 3. Quality control data of nuclei sequenced in snATACseq assay for Panc-1 3D (A, B), and Panc-1 2D (C, D) samples. TSS Enrichment Plots for Panc-1 3D (B) and Panc-1 2D (D) depicting the mean TSS enrichment score across nuclei are shown. Nuclei data were subsequently filtered according to appropriate QC parameters.

5. *A number of the figure panels are very hard to interpret due to illegibly small or blurry text, clutter, and/or lack of labeling (e.g. 1C,D,G,H; 2B,D,G). This detracts from the manuscript because I am unable to*

A. Panc-1 3D snATAC QC

C. Panc-1 2D snATAC QC

independently assess these data as presented, and am left to rely only on the authors description/interpretation in text. I suggest a major overhaul of these panels in any resubmission.

- The reviewer is entirely correct. All figures have been updated to a higher resolution, and currently appear legible. However, if the figures are still considered too small or unclear in their labeling, we are more than glad to oblige by making further edits.

References Cited for Reviewer #1

1. Marchesi, F., et al. (2004). "Increased survival, proliferation, and migration in metastatic human pancreatic tumor cells expressing functional CXCR4." *Cancer Res* 64(22): 8420-8427.
2. Huanwen, W., et al. (2009). "Intrinsic chemoresistance to gemcitabine is associated with constitutive and laminin-induced phosphorylation of FAK in pancreatic cancer cell lines." *Mol Cancer* 8: 125
3. Samuel, N. and T. J. Hudson (2011). "The molecular and cellular heterogeneity of pancreatic ductal adenocarcinoma." *Nat Rev Gastroenterol Hepatol* 9(2): 77-87.
4. Rathos, M. J., et al. (2012). "Molecular evidence for increased antitumor activity of gemcitabine in combination with a cyclin-dependent kinase inhibitor, P276-00 in pancreatic cancers." *J Transl Med* 10: 161.
5. Thu, K. L., et al. (2014). "SOX15 is a candidate tumor suppressor in pancreatic cancer with a potential role in Wnt/beta-catenin signaling." *Oncogene* 33(3): 279-288.
6. Deer, E. L., et al. (2010). "Phenotype and genotype of pancreatic cancer cell lines." *Pancreas* 39(4): 425-435.
7. Kim, Y., et al. (2014). "Comparative proteomic profiling of pancreatic ductal adenocarcinoma cell lines." *Mol Cells* 37(12): 888-898.
8. Fredebohm, J., et al. (2012). "Establishment and characterization of a highly tumorigenic and cancer stem cell enriched pancreatic cancer cell line as a well defined model system." *PLoS One* 7(11): e48503.

References Cited for Reviewer #2

9. Subramanian, A., et al. (2017). "A Next Generation Connectivity Map: LI000 Platform and the First 1,000,000 Profiles." *Cell* 171(6): 1437-1452 e1417.

References Cited for Reviewer #3

10. Semaan, A., et al. (2020). "Defining the Comprehensive Genomic Landscapes of Pancreatic Ductal Adenocarcinoma Using Real World Endoscopic Aspiration Samples." *Clinical Cancer Research: clincanres.2667.2020*.
11. Moffitt, R. A., et al. (2015). "Virtual microdissection identifies distinct tumor- and stroma-specific subtypes of pancreatic ductal adenocarcinoma." *Nat Genet* 47(10): 1168-1178.

References Cited for Reviewer #4

12. Mandric, I., et al. (2020). "Optimized design of single-cell RNA sequencing experiments for cell-type-specific eQTL analysis." *Nature Communications* 11(1): 5504.
13. Haque, A., et al. (2017). "A practical guide to single-cell RNA-sequencing for biomedical research and clinical applications." *Genome Med* 9(1): 75.
14. Nguyen, Q. H., et al. (2018). "Experimental Considerations for Single-Cell RNA Sequencing Approaches." *Front Cell Dev Biol* 6: 108.
15. Lee, J. J., et al. (2021). "Elucidation of tumor-stromal heterogeneity and the ligand-receptor interactome by single cell transcriptomics in real-world pancreatic cancer biopsies." *Clin Cancer Res*.

16. Krieger, T. G., et al. (2021). "Single-cell analysis of patient-derived PDAC organoids reveals cell state heterogeneity and a conserved developmental hierarchy." *Nature Communications* 12(1): 5826.
17. Li, R., et al. (2021). "A simple and robust method for simultaneous dual-omics profiling with limited numbers of cells." *Cell Rep Methods* 1(3).
18. Yan, F., et al. (2020). "From reads to insight: a hitchhiker's guide to ATAC-seq data analysis." *Genome Biol* 21(1): 22.

Reviewer-only Revisions Table 1. Sequencing metrics for scRNAseq data, provided in response to concerns raised by Reviewer #1.

	3' RNA			
Cell line	# cells targeted	# cells sequenced	Mean reads/cell	Median genes/cell
HPNE	10,000	12,810	12,458	1,809
HPDE	10,000	5,233	13,831	1,941
BxPc3	10,000	10,405	12,426	1,819
HPAF-II	10,000	5,104	27,166	1,989
Panc1- 3D	10,000	23,118	11,828	780
Panc1- 2D	10,000	5,559	3,407	3,407
MP2-A	10,000	9,913	6,372	1,191
MP2-B	10,000	7,171	9,811	1,663
MP2-C	10,000	11,177	6,209	1,083

Reviewer-only Revisions Figure 1. Pearson correlation analysis of cell lines to corroborate reported findings of Diffusion Space scRNAseq analysis. To address reviewer comments regarding technical replicates of samples, we show that a technical replicate of Panc-1 2D is ranked as highly similar to the other Panc-1 samples by both cladogram organization (A) and heatmap (B). This provides evidence that cell number and sequencing depth in the original presented dataset are sufficient to both ameliorate batch effects and to proscribe against generation of technical replicates for all samples. Figures C and D demonstrate nearly identical intersample relatedness by Pearson correlation as previously reported by Diffusion Space analysis in the Main Text.

Reviewer-only Revisions Figure 2. Cell Cycle Scoring Analysis of scRNAseq data across Panc-1 samples and MP2 samples. A) Table shows sample (row) by cell cycle phase (columns), numbers are percentages of cells in each sample occupying each cycle phase. B) UMAP embedding of MP2 samples across clusters and cycle phases. C) Same as (B) for Panc-1 2D and 3D samples.

A.

	G1	G2M	S
MP2-A	38.6	41.3	20.2
MP2-B	37.4	41.5	21.0
MP2-C	48.5	32.8	18.7
Panc1_3D	48.1	30.6	11.6
Panc1_2D	57.8	32.7	19.2

B.

C.

Reviewer-only Revisions Figure 3. Quality control data of nuclei sequenced in snATACseq assay for Panc-1 3D (A, B), and Panc-1 2D (C, D) samples. TSS Enrichment Plots for Panc-1 3D (B) and Panc-1 2D (D) depicting the mean TSS enrichment score across nuclei are shown. Nuclei data were subsequently filtered according to appropriate QC parameters.

A. Panc-1 3D snATAC QC

C. Panc-1 2D snATAC QC

C. Panc-1 3D TSS Enrichment Plot

D. Panc-1 2D TSS Enrichment Plot

Reviewer-only Revisions Figure 4. Comparative analysis of scRNA subclusters derived from Panc1 3D (A) and 2D (B) models. Using Clue.io database, genes differentially up or downregulated per cluster were correlated to chemical agents that would induce transcriptional reprogramming (red) or not (blue). Clear differences between subclusters across samples demonstrate intrasample heterogeneity, as well as spheroid reprogramming resulting in predicted sensitivity to MTOR inhibition.

A. Panc1 3D Clue.io Predictions

B. Panc1 2D Clue.io Predictions

Reviewer-only Revisions Tables 2 + 3. Gene lists and expression values across cell lines for RNA-based Moffitt PDAC Molecular Subtypes.

Moffitt Basal Gene Set Expression Values

	avg.exp	pct.exp	features.plot	id	avg.exp.scaled
AREG	0.41885346	34.2384888	AREG	PD01early	-0.707106781
CST6	0.32663247	32.1920504	CST6	PD01early	0.707106781
DHRS9	0.0306013	4.64384101	DHRS9	PD01early	-0.707106781
FAM83A	7.77524101	97.9929162	FAM83A	PD01early	0.707106781
FGFBP1	0.5395463	52.2628886	FGFBP1	PD01early	-0.707106781
GPR87	0.23653783	34.9468713	GPR87	PD01early	0.707106781
KRT15	0.35558438	44.077135	KRT15	PD01early	0.707106781
KRT17	0.00095855	0.19677292	KRT17	PD01early	-0.707106781
KRT6A	0.00148028	0.27548209	KRT6A	PD01early	-0.707106781
KRT7	4.99248259	98.2683983	KRT7	PD01early	-0.707106781
LEMD1	0.04245986	6.02125148	LEMD1	PD01early	-0.707106781
LY6D	0.61249558	45.4939	LY6D	PD01early	-0.707106781
SLC2A1	0.56566447	62.5737898	SLC2A1	PD01early	-0.707106781
SPRR1B	0.0181479	1.21999213	SPRR1B	PD01early	0.707106781
TNS4	0.18768806	23.5733963	TNS4	PD01early	0.707106781
VGLL1	0.9381014	75.2853207	VGLL1	PD01early	-0.707106781
AREG1	2.40283871	73.313897	AREG	PD01late	0.707106781
CST61	0.22494331	15.1603499	CST6	PD01late	-0.707106781
DHRS91	0.04226508	3.45966958	DHRS9	PD01late	0.707106781
FAM83A1	3.73043495	85.9280855	FAM83A	PD01late	-0.707106781
FGFBP11	1.33601402	57.7259475	FGFBP1	PD01late	0.707106781
GPR871	0.10932271	11.1175899	GPR87	PD01late	-0.707106781
KRT151	0.32626943	27.9105928	KRT15	PD01late	-0.707106781
KRT171	0.16077655	5.48104956	KRT17	PD01late	0.707106781
KRT6A1	0.00314922	0.15549077	KRT6A	PD01late	0.707106781
KRT71	9.11948105	97.0845481	KRT7	PD01late	0.707106781
LEMD11	0.13020025	11.6423712	LEMD1	PD01late	0.707106781
LY6D1	2.61900164	67.696793	LY6D	PD01late	0.707106781
SLC2A11	0.6335786	47.5218659	SLC2A1	PD01late	0.707106781
SPRR1B1	0.00936573	0.56365403	SPRR1B	PD01late	-0.707106781
TNS41	0.04630835	4.72303207	TNS4	PD01late	-0.707106781
VGLL11	1.11406856	61.0495627	VGLL1	PD01late	0.707106781

Moffitt Classical Gene Set Expression Values

	avg.exp	pct.exp	features.plot	id	avg.exp.scaled
AGR2	2.8211069	92.483274	AGR2	PD01early	-0.707107
AGR3	1.8740684	90.633609	AGR3	PD01early	-0.707107
ANXA10	0.0762617	12.081858	ANXA10	PD01early	-0.707107
BTNL8	0	0	BTNL8	PD01early	-0.707107
CDH17	0.00157	0.3148367	CDH17	PD01early	-0.707107
CEACAM6	4.5521738	89.177489	CEACAM6	PD01early	-0.707107
CLRN3	0.0009363	0.1574183	CLRN3	PD01early	-0.707107
CTSE	0.0203259	2.8728847	CTSE	PD01early	-0.707107
CYP3A7	0	0	CYP3A7	PD01early	-0.707107
FAM3D	0.0016949	0.3148367	FAM3D	PD01early	0.7071068
KRT20	0.0016988	0.1180638	KRT20	PD01early	-0.707107
LGALS4	0.5073716	47.540338	LGALS4	PD01early	-0.707107
LYZ	0.001626	0.2361275	LYZ	PD01early	-0.707107
MYO1A	0	0	MYO1A	PD01early	-0.707107
PLA2G10	0.3390262	42.030697	PLA2G10	PD01early	-0.707107
SPINK4	0.021462	3.227076	SPINK4	PD01early	-0.707107
ST6GALNAC	0.0311629	5.588351	ST6GALNAC	PD01early	0.7071068
TFF1	0.0054617	1.0232192	TFF1	PD01early	-0.707107
TFF2	1.2610681	73.396301	TFF2	PD01early	-0.707107
TFF3	0.0123643	2.3612751	TFF3	PD01early	-0.707107
TSPAN8	0.0606379	9.4844549	TSPAN8	PD01early	-0.707107
VSIG2	0.0825113	14.049587	VSIG2	PD01early	-0.707107
AGR21	5.3796299	92.225462	AGR2	PD01late	0.7071068
AGR31	2.5813036	83.945578	AGR3	PD01late	0.7071068
ANXA101	0.5133468	37.356657	ANXA10	PD01late	0.7071068
BTNL81	0.0004682	0.058309	BTNL8	PD01late	0.7071068
CDH171	0.0038318	0.505345	CDH17	PD01late	0.7071068
CEACAM61	10.601322	94.499514	CEACAM6	PD01late	0.7071068
CLRN31	0.0206115	2.1963071	CLRN3	PD01late	0.7071068
CTSE1	0.0816948	5.3838678	CTSE	PD01late	0.7071068
CYP3A71	0.0007674	0.0971817	CYP3A7	PD01late	0.7071068
FAM3D1	0.0012943	0.1749271	FAM3D	PD01late	-0.707107
KRT201	0.0134115	1.1273081	KRT20	PD01late	0.7071068
LGALS41	0.7265637	35.393586	LGALS4	PD01late	0.7071068
LYZ1	0.0391218	3.1098154	LYZ	PD01late	0.7071068
MYO1A1	0.0021196	0.1943635	MYO1A	PD01late	0.7071068
PLA2G101	0.7553368	49.757046	PLA2G10	PD01late	0.7071068
SPINK41	0.1994586	13.100097	SPINK4	PD01late	0.7071068
ST6GALNAC1	0.0256968	2.9931973	ST6GALNAC	PD01late	-0.707107
TFF11	0.2841931	24.586978	TFF1	PD01late	0.7071068
TFF21	5.5394443	90.631681	TFF2	PD01late	0.7071068
TFF31	0.0207815	1.7881438	TFF3	PD01late	0.7071068
TSPAN81	1.2002164	55.918367	TSPAN8	PD01late	0.7071068
VSIG21	0.1753376	17.55102	VSIG2	PD01late	0.7071068

Reviewer-only Revisions Table 4. Percent of genome non-diploid on a per-sample basis as derived from scCNVseq. This serves as a measure of genomic heterogeneity, as per request from Reviewer #4.

	HPAF2	MP2-C	HPNE	HPDE	BXPC3	MP2-B
CN not 2	0.628328	0.815911	0.302481	0.794937	0.746022	0.808391

Reviewer-only Revisions Figure 5. Violin plot of expression of common epithelial marker genes in PDAC cell lines, showing very low relative expression in HPNE cells.

Reviewer-only Revisions Figure 6. Heatmap annotated per cell in Early versus Late PDO culture (columns) and genes (rows) associated with Moffitt Classical and Basal molecular subtypes. PDO1 early expresses genes almost exclusively associated with the Basal Subtype, while many cells within PDO1 late express genes that are either classical or basal in nature.

REVIEWER COMMENTS

Reviewer #1 (Remarks to the Author):

This reviewer appreciates the authors for responding to the comments and issues raised about their original submission and thank them for the additional clarifications and data provided. A key issue remains the lack of “phenotypic” validation. The absence of a “phenotypic” characterization of the occult heterogeneity makes it hard to appreciate the translational directions that those heterogeneities unravel. In practice, most cell culture studies go beyond two passages – typically upwards of 10-15, and it is widely accepted that cell lines undergo changes as they are passaged hence the need to limit passage number. Granted, this study leverages on scRNAseq and scCNVseq to pinpoint what those “changes” might be a single cell level and on that basis is quite innovative.

That said, as the authors rightly alluded to, it is hard to account for the number of passages that cells obtained from external sources have already undergone. Although the authors controlled for passage number in their own study, one cannot rule out that any additional heterogeneities the authors observed may be due to preexisting alterations beyond the authors’ control. Therefore, it is a bit difficult to visualize the context in which knowing that heterogeneities arose after two passages would matter if there is no accompanying phenotype.

In sum, other than the concern on phenotypic link, the study is both innovative and interesting.

Reviewer #2 (Remarks to the Author):

I am happy with the authors detailed responses to the review comments. All queries were met with sufficient detail and I have no further comments to make on the study.

Reviewer #3 (Remarks to the Author):

The authors have done an impressive round of revisions and have addressed all my concerns.

Reviewer #4 (Remarks to the Author):

The authors have not fundamentally addressed my concerns. I reiterate the most serious concerns below:

- Lack of validation. None of the inferences from scRNA-seq are validated with independent methods, even though methods are readily available for validating differentially expressed genes between cell clones/populations (e.g. RT-qPCR), even at the single-cell level (e.g. RNA FISH). In their rebuttal the authors argue that that their clustering can't be due to technical or sampling artifacts because they find differential gene enrichments by GSEA as well as differences using computationally inferred CNVs. However, this is a circular argument – both GSEA and inferCNV are analysis run on the same underlying data scRNA data that has not been reproduced.

With regard to GSEA (e.g. 2c) – the authors do not provide orthogonal validation that any of the genes in the analysis are actually differentially expressed in their cell cultures. They also do not indicate which enrichments exceed a significance level of $FDR < 0.05$ (do any?). Are those FDR values adjusted for multiple testing across many samples (rather than just within each sample)? For example, if FDR is calculated for each sample independently, the significance level should be adjusted even lower when analyzing 3 samples together as in 2C. The authors do not even provide quantitative analysis or heatmaps of the expression values of the genes in those sets in each sample to evaluate the magnitude of expression differences. GSEA are notorious for spurious associations. The authors also seem to perform GSEA using a derived and unconventional metric where they multiply the $-\log_{10}p$ value from “FindMarkers” function in Seurat by the fold-change directions. However, the methods section does not adequately describe what was actually done here. Was this derived metric defined for the same superset of genes for all three samples? – this would be necessary to enable a fair pathway GSEA comparison across samples. Also, FindMarkers and fold-change calculations require definition of a foreground and background set of cells – how were these determined with 3 cell clones? 1 vs others? 1 vs 2? Etc? These critical details are unclear, but in any case, GSEA analysis are not validation of the underlying scRNA data (they are derived from it) and do not address my concern.

Re inferCNV – these data seem also to be derived from the scRNA-seq data. The authors claim in their rebuttal that “we ran inferCNV on scRNAseq data per sample, and find a concordance between events predicted at the transcriptomic level with the actual scCNVseq groupings (Main Figure 2D, Supplementary Fig. 6E).” The authors emphasize this as an “orthogonal” validation of their data.

However, Figure 2D and S6E are simply descriptive heatmaps that do not justify the claims made in text. There are no quantitative comparisons that I can find of these predictions to the actual measured scCNVs. This is a glaring omission. In any case, analysis of the inferCNV data, though potentially informative if performed in more depth than done here, would not be an orthogonal validation of scRNA-seq-derived conclusions. Most of the expression differences driving the clustering and GSEA are not due to genes in CNVs (or at least have not been shown to be such), and of course CNV analysis simply does not measure expression. The analysis of inferCNV to the measured CNV would simply validate that the “inferCNV” algorithm can accurately identify CNVs, at least in some cases. That is a fundamentally different type of validation that what is needed to validate transcriptional differences between cell cultures.

- **Reproducibility.** The authors are basing wide-ranging conclusions on single replicates of a technology in single-cell RNA-seq that is well known to suffer from technical noise, sampling limitations, dropout events, etc. Indeed, their own Reviewer-only Table 1 shows that they detect <2000 genes / cell, which is certainly a vast under-sampling of the genes expressed in a given cell, much less the quantitative range of transcripts expressed for each gene. I understand that single-cell experiments are expensive, but when basing conclusions about underlying biology on single-cell data (particularly with no independent experimental validation as per concern #1), replicates are strictly necessary to establish the reproducibility and thus validity of the conclusions. The fact that libraries were sequenced in parallel does not address this concern. It is well known that batch effects and other technical artifacts can have dramatic effects on the final clustering and other results of single-cell data analysis (<https://pubmed.ncbi.nlm.nih.gov/29608179/>). Indeed, there are myriad methods designed specifically to mitigate this problem by optimizing sample integration (e.g. Harmony, CCA / integration anchors in Seurat, etc; Recently benchmarked in <https://pubmed.ncbi.nlm.nih.gov/34949812/>). Have any of these methods been used to integrate samples? Without such integration methods, it is no surprise that UMAP dimensionality and graph-based clustering methods show differences between samples. And even with such integration, replicates and validation are essential. The need for experimental replicates (and validation) in differential gene/feature analysis for ‘omics data (e.g. RNA-seq, ATAC-seq, ChIP-seq, etc) has been well established for decades. It is even more important for sparsely sampled single-cell data with even higher risk of drawing spurious conclusions.

In this second revisions document for the manuscript entitled “Occult polyclonality of preclinical pancreatic cancer models drives in vitro evolution”, we are addressing all reviewer comments as requested in recent communications with the Nature Communications editor. We hope that the reviewers will consider our responses to be satisfactory.

Reviewer #1 (Remarks to the Author):

This reviewer appreciates the authors for responding to the comments and issues raised about their original submission and thank them for the additional clarifications and data provided. A key issue remains the lack of “phenotypic” validation. The absence of a “phenotypic” characterization of the occult heterogeneity makes it hard to appreciate the translational directions that those heterogeneities unravel. In practice, most cell culture studies go beyond two passages – typically upwards of 10-15, and it is widely accepted that cell lines undergo changes as they are passaged hence the need to limit passage number. Granted, this study leverages on scRNAseq and scCNVseq to pinpoint what those “changes” might be a single cell level and on that basis is quite innovative.

That said, as the authors rightly alluded to, it is hard to account for the number of passages that cells obtained from external sources have already undergone. Although the authors controlled for passage number in their own study, one cannot rule out that any additional heterogeneities the authors observed may be due to preexisting alterations beyond the authors’ control. Therefore, it is a bit difficult to visualize the context in which knowing that heterogeneities arose after two passages would matter if there is no accompanying phenotype.

In sum, other than the concern on phenotypic link, the study is both innovative and interesting.

Thank you so very much for your help and support in the revision of this manuscript!

Reviewer #2 (Remarks to the Author):

I am happy with the authors detailed responses to the review comments. All queries were met with sufficient detail and I have no further comments to make on the study.

Excellent! Thank you very much! We are so appreciative of your feedback!

Reviewer #3 (Remarks to the Author):

The authors have done an impressive round of revisions and have addressed all my concerns.

Happy to hear! Thank you for your gracious help and support in the publication of this study!

Reviewer #4 (Remarks to the Author)

The authors have not fundamentally addressed my concerns. I reiterate the most serious concerns below:

- Lack of validation. None of the inferences from scRNA-seq are validated with independent methods, even though methods are readily available for validating differentially expressed genes between cell clones/populations (e.g. RT-qPCR), even at the single-cell level (e.g. RNA FISH). In their rebuttal the authors*

argue that their clustering can't be due to technical or sampling artifacts because they find differential gene enrichments by GSEA as well as differences using computationally inferred CNVs. However, this is a circular argument – both GSEA and inferCNV are analysis run on the same underlying data scRNA data that has not been reproduced.

- This manuscript described multiple data types that are used to validate our findings across cell lines. Each dataset was generated in an independent experiment. scCNVseq libraries, which sequence the DNA of sampled cells were constructed from the same passage as cells used for scRNAseq, Whole Genome Sequencing (Bulk DNA) and snATACseq, where appropriate. For FISH validation, as part of this review process, another pool of cells for each sample was used. In this way, we have conducted at minimum 3 independent experiments for data acquisition per sample. For all of the cell lines, at the very least, we have conducted independent scCNV and scRNAseq experiments, across which we describe consistent findings to link independently sequenced genomic events with RNA expression gathered from an independent scRNAseq experiment. In the cases of HPDE and HPNE, we conducted FISH and Whole Exome Sequencing (WES) to confirm the most prominent and unexpected events which had been detected in the scCNVseq experiment (Figure 1, bottom). In the case of our Panc1 comparison, we paired scRNAseq with snATACseq experiments, and demonstrate how transcriptional differences between 2D and 3D models are due to epigenetic modifications. In the example of Panc1, scRNAseq experiments are used to orthogonally validate motif enrichment identified at the chromatin level in the snATACseq assay.
- A batch effect occurs when samples are sequenced in different batches, collected at different times, or processed by different library preparations that preclude a reliable analysis from being conducted (<https://sysbiowiki.soe.ucsc.edu/node/323>). All samples for appropriate comparisons were sequenced together, at the same time, on the same instrument. All libraries were built using the same library preparation kits, at the same time. In our first rebuttal, we included a technical replicate of Panc1-2D and conducted an analysis that was not a Seurat-based scRNA method (utilized a Pearson correlation of scRNAseq data, as the Reviewer had requested). We have shown that the cluster organization of the samples with respect to each other remains the same. This analysis provides clear evidence that any existing batch effect was not only ameliorated, but was not the culprit of the findings reported here. We have included our initial response for the reviewer here, so that they may reconsider.
 - From Response 1: “In our own recently published work, we also demonstrate that few cells from extremely limited tissue biopsies, not sequenced in technical replicates, were adequate to predict therapeutically-relevant inter-cellular interactions at the single-cell RNA level [Ref. 4] In Main Figure 1A, cells from each sample cluster together, a UMAP organization based on biological identity that is supported by another PDAC preclinical models paper recently published in Nature Communications [Ref. 5], that strongly suggests that batch effect does not drive the organization of cells in the scRNAseq analysis assay. To further confirm the non-existence of batch effect in a quantitative way in our scRNAseq dataset, we show in Reviewer-only Revisions Figure 1 below that within the scRNAseq datasets, the cells within each sample form their own “clades”, demonstrating that the cells are clustering as a function of biological similarity and not as a function of technical artifacts. If there was a batch effect present, we would see samples sequenced together in individual runs clustering in the same families. Instead, we see samples clustering in a way that supports our hypotheses of biological identities across the cell lines. Further, in Main Figures 2 and 3, where we dissect heterogeneity between MiaPaca2 and Panc1 samples, we show that clustering or UMAP separation at the scRNAseq level is driven by transcriptomic consequences of differential genomic

events/chromatin accessibility, rather than “technical variation” within the samples themselves. Given that cell lines are inherently less heterogeneous than tissue biopsies, that each sample in this study was sequenced to a depth well above the “minimum required” read count, and that the very nature of single-cell profiling guarantees hundreds of technical replicates within each sample, we are confident that the data presented here does truly capture the biological heterogeneity across these cell lines.”

Reviewer-only Revisions Figure 1. Pearson correlation analysis of cell lines to corroborate reported findings of Diffusion Space scRNAseq analysis. To address reviewer comments regarding technical replicates of samples, we show that a technical replicate of Panc-1 2D is ranked as highly similar to the other Panc-1 samples by both cladogram organization (A) and heatmap (B). This provides evidence that cell number and sequencing depth in the original presented dataset are sufficient to both ameliorate batch effects and to proscribe against generation of technical replicates for all samples.

With regard to GSEA (e.g. 2c) – the authors do not provide orthogonal validation that any of the genes in the analysis are actually differentially expressed in their cell cultures. They also do not indicate which enrichments exceed a significance level of $FDR < 0.05$ (do any?). Are those FDR values adjusted for multiple testing across many samples (rather than just within each sample)? For example, if FDR is calculated for each sample independently, the significance level should be adjusted even lower when analyzing 3 samples together as in 2C. The authors do not even provide quantitative analysis or heatmaps of the expression values of the genes in those sets in each sample to evaluate the magnitude of expression differences. GSEA are notorious for spurious associations. The authors also seem to perform GSEA using a derived and unconventional metric where they multiply the $-\log_{10}p\text{value}$ from “FindMarkers” function in Seurat by the fold-change directions. However, the methods section does not adequately describe what was actually done here. Was this derived metric defined for the same superset of genes for all three samples? – this would be necessary to enable a fair pathway GSEA comparison across samples. Also, FindMarkers and fold-change calculations require definition of a foreground and background set of cells – how were these determined with 3 cell clones? 1 vs others? 1 vs 2? Etc? These critical details are unclear, but in any case, GSEA analysis are not validation of the underlying scRNA data (they are derived from it) and do not address my concern.

- We have made an adjustment to the Methods to more clearly describe what was done, with respect to GSEA analysis. The point of conducting GSEA analysis in this manuscript was, broadly, to understand what pathways might emerge from those genes which were **differentially enriched in one sample versus another**. To do this, we analyzed groups of samples (either Panc1 2D versus 3D, and then all 3 MiaPaca2 samples) as merged `seurat` objects, and identified which genes were significantly differentially expressed in one sample over the other (`FindMarkers`, `object = mp2`, `ident.1 = "MP2.A"`, `ident.2 = c("MP2.C", "MP2.B")`); `panc3_c2.markers <- FindMarkers(object = panc1, ident.1 = 2, ident.2 = c(0,1,3,4,5,6))` for Panc1 cluster-specific markers), and then ran GSEA using the `fgsea` package (Korotkevich 2019, <https://github.com/ctlab/fgsea>) on those genes which were significantly differentially expressed, according to the formatting specifications of that analysis package. This is what we have reported in Figures 2, 3, and Supplementary Figure 6. This is not a “convoluted” GSEA, but is a very commonplace analysis and is widely used in the field. We have provided code snippets at the bottom of this document to highlight the differences in these analyses, and have also submitted our source code for the Reviewer’s convenience.
- To specify the difference in marker extraction that was used for the Panc1 versus the MiaPaca2 analysis, the methods have been amended. A screenshot of our re-submitted Methods is included below for the Reviewer’s convenience. We hope that this addresses the reviewer’s concern.

Gene Set Enrichment Analysis (GSEA) for MiaPaca2 clusters

To obtain significantly enriched pathways on a per-cluster basis from MP2 cell lines, marker genes were extracted using `FindMarkers` function in `Seurat`, on a per-sample basis (`FindMarkers`, `ident.1 = "MP2.A"`, `ident.2 = c("MP2.C", "MP2.B")`). Resulting genes were ranked by $-\log_{10}(\text{FDR})$ multiplied by fold-change

directions, and the output file was formatted according to requirements for downstream input to GSEA. This yielded an output wherein upregulated genes had positive values and downregulated genes had negative values. Pre-ranked GSEA was thus performed using `fgsea` R package v1.12.0 (Korotkevich 2019) specifying 10,000 permutations, and `fgsea` output was filtered based on FDR and adjusted p-values for each pathway.

Copy number inference of MiaPaca2 cell lines from scRNA-seq

Copy number alterations from scRNA-seq were inferred using `inferCNV` R package v1.0.4 (<https://github.com/broadinstitute/inferCNV>). Raw gene counts from filtered cells were used, and cutoff was set to 0.1. HPNE cells were used as reference.

Gene Set Enrichment Analysis (GSEA) for Panc1 sample-specific clusters

To obtain significantly enriched pathways for clusters specific to either Panc1-2D or Panc1-3D, marker genes were extracted on a per-cluster basis using `FindMarkers` function in `Seurat` from a merged Panc1-2D + Panc1-3D object (as an example, `panc3_c2.markers <- FindMarkers(object = panc1, ident.1 = 2, ident.2 = c(0,1,3,4,5,6))`). Resulting genes were ranked by $-\log_{10}(\text{FDR})$ multiplied by fold-change directions. This yielded an output wherein upregulated genes had positive values and downregulated genes had negative values. Pre-ranked GSEA was thus performed using `fgsea` R package v1.12.0 (Korotkevich 2019) specifying 10,000 permutations, and `fgsea` output was filtered based on FDR and adjusted p-values for each pathway, as previously described for MiaPaca2 GSEA analysis.

Re inferCNV – these data seem also to be derived from the scRNA-seq data. The authors claim in their rebuttal that “we ran inferCNV on scRNAseq data per sample, and find a concordance between events predicted at the transcriptomic level with the actual scCNVseq groupings (Main Figure 2D, Supplementary Fig. 6E).” The authors emphasize this as an “orthogonal” validation of their data. However, Figure 2D and S6E are simply descriptive heatmaps that do not justify the claims made in text. There are no quantitative comparisons that I can find of these predictions to the actual measured scCNVs. This is a glaring omission. In any case, analysis of the inferCNV data, though potentially informative if performed in more depth than done here, would not be an orthogonal validation of scRNA-seq-derived conclusions. Most of the expression differences driving the clustering and GSEA are not due to genes in CNVs (or at least have not been shown to be such), and of course CNV analysis simply does not measure expression. The analysis of inferCNV to the measured CNV would simply validate that the “inferCNV” algorithm can accurately identify CNVs, at least in some cases. That is a fundamentally different type of validation that what is needed to validate transcriptional differences between cell cultures.

- We agree that GSEA is not validation of scRNAseq data, GSEA results are an analysis product of scRNAseq analysis methods. This is why we use GSEA as a tool to help interpret our paired scRNAseq data, snATACseq data, and scCNVseq data.
- inferCNV is derived from an scRNAseq experiment, but this software does not allow for the measurement of absolute copy numbers, only relative copy number as based on expression of genes at the RNA level. Therefore, if anything, inferCNV is used to provide a transcriptionally-relevant framework for genomic events that may be defined at the actual CNV level, by an scCNVseq experiment or a Whole Genome Sequencing assay (WES). Thus, it is not possible for an “absolute” quantification of events from inferCNV to the scCNV experiment, as inferCNV does not provide this information. Finding relative concordance between those independently-generated datasets is the state of the field at this time. Certain algorithms have arisen to provide a better genome-transcriptome linkage, such as clonealign, which we implemented for MiaPaca2 (Figure 2) and Panc1 (Supplementary Figure 6) samples, which begin to allow us to use the scRNAseq assay to orthogonally confirm amplification and deletion events in the scCNVseq experiment, and vice-versa. We subsequently conducted GSEA analysis on ONLY those scRNA-based clusters that correlated to genomic events in the scCNVseq experiment (Main Figures 2D-2I, Supplemental Figure 6E). **In that analysis, we do indeed show that scRNAseq clustering is driven by genomic events in Panc1 and MiaPaca2 samples.** This study is not a deep characterization of all PDAC cell lines, as that would not address the point of the manuscript.

• Reproducibility. The authors are basing wide-ranging conclusions on single replicates of a technology in single-cell RNA-seq that is well known to suffer from technical noise, sampling limitations, dropout events, etc. Indeed, their own Reviewer-only Table 1 shows that they detect <2000 genes / cell, which is certainly a vast under-sampling of the genes expressed in a given cell, much less the quantitative range of transcripts expressed for each gene. I understand that single-cell experiments are expensive, but when basing conclusions about underlying biology on single-cell data (particularly with no independent experimental validation as per concern #1), replicates are strictly necessary to establish the reproducibility and thus validity of the conclusions. The fact that libraries were sequenced in parallel does not address this concern. It is well known that batch effects and other technical artifacts can have dramatic effects on the final clustering and other results of single-cell data analysis (<https://pubmed.ncbi.nlm.nih.gov/29608179/>).

- Please see above explanation addressing batch effects and technical variation in this specific study. Further, the reviewer provides a publication as an explanation of their concerns. However, in our initial rebuttal, we addressed this by explaining that the field has indeed moved quickly to ensure that batch effects are not plaguing our studies, and new methods to improve sample collection and data analysis

have not only been innovated on, but were implemented here and described for the reviewer in the initial rebuttal.

- "... existing literature shows as few as 500 cells per sample may be used to draw significant conclusions from scRNAseq data, so long as adequate sequencing depth per cell within a sample is achieved, and appropriate filtering and quality control analysis of the dataset is conducted [Refs. 1-3]."
- Indeed, it is not at all the convention in the current literature to sequence technical replicates of all libraries for single cell studies. Of many landmark papers in PDAC (including organoid models, cell lines, mouse models, and human tissue studies) and other pathologies published within the last 2 years, not one demonstrated library construction of technical replicates in any of their experimental designs, and followed scRNAseq quality control methods highly similar to what we've demonstrated here [Refs. 5-12]. By sequencing technical replicates across multiple sequencing runs (due to sequencer capacity limits that very much exist) and integrating them into this analysis, we would also likely be introducing a batch effect to the data. In great deference to the reviewer, we will not be repeating all of our sequencing and analysis, especially when we've demonstrated (in Reviewer-only Figure 1) that inclusion of a technical replicate does not fundamentally change the findings we are reporting. We hope that this explanation and data inclusion is satisfactory to the reviewer.

Indeed, there are myriad methods designed specifically to mitigate this problem by optimizing sample integration (e.g. Harmony, CCA / integration anchors in Seurat, etc; Recently benchmarked in <https://pubmed.ncbi.nlm.nih.gov/34949812/>). Have any of these methods been used to integrate samples? Without such integration methods, it is no surprise that UMAP dimensionality and graph-based clustering methods show differences between samples. And even with such integration, replicates and validation are essential. The need for experimental replicates (and validation) in differential gene/feature analysis for 'omics data (e.g. RNA-seq, ATAC-seq, ChIP-seq, etc) has been well established for decades. It is even more important for sparsely sampled single-cell data with even higher risk of drawing spurious conclusions.

- We do not disagree that integration can be an extremely valuable tool in an analysis, and that batch correction is absolutely essential in datasets where a batch effect exists.
- We would like to point out that in our scRNAseq + snATACseq analysis of Panc1 samples, both datasets were co-embedded using seurat-based integration method (Main Figures 3C-3G), as it was essential to compare gene expression across those data modalities. We can only assume that the reviewer is therefore questioning our use of Seurat merge in the UMAP depicted in Main Figure 1, and we will focus our explanation on that analysis here.
- Seurat integrate and Harmony are both notorious for being "too heavy handed" in their assumptions for identifying anchors. As such, these algorithms have garnered criticism in the field [Refs. 13-17] and newer methods have been published to address those problems and understand the true nuances of biological diversity in the single cell assay [Ref. 13].
 - As an explanation, stated in the text of Ref. 13: "Moreover, these methods [LIGER, Seurat v3, and Harmony] did not explicitly distinguish technical variation from biological variation when aligning multiple single-cell datasets, which might mitigate biological variation as well when removing technical variation. In such case, these methods might favor the removal of batch effects over conservation of biological variation, leading to failure in detecting cell populations that only exist in one biological condition."

- The point of this study is not to provide an integrated analysis of all of the PDAC cell lines and describe them with respect to one another - this has been done in the literature and those studies have been cited in our main text. Save for the normal cell lines, HPDE and HPNE, all of these lines are canonically representative and PDAC epithelial cells. All of the cell lines were sequenced in the same batch, and the “normal” assumptions Seurat Integrate would make to look to rectify possible batch effects becomes inappropriate and also bioinformatically problematic. The nuances between these biological conditions do not lend well to Integrative algorithms, as explained by studies referenced above [Refs. 13-17].
- As proof of concept, in the setting of a Seurat-based integration analysis, these PDAC cell line samples are forced into forming one UMAP blob, as the Integrate algorithm is designed to group based upon technical variation under the assumption of a batch-correction method (Reviewer-only Revisions 2 Figure 2a). Given that these samples were all sequenced and processed together, technical variation between them is nearly non-existent, thus causing Integrate to falsely group the samples together: Integrate is unable to recognize that these samples were all from the same batch to begin with.
 - Given what is well-established about the underlying biology of each of these cell lines, the bioinformatic output of a “normal” pancreatic cell line having overlap with cell lines harboring both Wild Type KRAS (BxPC3), Mutant KRASG12D (Panc1 2D, 3D), and Mutant KRASG12C (MiaPaca2), as a simple example, would lead to erroneous conclusions about the transcriptional profiles of models.
- To further address the reviewer’s concern, we have included here an integrated analysis using scMC [Ref. 13] in Reviewer-only Revisions 2 Figure 2b. As explained above, this algorithm was developed after criticism of Seurat Integrate in the field, and works to mitigate biological variation over technical variation. With scMC, we show that the overall cluster structure (which samples group together) is preserved from what we report in Main Figure 1. We will not be including these analyses in our manuscript, as the goal of our paper is NOT to conduct a multi-modal comparison of scRNAseq integration methods.
- We would like to emphasize to the reviewer that we are interested in, for this study, are the differences across individual experimental conditions, such as 2D versus 3D (where we provide both snATACseq and scCNVseq data, along with scCNVseq from the parental Panc1 population), changes in laboratory geography (MiaPaca2, where we also provide scCNVseq to compliment scRNAseq), and the assumptions of “normal control” cell lines (where we have paired scRNAseq, scCNVseq, and FISH). We do not wish to deeply characterize all PDAC epithelial lines as individual entities, and are not interested in describing how similar Panc1 is to HPAF2 is to BXPC3 is to MiaPaca2, etc., and how those similarities/differences change as a function of integration analysis methods.

Reviewer-only Revisions 2 Figure 2. A) Seurat Integrate-based UMAP, showing a nonsensical clustering of scRNAseq from all 9 samples. B) scMC Integration-based UMAP, wherein sample organization based on a biologically-informed algorithm shows sample grouping identical to what is depicted in Main Figure 1A.

References Cited for Reviewer #4

1. Mandric, I., et al. (2020). "Optimized design of single-cell RNA sequencing experiments for cell-type-specific eQTL analysis." *Nature Communications* 11(1): 5504.
2. Haque, A., et al. (2017). "A practical guide to single-cell RNA-sequencing for biomedical research and clinical applications." *Genome Med* 9(1): 75.
3. Nguyen, Q. H., et al. (2018). "Experimental Considerations for Single-Cell RNA Sequencing Approaches." *Front Cell Dev Biol* 6: 108.
4. Lee, J. J., et al. (2021). "Elucidation of tumor-stromal heterogeneity and the ligand-receptor interactome by single cell transcriptomics in real-world pancreatic cancer biopsies." *Clin Cancer Res*.
5. Wang, Y., et al. (2021). "Single-cell analysis of pancreatic ductal adenocarcinoma identifies a novel fibroblast subtype associated with poor prognosis but better immunotherapy response." *Cell Discov* 7(1): 36.

6. Krieger, T. G., et al. (2021). "Single-cell analysis of patient-derived PDAC organoids reveals cell state heterogeneity and a conserved developmental hierarchy." *Nat Commun* 12(1): 5826.
7. Hosein, A. N., et al. (2021). "Loss of Rnf43 accelerates Kras-mediated neoplasia and remodels the tumor immune microenvironment in pancreatic adenocarcinoma." *Gastroenterology*.
8. Lin, W., et al. (2020). "Single-cell transcriptome analysis of tumor and stromal compartments of pancreatic ductal adenocarcinoma primary tumors and metastatic lesions." *Genome Med* 12(1): 80.
9. Bernard, V., et al. (2019). "Single-Cell Transcriptomics of Pancreatic Cancer Precursors Demonstrates Epithelial and Microenvironmental Heterogeneity as an Early Event in Neoplastic Progression." *Clin Cancer Res* 25(7): 2194-2205.
10. Chen, K., et al. (2021). "Single-cell RNA-seq reveals dynamic change in tumor microenvironment during pancreatic ductal adenocarcinoma malignant progression." *EBioMedicine* 66: 103315.
11. Hao, Y., et al. (2021). "Integrated analysis of multimodal single-cell data." *Cell* 184(13): 3573-3587 e3529.
12. Steele, N. G., et al. (2020). "Multimodal Mapping of the Tumor and Peripheral Blood Immune Landscape in Human Pancreatic Cancer." *Nat Cancer* 1(11): 1097-1112.
13. Zhang, L. and Q. Nie (2021). "scMC learns biological variation through the alignment of multiple single-cell genomics datasets." *Genome Biol* 22(1): 10.
14. Tran, H. T. N., et al. (2020). "A benchmark of batch-effect correction methods for single-cell RNA sequencing data." *Genome Biol* 21(1): 12.
15. Chazarra-Gil, R., et al. (2021). "Flexible comparison of batch correction methods for single-cell RNA-seq using BatchBench." *Nucleic Acids Res* 49(7): e42.
16. Traag, V. A., et al. (2019). "From Louvain to Leiden: guaranteeing well-connected communities." *Sci Rep* 9(1): 5233.
17. Luecken, M. D., et al. (2022). "Benchmarking atlas-level data integration in single-cell genomics." *Nat Methods* 19(1): 41-50.

Code Snippets Corresponding to MiaPaca2 and Panc1 GSEA analysis from Figures 2 and 3 in the Main text.

```
MP2.A[["percent.mt"]] <- PercentageFeatureSet(object = MP2.A, pattern = "^MT-")
plot1 <- FeatureScatter(object = MP2.A, feature1 = "nCount_RNA", feature2 = "percent.mt")
plot2 <- FeatureScatter(object = MP2.A, feature1 = "nCount_RNA", feature2 = "nFeature_RNA")
CombinePlots(plots = list(plot1, plot2))
MP2.A <- subset(x = MP2.A, subset = percent.mt < 20)

#MP2.B <- CreateSeuratObject(counts = MP2.B.data, project = "MP2.B", min.cells = 3, min.features = 200)
MP2.B[["percent.mt"]] <- PercentageFeatureSet(object = MP2.B, pattern = "^MT-")
plot1 <- FeatureScatter(object = MP2.B, feature1 = "nCount_RNA", feature2 = "percent.mt")
plot2 <- FeatureScatter(object = MP2.B, feature1 = "nCount_RNA", feature2 = "nFeature_RNA")
CombinePlots(plots = list(plot1, plot2))
MP2.B <- subset(x = MP2.B, subset = percent.mt < 20)

MP2.C[["percent.mt"]] <- PercentageFeatureSet(object = lab, pattern = "^MT-")
plot1 <- FeatureScatter(object = lab, feature1 = "nCount_RNA", feature2 = "percent.mt")
plot2 <- FeatureScatter(object = lab, feature1 = "nCount_RNA", feature2 = "nFeature_RNA")
CombinePlots(plots = list(plot1, plot2))
MP2.C <- subset(x = MP2.C, subset = percent.mt < 20)

norms <- merge(MP2.A, y = c(MP2.C, MP2.B), project = "mp2_merged")
norms <- NormalizeData(object = norms, scale.factor = 10000, verbose = F)
norms <- FindVariableFeatures(object = norms, selection.method = "vst", nfeatures = 2000)
norms_genes <- rownames(x = norms)
norms <- ScaleData(object = norms, verbose = FALSE)
norms <- RunPCA(object = norms, npcs = 30, verbose = FALSE)

# t-SNE and Clustering
#norms <- RunTSNE(object = norms, reduction = "pca", dims=1:10)
norms <- RunUMAP(object = norms, reduction = "pca", dims = 1:10)
norms <- FindNeighbors(object = norms, reduction = "pca", dims = 1:10)
norms <- FindClusters(norms, resolution = 0.5)
merged_mp2 <- DimPlot(object = norms, reduction = "umap", group.by = "orig.ident", pt.size = 0.5)
mp2_cluster <- DimPlot(object = norms, reduction = "umap", pt.size = 0.5, lable = T)
by_samp <- DimPlot(object = norms, reduction = "umap", split.by = "orig.ident")
```

```
406 #subclustering
407
408 clust9 markers <- FindMarkers(norms, ident.1 = 9, ident.2 = c(0,1,2,3,4,5,6,7,8), min.pct = 0.25)
409 write.csv(clust9_markers, "mp2_overlap_cluster.csv")
410
411 #cluster 3 is unique to MP2.B and off on the bottom right hand corner of the plot
412 clust3 markers <- FindMarkers(norms, ident.1 = 3, ident.2 = c(0,1,2,9,4,5,6,7,8), min.pct = 0.25)
413 write.csv(clust3_markers, "MP2.B_singleton_cluster.csv")
414
415 MP2.A_markers <- FindMarkers(norms, ident.1 = c(2,4, 6 ), min.pct = 0.25)
416 MP2.B_markers <- FindMarkers(norms, ident.1 = c(1,3), min.pct = 0.25)
417 MP2.C_markers <- FindMarkers(norms, ident.1 = c(0,5,7,8,9), min.pct = 0.25)
418
419 write.csv(MP2.A_markers, "MP2.A_markers.csv")
420 write.csv(MP2.B_markers, "MP2.B_markers.csv")
421 write.csv(MP2.C_markers, "MP2.C_markers.csv")
422
423 #running fgsea on individual subsets basted on cluster proportion from the stacked bar plot made above
424 library(fgsea)
425 library(ggplot2)
426
427 hallmark <- gmtPathways("/Users/maitralab/Downloads/h.all.v7.1.symbols.gmt")
428 go <- gmtPathways("/Users/maitralab/Downloads/c5.all.v7.1.symbols.gmt")
429 kegg <- gmtPathways("/Users/maitralab/Downloads/c2.cp.kegg.v7.1.symbols.gmt")
430
```

```

473 #MP2.A
474 testc5 = MP2.A_markers
475 testc5 <- as.data.frame(testc5)
476 testc5$y=(-log10(testc5$p_val_adj))
477 testc5$k=is.infinite(testc5$y)
478
479 d5 = testc5
480 d5$y[d5$k]=runif(sum(d5$k), max(d5$y[!d5$k]), max(d5$y[!d5$k])*1.25)
481
482 #next is changing the y value to negative if the -log value is, in fact, negative
483 d5$y = ifelse(d5$avg_logFC<0, -d5$y, d5$y)
484
485 #order the d5 by the y
486 d5 = d5[order(d5$y), ]
487
488 #convert the d5 to a matrix
489 mat5 = as.matrix(d5)
490
491 stats5 = mat5[,6] #must read "Named number" when done
492
493 c5_fgseaRes <- fgsea(pathways = hallmark,
494                     stats = stats5,
495                     minSize=5,
496                     maxSize=Inf,
497                     nperm=10000)
498
499
500 c5_fgseaRes_kegg <- fgsea(pathways = kegg,
501                          stats = stats5,
502                          minSize=5,
503                          maxSize=Inf,
504                          nperm=10000)
505
506 c5_fgseaRes_go <- fgsea(pathways = go,
507                        stats = stats5,
508                        minSize=5,
509                        maxSize=Inf,
510                        nperm=10000)

```

```

#made a merged file of all the samples labeled by orig.ident and pathways

ggplot(gsea_mp2_combined, aes(orig.ident, pathway)) + # change replacement to pathway prn
  geom_point(aes(size = -log10(padj), col = NES)) +
  scale_color_gradientn(colours = c("blue2", "blue", "white", "red", "red2"), na.value = "grey80", limits = c(-1,1)*max(abs(gsea_mp2_combined$NES))) +
  scale_size(name = "FDR",
             breaks = c(-log10(0.005), -log10(0.05), -log10(0.5)),
             labels = c(0.005, 0.05, 0.5)) +
  theme(panel.border=element_blank(),
        panel.background=element_rect(fill="white", color="black", size=0.5, linetype="solid"),
        panel.grid.major=element_blank(),
        panel.grid.minor=element_blank(),
        # plot.title=element_text(size=rel(1.25), hjust=0.5),
        axis.line=element_blank(),
        axis.text=element_text(color="black", size=rel(.75)),
        axis.text.x=element_text(angle=90, vjust=.5),
        axis.title.x=element_text(color="black", size=rel(1), angle=0),
        axis.title.y=element_text(color="black", size=rel(1), angle=90),
        axis.ticks.length=unit(0.1, "cm"),
        axis.ticks=element_line(color="black"),
        legend.key = element_blank(),
        aspect.ratio = 4) + # ratio y / x
  labs(x="Origin", y="Pathway")

```

##Hallmark pathway dataframes were generated for all 3 MP2 samples , which were combined manually in Excel as an input for the ggplot script above.

##Begin Panc1 Code Snippets here

```
panc1 <- merge(panc2d, y = panc3d, project = "panc1")

panc1[["percent.mt"]] <- PercentageFeatureSet(panc1, pattern = "^MT-")
panc1 <- subset(panc1, subset = percent.mt < 20)
#FeaturePlot(panc1, features = "percent.mt", pt.size = 1, split.by = "orig.ident", reduction = "umap")

mito.genes <- grep(pattern = "^MT-", x = rownames(x = panc1@meta.data), value = TRUE)

# AddMetaData adds columns to object@meta.data, and is a great place to
# stash QC stats

panc1 <- NormalizeData(object = panc1, normalization.method = "LogNormalize", scale.factor = 10000)
panc1 <- FindVariableFeatures(object = panc1, selection.method = "vst", nfeatures = 2000)
all.genes <- rownames(x = panc1)
panc1 <- ScaleData(object = panc1, verbose = FALSE)
panc1 <- RunPCA(object = panc1, npcs = 30, verbose = FALSE)

#View PCA outputs
print(panc1[["pca"]], dims = 1:5, nfeatures = 10)
VizDimLoadings(panc1, dims = 1:2, reduction = "pca")

# t-SNE and Clustering
panc1 <- RunTSNE(object = panc1, reduction = "pca", dims=1:10)
panc1 <- RunUMAP(object = panc1, reduction = "pca", dims = 1:10)
panc1 <- FindNeighbors(object = panc1, reduction = "pca", dims = 1:10)
panc1 <- FindClusters(panc1, resolution = 0.4)

p1 <- DimPlot(object = panc1, reduction = "umap", group.by = "orig.ident")
p2 <- DimPlot(object = panc1, reduction = "umap", label = TRUE)
samplsplit <- DimPlot(object = panc1, reduction = "umap", split.by = "orig.ident", label = TRUE)
plot_empt <- FeaturePlot(panc1, c("EPCAM", "VIM", "KRT8", "GATA6"), min.cutoff = "q9", reduction="umap")

Idents(panc1) <- panc1$orig.ident
Idents(panc1) <- panc1$seurat_clusters

panc2.markers <- FindMarkers(object = panc1, ident.1 = c(3, 5, 6), min.pct = 0.25)
write.csv(panc2.markers, file = "panc2_markers.csv")
panc3.markers <- FindMarkers(object = panc1, ident.1 = c(0, 1, 2, 4), min.pct = 0.25)
write.csv(panc3.markers, file = "panc3_markers.csv")
```

```

#looking for panc2d-specific markers here. Did not run on 3/18/2021.
#panc2_c3.markers <- FindMarkers(object = panc1, ident.1 = 3, ident.2 = c(0,1,2,4,5,6))
#panc2_c5.markers <- FindMarkers(object = panc1, ident.1 = 5, ident.2 = c(0,1,2,3,4,6))
#panc2_c6.markers <- FindMarkers(object = panc1, ident.1 = 6, ident.2 = c(0,1,2,3,4,5))

#looking for panc3d-specific markers here (comparisons to 2d clusters 2,6,7). Also did not run on 3/18/2021.
#panc3_c0.markers <- FindMarkers(object = panc1, ident.1 = 0, ident.2 = c(1,2,3,4,5,6))
#panc3_c2.markers <- FindMarkers(object = panc1, ident.1 = 2, ident.2 = c(0,1,3,4,5,6))
#panc3_c1.markers <- FindMarkers(object = panc1, ident.1 = 1, ident.2 = c(0,2,3,4,5,6))
#panc3_c12.markers <- FindMarkers(object = panc1, ident.1 = c(1,2), ident.2 = c(0,3,4,5,6))
#panc3_c4.markers <- FindMarkers(object = panc1, ident.1 = 4, ident.2 = c(0,1,2,3,5,6))

#in clue, clusters 2 and 1 have very few markers, i will combine them here. Did not run on 3/18/2021.
#panc3_cx.markers <- FindMarkers(object = panc1, ident.1 = c(1, 2), ident.2 = c(0,3,4,5,6))

#a 3d collection vs a bulky 2d collection
#panc1_subset.markers <- FindMarkers(object = panc1, ident.1 = c(1,3,0), ident.2 = c(2,6,7))

#panc1_subset.markers <- merge(panc3.markers, panc2.markers, by.y = "orig.ident")

#fat csv writing going here
#write.csv(panc2_c3.markers, file = "panc2_c3.csv")
#write.csv(panc2_c5.markers, file = "panc2_c5.csv")
#write.csv(panc2_c6.markers, file = "panc2_c6.csv")
#write.csv(panc3_c0.markers, file = "panc3_c0.csv")
#write.csv(panc3_c1.markers, file = "panc3_c1.csv")
#write.csv(panc3_c2.markers, file = "panc3_c2.csv")
#write.csv(panc3_c4.markers, file = "panc3_c4.csv")

#read in all marker files as lists to be used in GSEA. Example with hallmark below:
hallmark <- gmtPathways("~/Downloads/h.all.v7.0.symbols.gmt")
go <- gmtPathways("~/Downloads/c5.all.v7.0.symbols.gmt")
kegg <- gmtPathways("~/Downloads/c2.cp.kegg.v7.0.symbols.gmt")
c2 <- gmtPathways("~/Downloads/c2.cp.v7.0.symbols.gmt")

```

```

803 #to use fgsea on both sets
804 library(fgsea)
805 library(ggplot2)
806
807 #for cluster panc3d
808 #panc3.markers <- write.csv(panc3.markers, file = "panc3d_markers.csv")
809
810 test4 = panc3.markers
811 test4 <- as.data.frame(test4)
812 test4$y=(-log10(test4$p_val_adj))
813 test4$k=is.infinite(test4$y)
814
815 d4 = test4
816 d4$y[d4$k]=runif(sum(d4$k), max(d4$y[!d4$k]), max(d4$y[!d4$k])*1.25)
817
818 #next is changing the y value to negative if the -log value is, in fact, negative
819 d4$y = ifelse(d4$avg_logFC<0, -d4$y, d4$y)
820
821 #order the d4 by the y
822 d4 = d4[order(d4$y), ]
823
824 #convert the d4 to a matrix
825 mat4 = as.matrix(d4)
826
827 stats4 = mat4[,6] #must read "Named number" when done
828
829 c4_fgseaRes <- fgsea(pathways = hallmark,
830                      stats = stats4,
831                      minSize=5,
832                      maxSize=Inf,
833                      nperm=10000)
834
835

```

```
##### IMPLEMENTING DOTPLOT SCRIPT HERE#####

#made a merged file of all the samples labeled by orig.ident and pathways. This was done by converting the fgsea results into csv files and manipulating them in Excel.

ggplot(panc1_gsea_combined, aes(orig.ident, pathway)) + # change replacement to pathway prn
  geom_point(aes(size = -log10(padj), col = NES)) +
  scale_color_gradientn(colours = c("blue2", "blue", "white", "red", "red2"), na.value = "grey80", limits = c(-1,1)*max(abs(panc1_gsea_combined$NES))) +
  scale_size(name = "FDR",
             breaks = c(-log10(0.005), -log10(0.05), -log10(0.5)),
             labels = c(0.005, 0.05, 0.5)) +
  theme(panel.border=element_blank(),
        panel.background=element_rect(fill="white", color="black", size=0.5, linetype="solid"),
        panel.grid.major=element_blank(),
        panel.grid.minor=element_blank(),
        # plot.title=element_text(size=rel(1.25), hjust=0.5),
        axis.line=element_blank(),
        axis.text=element_text(color="black", size=rel(.75)),
        axis.text.x=element_text(angle=90, vjust=.5),
        axis.title.x=element_text(color="black", size=rel(1), angle=0),
        axis.title.y=element_text(color="black", size=rel(1), angle=90),
        axis.ticks.length=unit(0.1, "cm"),
        axis.ticks=element_line(color="black"),
        legend.key = element_blank(),
        aspect.ratio = 4) + # ratio y / x
  labs(x="Origin", y="Pathway")
```

##Code Snippets complete.

REVIEWER COMMENTS

Reviewer #5 (Remarks to the Author): Expert in ATAC-seq, transcriptomics and bioinformatics

In this manuscript, Monberg and Geiger et al describe a deep molecular characterization of pancreatic cell lines and patient-derived organoids to describe an extensive genomic, transcriptomic, and epigenomic divergence across passages. The authors provide valuable analyses describing extensive genomic alterations resulting from cell culture passages over time, which manifest in a subset of cells, reflect in divergences in transcriptional states, and may be missed by bulk sequencing strategies. The results presented by the authors identify a need for other benchmarking assays besides the traditional microsatellite assays. Overall, the manuscript presents important results that should be considered for future studies with these (and others) cell lines. However, the authors provide little in terms of quality control to help the reader evaluate their analyses, and the description of the methods does not allow the reproducibility of the computational analyses. In addition, all figures need extensive polishing and currently do not allow the reader to reach some of the same conclusions as the authors. Therefore, I recommend major revisions before this manuscript is acceptable for publication in Nature Communications.

Major points

1) The dearth of QC metrics reported for the functional genomics experiments and lack of detailed methods make it hard for the reader to evaluate the reported results in a more detailed manner. The manuscript would greatly benefit from supplementary tables with libraries overview (#cells/nuclei, average number of genes, nUMIs, read coverage, etc). It would be helpful to include nUMI by % mitochondrial scatter plots for barcodes that pass QC in each scRNA-seq library, and nReads by TSS enrichment for the snATAC-seq. See also next comment.

2) The authors performed clustering using diffusion maps to determine batch effects across replicates in the scRNA-seq data. While this is informative to detect gross technical artifacts, it is expected that similar libraries will generally cluster together. However, this approach will likely miss less pronounced effects that lead to differences at the cluster level when integrating across libraries. Therefore, the manuscript would greatly benefit from additional QC metrics of the data to help detect less obvious technical factors potentially driving clustering. These would include UMAP projections colored by the % mitochondrial reads and number of UMIs (scRNA) and number of reads and TSS enrichment (snATAC), all faceted by library. All library integration analysis should include these supporting information (1A, 2A, 2G, 2J, 3A, 3C, 3F, 3G).

3) There seem to be considerably fewer nuclei in the snATAC-seq data in Figures 3F and G compared to 3C. What fraction of snATAC nuclei from Figure 3C were mapped to the scRNA-seq data? If any nuclei

were removed, what was the reason? The authors should clearly describe the criteria in the methods sections.

4) The authors claim that the MiaPaCa2 lines segregate by the laboratory of origin but do not show direct evidence that this result is not from technical differences in sequence coverage and/or cell integrity. These aspects would likely not be captured by the diffusion map approach used by the authors, although it is encouraging to see very short branch heights for these libraries in Supp. Fig 1. My concern is that even after QC, there are still differences in technical aspects across cells. Suppose similar quality cells are clustering together in the shared cluster, but others are forming separate clusters. In that case, it could indicate that the clustering resolution was too high and the algorithm is forcing clusters based on technical aspects such as gene coverage. This can lead to artificial differential gene expression and gene enrichments downstream. It's important to note that unless the libraries were multiplexed directly in the 10X controller (e.g. using 10X CellHash CMOs), there may be technical differences driven by aspects such as background RNA (PMIDs 32138770 and 33367645) even if the cells were processed in the same batch of experiments or nuclei pool. Therefore, extensive exploration of the QC metrics is warranted. The same concerns apply to the Panc1 2D and 3D cultures. The plots requested in comment 2 will allow to determine how much technical differences are driving the clustering results in these analyses.

5) For the integration between scCNV and scRNA-seq, additional QC should be provided. In addition to the QC plots described in the previous comments, the analyses in Figures 2G and J will benefit from a supplementary plot of the gene expression distribution in each cluster for genes associated with the regions with high CNV. One would expect that scRNA-seq barcodes assigned to specific CNVs will have higher gene expression for genes within the CNV region compared to barcodes not assigned to the same CNV cluster. This will help QC the mapping between scCNV and scRNA-seq data made by clonealign and improve confidence in the reported results.

6) Despite comments from other reviewers, the figures are still not up to standard. It took me several attempts to identify some of the findings mentioned in the text in their corresponding figures. The authors should thoroughly polish their figures before this work is suitable for publication. In general, one gets the feeling that the authors copied and pasted raw plot outputs from R and other software directly into the manuscript without any additional polishing, which is not acceptable for a high-impact publication. For example, it is impossible to read any of the axes and color labels in Figure 1 without zooming in. Same for Figures 3D-E. Furthermore, almost half of the main figure panels have at least one axis label missing (Figures 1C-G, 2C,E,H, 3B,D,E, 4C,D). Figure 1 is particularly egregious: panel C is a confusing collage of heatmaps with no x and y labels, and a symbol legend (copy num, read depth, etc.) that does not relate in any way the data presented in the plots in 1C (and may be related to panels F and G, instead). Panels F and G, besides not having any axes labels, have no color scale (which one has to guess is shared with panel C) and have line plots in the bottom with no axes or annotations indicating what the numbers mean, which the reader is once again left to guess their meaning. Panels F and G also have plots depicting the gene expression of BRCA1 and AURKA genes. The plots are overplotted and

depict bottom-heavy violin plots without any statistical analyses, which does not allow the reader to confidently determine if the extent of variation in the gene expression levels is related to CNVs.

7) The authors should better annotate figures to guide the reader and support their main claims. In several cases, the reader is left to take authors on their word for the conclusions. For instance, Figure 1G besides not having any axes labels, also has no mention of which region in the plot corresponds to chr20q or the AURKA locus. The authors mention a "purple bar" in the main text, which I believe is the region with a high "Het" value in the plot. Why not clearly label that region, instead? When space allows, authors should avoid unnecessary abbreviations in legends, such as the aforementioned "Het" label, to add less work to the reader. Similarly, the authors mention "evidence for a highly similar ancestral clone" (page 9) for the MiaPaCa2 strains. Yet, there is no corresponding annotation of this in Figure 1C nor any way to help the reader interpret Figure 2D with this finding in mind. Making these changes will make the manuscript more appealing to the broad readership of Nature Communications.

Minor points

1) Figures are unnecessarily large due to abundant whitespace, and font sizes vary wildly across figure panels.

2) The order of the figures should be changed according to the text to improve flow (Figure 1D is mentioned after Figure 1G, Figure 1G is mentioned after 1F, etc). This unnecessary back-and-forth makes the manuscript confusing for the reader.

3) The authors use non-quantitative terms throughout the text to describe their results: "a swath of cells" (page 6). What fraction of cells does "a swath" means? Other examples include "tranquil" (page 5), "nearly all cells" (page 9), "relatively neutral" (page 7), "correspond almost perfectly" (page 10). Besides their non-quantitative nature, some of these terms are open to subjective interpretation ("relative genomic quiescence," "tranquil"). These terms should be substituted for accurate quantitative data representations (percentages, effect sizes, and p-values when appropriate).

4) The authors should include volcano plots for the DE analyses to help QC results and understand why they chose to report the subset of genes on page 10. All DE results should be reported in a separate supplementary table.

5) Supp table 1 further adds to the overall feeling of unpolishedness of the manuscript: it is a screenshot of an excel table where the spellcheck red underline is visible.

6) Supp Fig 1 should have a label to the color scale.

REVIEWER COMMENTS

Reviewer #5 (Remarks to the Author): Expert in ATAC-seq, transcriptomics and bioinformatics

In this manuscript, Monberg and Geiger et al describe a deep molecular characterization of pancreatic cell lines and patient-derived organoids to describe an extensive genomic, transcriptomic, and epigenomic divergence across passages. The authors provide valuable analyses describing extensive genomic alterations resulting from cell culture passages over time, which manifest in a subset of cells, reflect in divergences in transcriptional states, and may be missed by bulk sequencing strategies. The results presented by the authors identify a need for other benchmarking assays besides the traditional microsatellite assays. Overall, the manuscript presents important results that should be considered for future studies with these (and others) cell lines. However, the authors provide little in terms of quality control to help the reader evaluate their analyses, and the description of the methods does not allow the reproducibility of the computational analyses. In addition, all figures need extensive polishing and currently do not allow the reader to reach some of the same conclusions as the authors. Therefore, I recommend major revisions before this manuscript is acceptable for publication in Nature Communications.

Major points

1) The dearth of QC metrics reported for the functional genomics experiments and lack of detailed methods make it hard for the reader to evaluate the reported results in a more detailed manner. The manuscript would greatly benefit from supplementary tables with libraries overview (#cells/nuclei, average number of genes, nUMIs, read coverage, etc). It would be helpful to include nUMI by % mitochondrial scatter plots for barcodes that pass QC in each scRNA-seq library, and nReads by TSS enrichment for the snATAC-seq. See also next comment.

2) The authors performed clustering using diffusion maps to determine batch effects across replicates in the scRNA-seq data. While this is informative to detect gross technical artifacts, it is expected that similar libraries will generally cluster together. However, this approach will likely miss less pronounced effects that lead to differences at the cluster level when integrating across libraries. Therefore, the manuscript would greatly benefit from additional QC metrics of the data to help detect less obvious technical factors potentially driving clustering. These would include UMAP projections colored by the % mitochondrial reads and number of UMIs (scRNA) and number of reads and TSS enrichment (snATAC), all faceted by library. All library integration analysis should include these supporting information (1A, 2A, 2G, 2J, 3A, 3C, 3F, 3G).

We thank the reviewer for their attention to data quality and commitment to ensuring that our analysis is up to par. Given the similarity in content of Points 1 and 2, we have responded to both comments below.

We agree that diffusion maps alone do not fully address batch effect mitigation! As described in our Methods, we filtered out all cells where mitochondrial genes exceeded 20%. The cells analyzed in our presented UMAPs are only those cells that met QC criteria.

- Regarding the QC of snATACseq data, we kindly direct the Reviewer's attention to **Supplementary Figure 7**, where we have included in our submission the plots they are requesting here. For QC of scRNAseq data, we agree that it would be generally helpful to include graphs in our supplementary data, but given our explanation of QC Methods (described in our submitted methods) for filtering criteria, for the sake of manuscript "flow" we do not wish to also incorporate these data elements. However, to ensure the reviewer that adequate sequencing depth, cell count, and mitochondrial genes were taken into account, we have included Reviewer-only Figures and Tables below.

- Data for the number of cells and sequencing depth per sample is contained in Reviewer-only Revisions Table 1 below, and QC metrics for scRNAseq data in Reviewer-only Revisions Figure 1. In our own recently published work, we also demonstrate that few cells from extremely limited tissue biopsies, not sequenced in technical replicates, were adequate to predict therapeutically-relevant inter-cellular interactions at the single-cell RNA level [Ref. 1]. In Main Figure 1A, cells from each sample cluster together, a UMAP organization based on biological identity that is supported by another PDAC preclinical models paper recently published in Nature Communications [Ref. 2], that strongly suggests that batch effect does not drive the organization of cells in the scRNAseq analysis assay. To further confirm the limited influence of an assumed batch effect in a quantitative way in our scRNAseq dataset, we show in Reviewer-only Revisions Figure 3 below, by Pearson Correlation instead of Diffusion Space analysis, that across the scRNAseq datasets, the cells within each sample form their own “clades”, demonstrating that the cells are clustering as a function of biological similarity and not as a function of technical artifacts. If there was a batch effect present, we would see samples sequenced together in individual runs clustering in the same families. Instead, we see samples clustering in a way that supports our hypotheses of biological identities across the cell lines.
- To address additional concerns of bath effect mitigation, we also provide here a secondary analysis using scMC [Ref. 3] for scRNAseq sample integration. We do not wish to include these results in the main text of our manuscript, as we do not wish to redirect the point of our paper towards a “methods comparison” for scRNAseq analysis. However, scMC was developed after criticism of Seurat-based normalization in the field, where Seurat algorithms were actually a culprit of grouping samples by technical instead of biological similarity. scMC instead works to mitigate biological variation over technical variation, uses a different method for identifying clusters and cellular neighbors than Seurat, and has been appreciated in the literature for its accuracy at structuring UMAP groups based on cell type. With scMC, we show that the overall cluster structure (which samples group together) is preserved from what we report in Main Figure 1. These results are shown below in Reviewer-Only Revisions Figure 3. This appears to be aligned with the Reviewer’s primary concern in our analysis, and we hope that this analysis addresses “biological versus technical” worries in our data.
- In Main Figures 2 and 3, where we dissect heterogeneity between MiaPaca2 and Panc1 samples, we show that clustering or UMAP separation at the scRNAseq level is driven by transcriptomic consequences of differential genomic events in the case of MiaPaca2 cells, and chromatin accessibility in the case of Panc1 cultures, rather than “technical variation” within the samples themselves. In this regard, the orthogonally-collected genomic and chromatin data are used to draw conclusions about the biological differences between the cell lines described, and the scRNAseq data is used to “confirm” what is found to be significantly different in DNA-based assays.
- Given that cell lines are inherently less heterogeneous than tissue biopsies, that each sample in this study was sequenced to a depth well above the “minimum required” read count, and that the very nature of single-cell profiling guarantees hundreds of technical replicates within each sample, we are confident that the data presented here does truly capture the biological heterogeneity across these cell lines.

Reviewer-only Revisions Table 1. Sequencing metrics for scRNAseq data, provided in response to concerns raised by Reviewer #5.

Cell line	3' RNA			
	# cells targeted	# cells sequenced	Mean reads/cell	Median genes/cell
HPNE	10,000	12,810	12,458	1,809
HPDE	10,000	5,233	13,831	1,941
BxPc3	10,000	10,405	12,426	1,819
HPAF-II	10,000	5,104	27,166	1,989
Panc1- 3D	10,000	23,118	11,828	780
Panc1- 2D	10,000	5,559	3,407	3,407
MP2-A	10,000	9,913	6,372	1,191
MP2-B	10,000	7,171	9,811	1,663
MP2-C	10,000	11,177	6,209	1,083

Reviewer-only Revisions Figure 1. QC of scRNAseq data of PDAC cell lines showing RNA count, mitochondrial gene percentages, and number of features per sample.

Reviewer-only Revisions Figure 2. Pearson correlation analysis of cell lines to corroborate reported findings of Diffusion Space scRNAseq analysis. To address reviewer comments regarding technical artifacts driving the clustering of samples, we show that inclusion a technical replicate of Panc-1 2D sequenced from a different batch is ranked as highly similar to the Panc-1 samples described in our manuscript by both cladogram organization (A) and heatmap (B). This provides evidence that cell number and sequencing depth in the original presented dataset are sufficient to both ameliorate assumed technical/batch effects and to proscribe against generation of technical replicates for all samples.

Reviewer-only Revisions Figure 3. A) scMC Integration-based UMAP, wherein sample organization based on a biologically-informed algorithm shows sample grouping identical to what is depicted in B) Main Figure 1A, providing evidence to support that clustering and gene expression analysis is not a function of technical variation in the data.

3) *There seem to be considerably fewer nuclei in the snATAC-seq data in Figures 3F and G compared to 3C. What fraction of snATAC nuclei from Figure 3C were mapped to the scRNA-seq data? If any nuclei were removed, what was the reason? The authors should clearly describe the criteria in the methods sections.*

- We agree with the reviewer that the number of cells used in snATACseq may appear quite small. However, rigorous quality control metrics were indeed run on the data, the results of which are included in supplemental data (**Supplementary Fig. 7A, 7B**), and are also attached here (Reviewer-only Revisions Figure 4) for the reviewer's consideration. We subset nuclei according to the following criteria: TSS Enrichment > 2, nucleosome signal < 10, blacklist ratio < 0.05, percent reads in peaks > 15, peak region fragments between 1,000 and 20,000. The methods have been amended to reflect these parameters. After ensuring the quality of nuclei were sufficient, as proscribed by the Signac Package developed by the Satija Lab (cited in Main Text and discussed in Methods) as well as recent publications including this data type [Refs. 4-5], a final count of 9,785 nuclei are represented by the Panc1 2D sample, and 7,453 nuclei of Panc1 3D are depicted in the figures. As discussed above, the resulting number of nuclei is more than sufficient, by current research standards, to draw accurate conclusions and conduct correlative investigations across samples in this dataset.
- As per the scRNAseq data quality control metrics, those are included in our submission to the GEO portal and are now publicly available. It is not common practice in the literature to list quality control metrics of scRNAseq data in the supplementary or main figures, so we did not include them in this manuscript. However, cell count and sequencing depth per scRNAseq sample are listed in Reviewer-only Revisions Table 1.

Reviewer-only Revisions Figure 4. Quality control data of nuclei sequenced in snATACseq assay for Panc-1 3D (A, B), and Panc-1 2D (C, D) samples. TSS Enrichment Plots for Panc-1 3D (B) and Panc-1 2D (D) depicting the mean TSS enrichment score across nuclei are shown. Nuclei data were subsequently filtered according to appropriate QC parameters.

A. Panc-1 3D snATAC QC

C. Panc-1 2D snATAC QC

4) The authors claim that the *MiaPaCa2* lines segregate by the laboratory of origin but do not show direct evidence that this result is not from technical differences in sequence coverage and/or cell integrity. These aspects would likely not be captured by the diffusion map approach used by the authors, although it is

encouraging to see very short branch heights for these libraries in Supp. Fig 1. My concern is that even after QC, there are still differences in technical aspects across cells. Suppose similar quality cells are clustering together in the shared cluster, but others are forming separate clusters. In that case, it could indicate that the clustering resolution was too high and the algorithm is forcing clusters based on technical aspects such as gene coverage. This can lead to artificial differential gene expression and gene enrichments downstream. It's important to note that unless the libraries were multiplexed directly in the 10X controller (e.g. using 10X CellHash CMOs), there may be technical differences driven by aspects such as background RNA (PMIDs 32138770 and 33367645) even if the cells were processed in the same batch of experiments or nuclei pool. Therefore, extensive exploration of the QC metrics is warranted. The same concerns apply to the Panc1 2D and 3D cultures. The plots requested in comment 2 will allow to determine how much technical differences are driving the clustering results in these analyses.

- The reviewer makes an excellent point that scRNAseq data can be insufficient to determine whether technical artifacts are driving data behavior; however, we have provided our QC metrics above, and hope that these data are convincing enough of the quality of the sequencing results we describe.
- With MiaPaca2 samples, while we do make a note of describing differences in the scRNAseq of each sample and conducted GSEA for pathway relevance, we provide evidence through orthogonally-collected scCNVseq data that differences in MP2 strains are likely due to genomic events specifically at chrs. 5, 10, 7, 12, and 14, as depicted in Main Figure 2. We use scRNAseq data to confirm gene expression for corresponding scCNV-determined scRNA clusters. In this regard, even if there were “technical” differences influencing the construction and analysis of scRNAseq libraries, the likelihood of a technical artifact/difference inducing differential amplifications and deletions at the chromosome level across these samples is highly unlikely.

5) For the integration between scCNV and scRNA-seq, additional QC should be provided. In addition to the QC plots described in the previous comments, the analyses in Figures 2G and J will benefit from a supplementary plot of the gene expression distribution in each cluster for genes associated with the regions with high CNV. One would expect that scRNA-seq barcodes assigned to specific CNVs will have higher gene expression for genes within the CNV region compared to barcodes not assigned to the same CNV cluster. This will help QC the mapping between scCNV and scRNA-seq data made by clonealign and improve confidence in the reported results.

- We agree. In Figures 2F and 2I, we provide this information in a different way! These figures are a mapping of the percentage of cells/UMIs in each assay (CNV and RNA) that correspond to clonal assignments (demarcated by colors green, yellow, or blue). Clonealign does this by working with a gene x clone matrix that we derive, as described in our Methods. We have attached a copy of our clonealign workflow that we ran for Panc1 and MP2 samples (Reviewer-only Revisions Figure 5). As the reviewer may appreciate, there are many intermediary QC steps for running this analysis and determining the confidence with which clones are called. We do not wish to include metrics from each step of this pipeline in our manuscript submission, but will be happy to provide workflow information and QC metrics to any laboratory/individual who wishes to recapitulate our results, as our data is also now publicly available.
- In order to understand which genes are associated with clone-defined clusters, we provide **Supplementary Table 2**. In this table, we show the results of significantly enriched GSEA pathways in each clone, as defined per sample by specific chromosomal events. If this is insufficient to address Reviewer's concern, and the exact gene lists used as inputs for GSEA would be better to demonstrate these results, we are happy to provide those gene lists for review.

Reviewer-only Revisions Figure 5: Clonealign workflow applied for calling scRNA clusters from scCNV-derived clonal events. This workflow follows the published guidelines for implementing clonealign and will not be included in our submitted manuscript.

Part 1

Part 2

6) *Despite comments from other reviewers, the figures are still not up to standard. It took me several attempts to identify some of the findings mentioned in the text in their corresponding figures. The authors should thoroughly polish their figures before this work is suitable for publication. In general, one gets the feeling that the authors copied and pasted raw plot outputs from R and other software directly into the manuscript without any additional polishing, which is not acceptable for a high-impact publication. For example, it is impossible to read any of the axes and color labels in Figure 1 without zooming in. Same for Figures 3D-E. Furthermore, almost half of the main figure panels have at least one axis label missing (Figures 1C-G, 2C,E,H, 3B,D,E, 4C,D). Figure 1 is particularly egregious: panel C is a confusing collage of heatmaps with no x and y labels, and a symbol legend (copy num, read depth, etc.) that does not relate in any way the data presented in the plots in 1C (and may be related to panels F and G, instead). Panels F and G, besides not having any axes labels, have no color scale (which one has to guess is shared with panel C) and have line plots in the bottom with no axes or annotations indicating what the numbers mean, which the reader is once again left to guess their meaning. Panels F and G also have plots depicting the gene expression of BRCA1 and AURKA genes. The plots are overplotted and depict bottom-heavy violin plots without any statistical analyses, which does not allow the reader to confidently determine if the extent of variation in the gene expression levels is related to CNVs.*

- In our submission process, we have created all figures from vectorized JPEG-7000s in Adobe Illustrator after exporting data from R at the highest-quality allowed by the software. We then adjusted the resolution to 1000 ppi. We have tried our best to polish the figures as much as possible, and hope this is now more acceptable to the reviewer.
- For figure labels, especially regarding 1C: we had included in the submitted Figure Legend the axes: "Representative scCNV plots for cell lines. Columns indicate chromosomes, rows indicate individual cells." These heatmaps are exactly that: cells (in rows) are phylogenetically organized and events on a per-chromosome level are depicted. Heatmap legend itself contains the color-coding for the events. We have enlarged Figure 1 and split it into 2 pages- hopefully now it is easier to read. Similarly for Figures 1G-H, we have redone the labeling and hope that everything is now more clear.
- In order to be more quantitative of our assessment of BRCA1 and AURKA expression, we have replaced the Violin Plots in the original Figure 1 with a DotPlot, clearly stating the percentage of cells in each population that express the specific genes, and their relative levels of expression. This is now in Figure 1, continued page, labeled as Figure 1H. Thank you for helping us to be more precise in the presentation of our data!

7) *The authors should better annotate figures to guide the reader and support their main claims. In several cases, the reader is left to take authors on their word for the conclusions. For instance, Figure 1G besides not having any axes labels, also has no mention of which region in the plot corresponds to chr20q or the AURKA locus. The authors mention a "purple bar" in the main text, which I believe is the region with a high "Het" value in the plot. Why not clearly label that region, instead? When space allows, authors should avoid unnecessary abbreviations in legends, such as the aforementioned "Het" label, to add less work to the reader. Similarly, the authors mention "evidence for a highly similar ancestral clone" (page 9) for the MiaPaCa2 strains. Yet, there is no corresponding annotation of this in Figure 1C nor any way to help the reader interpret Figure 2D with this finding in mind. Making these changes will make the manuscript more appealing to the broad readership of Nature Communications.*

- We apologize for the confusion in our figures and have diligently worked to address the reviewer's concerns. Specifically, we have clarified in the language of the text what is depicted in Figure 1G in both the manuscript text and the figure legend. This is actually a "zoomed in" scCNV heatmap of chr20,

which can be generated with the 10x Genomics scCNV analysis software cited in our Methods. The “x-axis”, so to speak, is labeled with chromosome regions within chr20, showing ranges from 8-16+ copy numbers of that region across HPNE cells (the rows of the heatmap). We apologize for the small labels that had been in our initial figure version, and hope that this addresses the reviewer’s concern.

- The figure legend for Figure 1 now reads as screenshotted below:

MAIN

Figure 1: PDAC Cell Lines display heterogeneity at single-cell level

A) Uniform manifold approximation projection (UMAP) plot of single cells from PDAC cell lines used in this study.

B) UMAP Feature Plotting for known cell-type markers (EPCAM, MUC1 for epithelial; vimentin, and KRT8 for mesenchymal cellular origin).

C) Representative scCNV plots for cell lines. Columns indicate chromosomes (numbers labeled in gray and white), rows indicate individual cells organized into clonal clades. Phylogeny determined using 10x Genomics scCNV analysis software.

D) scCNV comparison to WES of HPNE cell line depicting amplified 17q as the only notable CNV event.

E) scCNV comparison to WES of HPDE cell line. CNV events representing losses of chromosomal arms 3p, 8p, and 9p and amplifications at chromosome 20.

F) scCNV high-resolution cell phylogeny of HPNE for all chromosome 17 locations showing ploidy = 3 for all cells at 17q arm, region inclusive of BRCA1 loci. Columns indicate chromosomal intervals measured in Mb, rows indicate individual cells. Chromosomal regions depicted (labeled along top x-axis) are representative of CNV segments outlined in red in Figure 1D.

G) scCNV high-resolution cell phylogeny of HPDE for all chromosome 20 locations showing ploidy >3 (as high as 13 in some cells at some locations) for all cells. Columns indicate chromosomal intervals measured in Mb, rows indicate individual cells. Chromosomal regions depicted (labeled along top x-axis) are representative of CNV segments outlined in red in Figure 1E. Corresponding scRNA data shows elevated expression of AURKA, located within amplified HPDE region as a potential target of amplification.

I) scRNAseq data of HPNE and HPDE cell lines shows elevated expression of BRCA1 in >20% of HPNE cells, increased AUKRA expression in >20% of HPDE cells.

Minor points

- 1) Figures are unnecessarily large due to abundant whitespace, and font sizes vary wildly across figure panels.
- 2) The order of the figures should be changed according to the text to improve flow (Figure 1D is mentioned after Figure 1G, Figure 1G is mentioned after 1F, etc). This unnecessary back-and-forth makes the manuscript confusing for the reader.

- Thank you, we have adjusted figure label/flow wherever we could without disrupting the reporting of results.

3) *The authors use non-quantitative terms throughout the text to describe their results: "a swath of cells" (page 6). What fraction of cells does "a swath" means? Other examples include "tranquil" (page 5), "nearly all cells" (page 9), "relatively neutral" (page 7), "correspond almost perfectly" (page 10). Besides their non-quantitative nature, some of these terms are open to subjective interpretation ("relative genomic quiescence," "tranquil"). These terms should be substituted for accurate quantitative data representations (percentages, effect sizes, and p-values when appropriate).*

- The reviewer is correct, thank you very much. We have adjusted our language throughout the text to be more quantitative and professional in tone.

4) *The authors should include volcano plots for the DE analyses to help QC results and understand why they chose to report the subset of genes on page 10. All DE results should be reported in a separate supplementary table.*

- On page 10, we describe the results of GSEA enrichment in scCNV-scRNA clones, as determined by extensive clonealign filtering and analysis (see above, and Methods). The results of these analyses are further divulged in Supplementary Table 1. The genes that were used as inputs for lists in clonealign-based GSEA were deemed significant not by a "classical" DE gene analysis, but instead through extensive clonealign analysis. From the GSEA output of significant pathway, we manually selected genes that were important to pancreatic cancer biology to discuss in the text of the manuscript. In this case, we did not select these genes based on their distribution in a volcano plot, but instead as a function of CNV-RNA-GSEA significance cutoffs. We are happy to provide full gene lists for all analyses conducted for review, but do not wish to include those lists in the published manuscript. We have adjusted the language in the Methods section to describe how genes were chosen. The Methods now read:

- **GSEA of clonealign-defined scRNA clones**

Pre-ranked GSEA was thus performed via the GSEA software version GSEA_4.0.3. Molecular Signature Databases h.all.v7.1.symbols_1.gmt (hallmark pathways) and c6.all.v7.2.symbols.gmt (oncogenic signature gene sets) were both used to align gene lists extracted from clonealign-defined-scRNA clones in Seurat (see figures 2G, 2J, Supp. 6C, Supp. 6D), as described above using FindMarkers. A false discovery rate (FDR) cutoff of 25% was used to analyze the GSEA results for significant pathway enrichment per sample. Manual curation of genes lists from significantly-enriched per-clone pathways was conducted to identify individual genes relevant to pancreatic cancer biology (ie KRT13, LGALS1).

5) *Supp table 1 further adds to the overall feeling of unpolishedness of the manuscript: it is a screenshot of an excel table where the spellcheck red underline is visible.*

- Thank you for catching this, we have revised the table.

6) *Supp Fig 1 should have a label to the color scale.*

- Agreed, the meaning of the numbers on the color scale is now described in the figure legend of Supp Fig 1.

References Cited For Reviewer 5

1. Lee, J. J., et al. (2021). "Elucidation of tumor-stromal heterogeneity and the ligand-receptor interactome by single cell transcriptomics in real-world pancreatic cancer biopsies." *Clin Cancer Res*.
2. Krieger, T. G., et al. (2021). "Single-cell analysis of patient-derived PDAC organoids reveals cell state heterogeneity and a conserved developmental hierarchy." *Nature Communications* 12(1): 5826.
3. Zhang, L. and Q. Nie (2021). "scMC learns biological variation through the alignment of multiple single-cell genomics datasets." *Genome Biol* 22(1): 10.
4. Krieger, T. G., et al. (2021). "Single-cell analysis of patient-derived PDAC organoids reveals cell state heterogeneity and a conserved developmental hierarchy." *Nat Commun* 12(1): 5826.
5. Hosein, A. N., et al. (2021). "Loss of Rnf43 accelerates Kras-mediated neoplasia and remodels the tumor immune microenvironment in pancreatic adenocarcinoma." *Gastroenterology*.

REVIEWERS' COMMENTS

Reviewer #5 (Remarks to the Author):

* The scMC analysis performed by the authors in Reviewer Fig. 3 is very helpful and addresses my concerns about technical variation. I also share the concern regarding Seurat's (and Harmony's) clustering being driven by technical aspects, which is why I requested additional QC exploration.

* The authors did not clarify how many nuclei were mapped to the scRNA-seq data in Figs 3F-G. The authors reply "a final count of 9,785 nuclei are represented by the Panc1 2D sample, and 7,453 nuclei of Panc1 3D are depicted in the figures". There are certainly not 7-9K ATAC nuclei in Figs 3 F-G. At most, there are a few hundred on each ATAC panel. Therefore, the vast majority of ATAC nuclei that passed QC in Fig 3C were not mapped to the scRNA-seq data. Again, I ask what fraction of snATAC nuclei from Figure 3C were mapped to the scRNA-seq data? If any nuclei were removed after integration, what was the reason? If only a minuscule fraction of ATAC nuclei mapped to the RNA data, then the ATAC-RNA integration most likely failed and the authors lack evidence to support their claim that "and [we] confirmed the overexpression of putative target candidate genes affiliated with the enriched motifs that we had identified in corresponding snATAC-seq datasets" (page 14).

* The figure legend of Supp. Fig. 7 does not seem to correspond to the figure panels or it is out of order.

* All other items were satisfactorily addressed. I believe the authors could still do simple changes to make the manuscript figures more interpretable to the reader, such as adding axis labels directly in the plots instead of the figure legends and adding additional annotations to guide the reader (e.g. gene tracks for AURKA and BRCA1 genes in Figs 1F-G).

REVIEWERS' COMMENTS

Reviewer #5 (Remarks to the Author):

* The scMC analysis performed by the authors in Reviewer Fig. 3 is very helpful and addresses my concerns about technical variation. I also share the concern regarding Seurat's (and Harmony's) clustering being driven by technical aspects, which is why I requested additional QC exploration.

* The authors did not clarify how many nuclei were mapped to the scRNA-seq data in Figs 3F-G. The authors reply "a final count of 9,785 nuclei are represented by the Panc1 2D sample, and 7,453 nuclei of Panc1 3D are depicted in the figures". There are certainly not 7-9K ATAC nuclei in Figs 3 F-G. At most, there are a few hundred on each ATAC panel. Therefore, the vast majority of ATAC nuclei that passed QC in Fig 3C were not mapped to the scRNA-seq data. Again, I ask what fraction of snATAC nuclei from Figure 3C were mapped to the scRNA-seq data? If any nuclei were removed after integration, what was the reason? If only a minuscule fraction of ATAC nuclei mapped to the RNA data, then the ATAC-RNA integration most likely failed and the authors lack evidence to support their claim that "and [we] confirmed the overexpression of putative target candidate genes affiliated with the enriched motifs that we had identified in corresponding snATAC-seq datasets" (page 14).

Our apologies! We had misunderstood your comment in the previous revisions round, and hope we can now properly address this!

- *The reviewer is correct in their observation that not all nuclei mapped to RNA data. This was due to more stringent QC requirements in Seurat's merging algorithm. However, the "merged" analysis was not actually used to assay the putative gene targets described and depicted in scRNAseq dotplots in the bottom panels for figure 3, and the confusing explanation of that analysis is entirely our fault. As such, we have removed verbiage from the manuscript describing the merged analysis, and have kept the snATACseq-assay figure panels separated from the scRNAseq-assay figure panels. Now, each assay serves as an "orthogonal validation" for the other, and the methods section regarding this alteration has been amended.*

* The figure legend of Supp. Fig. 7 does not seem to correspond to the figure panels or it is out of order.

Thank you very much for bringing this to our attention, and we apologize for the oversight. We have adjusted the figure legend.

* All other items were satisfactorily addressed. I believe the authors could still do simple changes to make the manuscript figures more interpretable to the reader, such as adding axis labels directly in the plots instead of the figure legends and adding additional annotations to guide the reader (e.g. gene tracks for AURKA and BRCA1 genes in Figs 1F-G).